# DISTRIBUTION-DEPENDENT RATES FOR MULTI-DISTRIBUTION LEARNING

## ABSTRACT

To address the needs of modeling uncertainty in sensitive machine learning applications, the setup of distributionally robust optimization (DRO) seeks good performance uniformly across a variety of tasks. The recent multi-distribution learning (MDL) framework Awasthi et al. (2023) tackles this objective in a dynamic interaction with the environment, where the learner has sampling access to each target distribution. Drawing inspiration from the field of pure-exploration multi-armed bandits, we provide *distribution-dependent* guarantees in the MDL regime, that scale with suboptimality gaps and result in superior dependence on the sample size when compared to the existing distribution-independent analyses. We investigate two non-adaptive strategies, uniform and non-uniform exploration, and present non-asymptotic regret bounds using novel tools from empirical process theory. Furthermore, we devise an adaptive optimistic algorithm, LCB-DR, that showcases enhanced dependence on the gaps, mirroring the contrast between uniform and optimistic allocation in the multi-armed bandit literature.

## 1 INTRODUCTION

Classical statistical learning operates under the assumption that data comes from a single source Hastie et al. (2009). However, the growing use of machine learning in safety-critical applications has brought forth the demand for more robust models that address stochastic heterogeneity. One well-established paradigm is *distributionally robust optimization (DRO)* Rahimian & Mehrotra (2022), which seeks good performance uniformly across a collection of distributions. Concretely, let $\mathcal{A}$ and $\mathcal{X}$ be decision and data spaces, respectively, and suppose that data is sampled from a distribution within some *uncertainty set* $\mathcal{U} \subset \mathcal{P}(\mathcal{X})$. Under a target reward function $r : \mathcal{A} \times \mathcal{X} \to \mathbb{R}$ and distribution $Q \in \mathcal{U}$, an action $a \in \mathcal{A}$ yields expected reward $\mu(a; Q) \coloneqq \mathbb{E}_{X_Q \sim Q}[r(a, X_Q)]$. DRO then focuses on the problem

$$\max_{a \in \mathcal{A}} \left\{ \mu_{\mathrm{DR}}(a) \coloneqq \min_{Q \in \mathcal{U}} \mu(a; Q) \right\} \tag{DR}$$

Recent works Blum et al. (2017); Sagawa* et al. (2020); Haghtalab et al. (2022) have studied the setting of finite $\mathcal{U}$ and tackle it via interactive dynamics with the environment. More precisely, the emergent *multi-distribution learning (MDL)* framework Awasthi et al. (2023) assumes sampling access to $\mathcal{U}$, where a learning agent sequentially selects which distributions to sample from given a fixed sampling budget.

The current literature is populated with *distribution-independent* rates; i.e., bounds that are independent of problem parameters. While broad in its applicability, this approach falls short in capturing the nuances of the underlying environment. Oftentimes, it is more intuitive to analyze the learner's performance in a fixed setting, as opposed to considering a worst-case instance for each sample size. When domain knowledge is available, a "one-size-fits-all" rate does not provide any insight on how to take advantage of this information.

To address these drawbacks, in this work, we study *distribution-dependent* guarantees for the MDL problem. Motivated by its close ties to the well-studied *pure exploration multi-armed bandits (PE-MAB)* Bubeck et al. (2011) paradigm, we analyze the simple strategies of uniform and non-uniform exploration, as well as their optimistic counterpart, ensuring regret guarantees that scale with suboptimality gaps and decay much faster with the sampling budget.

## 1.1 MAIN RESULTS

We place MDL algorithms into one of two categories: non-adaptive and adaptive. In the former, data is collected without any interaction with the environment and, in the latter, the learner sequentially selects distributions based on previously acquired samples. We introduce two strategies of the non-adaptive type: uniform (UE) and non-uniform (NUE) exploration (Section 3). As the names suggest, UE gathers the same number of samples from each distribution, while NUE can benefit from varied sample sizes. Using tools from empirical process theory, we provide non-asymptotic regret guarantees that scale with the suboptimality gaps of the problem and decay exponentially with the sampling budget $T$ (Section 3.1). This stands in contrast to the distribution-independent rates found in the recent literature, which hold under a worst-case environment and, thus, only scale with $O\left(\frac{1}{\sqrt{T}}\right)$. From a novel Bernstein-type concentration inequality, we then show how NUE can exploit distributional variability to allocate samples more effectively (Section 3.2).

While the non-adaptive methods already display exponentially decreasing regret, adaptivity can further improve the dependence on instance-specific variables. Motivated by the enhancements of UCB-E Audibert et al. (2010) over uniform exploration in the PE-MAB literature, we introduce the analogous LCB-DR algorithm (Section 4) and showcase how optimism can result in superior dependence on the suboptimality gaps when compared to UE (Section 4.1).

Let $\Delta_{\mathrm{DR}}(a) \coloneqq \max_{a' \in \mathcal{A}} \mu_{\mathrm{DR}}(a') - \mu_{\mathrm{DR}}(a)$ be the suboptimality gap of an action from a finite set $\mathcal{A}$. Given an algorithm, we denote its output after $T$ sampling rounds by $A_T^{\mathrm{o}}$. In short, we make the following contributions: assuming bounded rewards $r \in [0, 1]$ for (i) and (iii),

(i) With $n \in \mathbb{N}$ samples from each distribution, we show in Section 3.1 that UE has a simple regret decay of order

$$\mathbb{E}\left[\Delta_{\mathrm{DR}}\left(A_T^{\mathrm{o}}\right)\right] \leq \sum_{a \in \mathcal{A} : \Delta_{\mathrm{DR}}(a) > 0} \Delta_{\mathrm{DR}}(a) \exp\left(-n \Delta_{\mathrm{DR}}^2(a)\right)$$

Moreover, we present the distribution-independent rate $\mathbb{E}\left[\Delta_{\mathrm{DR}}\left(A_T^{\mathrm{o}}\right)\right] \lesssim \sqrt{\frac{|\mathcal{U}| \log(|\mathcal{U}||\mathcal{A}|)}{T}}$.

(ii) With $n_Q \in \mathbb{N}$ samples from distribution $Q \in \mathcal{U}$ over real-valued data and bounded reward $r \in [0, M]$, we show in Section 3.2 that NUE attains the rate

$$\mathbb{E}\left[\Delta_{\mathrm{DR}}\left(A_T^{\mathrm{o}}\right)\right] \leq \sum_{a \in \mathcal{A} : \Delta_{\mathrm{DR}}(a) > 0} \Delta_{\mathrm{DR}}(a) \exp\left(-\frac{\Delta_{\mathrm{DR}}^2(a)}{\sigma_T^2 + \Sigma_T^2 + V_T + \frac{M \Delta_{\mathrm{DR}}(a)}{\min_{Q \in \mathcal{U}} n_Q}}\right)$$

where $\sigma_T^2, \Sigma_T^2$ and $V_T$ are empirical process variance quantities that scale with the variances of each $Q \in \mathcal{U}$ and decrease with the $n_Q$.

(iii) Appealing to the principle of optimism, we devise the LCB-DR algorithm that, in a pre-specified permutation of the arms $\left(a_1, a_2, \ldots, a_{|\mathcal{A}|}\right)$, for $j = 1, \ldots, |\mathcal{A}|$, sequentially performs a modified version of UCB-E, for $T_j$ rounds, on "losses" $\{\mu(a_j, Q)\}_{Q \in \mathcal{U}}$ as a means of identifying the worst-case distribution for $a_j$. In Section 4, we show that this guarantees an error probability of

$$\mathbb{P}\left(\Delta_{\mathrm{DR}}\left(A_T^{\mathrm{o}}\right) > 0\right) \leq \sum_{j=1}^{|\mathcal{A}|} \exp\left(-\frac{\left(C_{a_j}^2 \wedge 1\right)\left(T_j + \tilde{T}_j - |\mathcal{U}_j|\right)}{H_j}\right)$$

This bound, which may be of independent interest, results from an analysis of UCB-E under a learner with previously acquired data (see Appendix D). Since the learner has already accumulated samples from previous iterations in each UCB-E batch, some "arms" can be identified as suboptimal a priori. We show that the algorithm essentially operates on a subset $\mathcal{U}_j \subset \mathcal{U}$ of the arms, whose total number $\tilde{T}_j$ of pre-collected samples contributes to the regret decay. Furthermore, while the standard analysis scales with the sum of the reciprocals of *all* suboptimality gaps, in this case, the quantity $H_j$ sum only over the smaller set $\mathcal{U}_j$. The quantity $C_a$ is a newly introduced complexity measure that captures the difference in difficulty between the two tasks we face: identifying $a$ as suboptimal and finding its worst-case distribution. Drawing parallels with the MAB literature, we compare this bound to that of UE, showing that the contrast is characterized by $C_a$.

(iv) In Section 5, we briefly discuss how the results can be extended to infinite decision sets, assuming the availability of a suitable cover.

For ease of exposition, we removed constants and terms decreasing with $T$ inside the exponential, as well as any quantities outside of it. The formal statements are deferred to the corresponding sections.

## 1.2 RELATED WORK

The predominance of machine learning in society has highlighted the need for robust models that maintain high-quality performance in a multitude of scenarios. Given the inherent uncertainty in identifying the environment, much attention has been given to the problem of learning under distribution shifts Ben-David et al. (2009); Mansour et al. (2009), where training data may not necessarily be sampled from the target distribution. To tackle this, several works Volpi et al. (2018); Zhang et al. (2021); Sutter et al. (2021) have applied the framework of DRO Scarf (1958); Delage & Ye (2010); Ben-Tal et al. (2013) by assuming that the shift occurs within a neighborhood $\mathcal{U}$ of some nominal distribution, typically generating data, and solving (DR). There are many ways to construct $\mathcal{U}$ and optimize the objective, and we refer to Shapiro et al. (2021); Rahimian & Mehrotra (2022) for a thorough review.

A more recent line of work has specialized to finite and unstructured $\mathcal{U} = \{Q_1, \ldots, Q_k\}$, under sampling access to each distribution. Agnostic federated learning Mohri et al. (2019) solves (DR) under mixtures of $\mathcal{U}$, providing high-probability bounds on the generalization gap of non-uniform exploration and an algorithm with empirical optimization guarantees. Collaborative PAC learning Blum et al. (2017) focuses on binary classification, with the aim of guaranteeing $\mathbb{P}\left(\Delta_{\mathrm{DR}}\left(A_T^{\mathrm{o}}\right) \leq \epsilon\right) \geq 1 - \delta$ under a minimal number of samples $T$. The original work of Blum et al. (2017) assumes realizability and subsequent studies Chen et al. (2018); Nguyen & Zakynthinou (2018); Carmon & Hausler (2022); Haghtalab et al. (2022) extended results to the agnostic case and gave improved rates, along with sample-complexity lower bounds. Awasthi et al. (2023) later solidified the theory and posed several open problems, some of which were recently addressed in Peng (2024); Zhang et al. (2024) via optimal algorithms.

In this work, we turn our attention to the simple regret $\mathbb{E}\left[\Delta_{\mathrm{DR}}\left(A_T^{\mathrm{o}}\right)\right]$. For finite decision sets $\mathcal{A}$, an integration of the tails reveals that the regret achieved by Haghtalab et al. (2022) is $O\left(\sqrt{\frac{\log|\mathcal{A}| + k\log k}{T}}\right)$. When $\mathcal{A} \subset \mathbb{R}^d$ has Euclidean diameter at most $B > 0$ and, for each $x \in \mathcal{X}$, the function $r\left(\cdot, x\right)$ is both convex and Lipschitz, several studies have proposed comparable rates using game dynamics. Group DRO Sagawa* et al. (2020) ensures a rate of $O\left(k\sqrt{\frac{B^2 + \log k}{T}}\right)$ and, in the fairness context, Abernethy et al. (2022) obtains $O\left(\frac{B}{\sqrt{T}}\right)$ plus a term that uniformly bounds the generalization gap with high-probability. Subsequently, Soma et al. (2022) was able to attain $O\left(\sqrt{\frac{B^2 + k}{T}}\right)$, showing a matching lower bound, and Zhang et al. (2023) devised strategies with $O\left(\sqrt{\frac{B^2 + k\log k}{T}}\right)$ regret and additionally studied the setting with distribution-specific sampling budget constraints.

Since the learner does not incur any costs when gathering data, MDL closely resembles PE-MAB Bubeck et al. (2011) under the *fixed budget* regime, where distributions represent the arms. It is standard in the MAB literature to distinguish between distribution-dependent and independent rates. The former typically depends on the suboptimality gaps and scales much faster with $T$. In contrast, the latter holds for worst-case environments for each $T$, resulting in slower regret decay. See (Lattimore & Szepesvari, 2020, Ch. 33) for an in-depth discussion. In PE-MAB, Audibert et al. (2010) introduced the UCB-E strategy, which improves performance relative to the gaps when compared to uniform exploration. Motivated by these results, we demonstrate analogous faster distribution-dependent rates in the MDL setting and explore a similar contrast between UE and LCB-DR.

## 2 PRELIMINARIES

**Notation.** We frequently use the notation $[k] := \{1, \ldots, k\}$, where $k \in \mathbb{N}$. For a measurable space $\mathcal{X}$ (we will omit the $\sigma$-algebras), we let $\mathcal{P}(\mathcal{X})$ denote the set of all distributions over it. For two real-valued functions $f$ and $g$, we let $f \lesssim g$ and $f \gtrsim g$ denote inequalities up to universal constants. Given values $a, b \in \mathbb{R}$, we define $a \vee b := \max\{a, b\}$ and $a \wedge b := \min\{a, b\}$.

### 2.1 MULTI-DISTRIBUTION LEARNING

Let $\mathcal{X}$ be the space where our data lives in and $\mathcal{A}$ the space where we make decisions. Given data $X_Q \sim Q \in \mathcal{P}(\mathcal{X})$, statistical learning aims to maximize the stochastic objective $\mu(a; Q) = \mathbb{E}[r(a, X_Q)]$ with respect to $a \in \mathcal{A}$, where $r : \mathcal{A} \times \mathcal{X} \to \mathbb{R}$ is an underlying reward function. In the MDL paradigm, we capture distributional uncertainty by assuming that the distributions come from some uncertainty set $\mathcal{U} \subset \mathcal{P}(\mathcal{X})$ and instead aim to solve the distributionally robust problem (DR), where our goal is to maximize $\mu_{\mathrm{DR}}(a) = \min_{Q \in \mathcal{U}} \mu(a; Q)$. We measure the performance of a decision $a \in \mathcal{A}$ via its *suboptimality gap* $\Delta_{\mathrm{DR}}(a) := \mu^*_{\mathrm{DR}} - \mu_{\mathrm{DR}}(a)$, where $\mu^*_{\mathrm{DR}} := \max_{a \in \mathcal{A}} \mu_{\mathrm{DR}}(a)$ is the optimal objective value. Throughout this work, we operate under the following assumptions.

**Assumption 1** (Finite decision/uncertainty sets). $|\mathcal{A}| = l$ and $|\mathcal{U}| = k$, where $2 \leq l, k < \infty$.

**Assumption 2** (Bounded rewards). The reward function $r$ is bounded in $[0, 1]$ for ease of presentation. This assumption will be relaxed in Section 3.2.

To solve (DR), we interact with the environment for a total of $T \in \mathbb{N}$ rounds. In each round $t \in [T]$, we (i) select a distribution $Q_t \in \mathcal{U}$ and (ii) receive independent data point $X_t \sim Q_t$. After the $T$ rounds, we output a decision $A^o_T \in \mathcal{A}$ with the goal of minimizing the *simple regret* $\mathbb{E}[\Delta_{\mathrm{DR}}(A^o_T)]$ or *error probability* $\mathbb{P}(\Delta_{\mathrm{DR}}(A^o_T) > 0)$. The strategies described in this work are of the form $A^o_T = \mathrm{argmax}_{a \in \mathcal{A}} \mu^o_T(a)$ for an appropriately constructed proxy $\mu^o_T : \mathcal{A} \to \mathbb{R}$.

*Remark* 1 (Simple regret v.s. error probability). Note that both performance measures are closely related: since $r \in [0, 1]$, we have that $\Delta_{\mathrm{DR}} \in [0, 1]$ and, thus,

$$\Delta_{\mathrm{DR,min}} \, \mathbb{P}(\Delta_{\mathrm{DR}}(A^o_T) > 0) \leq \mathbb{E}[\Delta_{\mathrm{DR}}(A^o_T)] \leq \mathbb{P}(\Delta_{\mathrm{DR}}(A^o_T) > 0)$$

where $\Delta_{\mathrm{DR,min}}$ is the minimal positive gap (see Section 2.2).

### 2.2 COMPLEXITY MEASURES

For each decision $a \in \mathcal{A}$, we define its worst performing distribution $Q^*_a := \mathrm{argmin}_{Q \in \mathcal{U}} \mu(a; Q)$ and the suboptimality gaps $\Delta_a(Q) := \mu(a; Q) - \mu_{\mathrm{DR}}(a)$. Much of the analysis that follows is characterized by the minimal positive gaps

$$\Delta_{\mathrm{DR,min}} := \min\{\Delta_{\mathrm{DR}}(a) > 0 : a \in \mathcal{A}\} \quad \text{and} \quad \Delta_{a,\mathrm{min}} := \min\{\Delta_a(Q) > 0 : Q \in \mathcal{U}\}$$

These quantities are additionally used to define complexity measures

$$H_a := \sum_{Q \in \mathcal{U}: \Delta_a(Q) > 0} \Delta_a^{-2}(Q) \quad \text{and} \quad C_a := \begin{cases} \dfrac{\Delta_{\mathrm{DR}}(a)}{\Delta_{a,\mathrm{min}}} & a \neq a^* \\[2mm] \dfrac{\Delta_{\mathrm{DR,min}}}{\Delta_{a^*,\mathrm{min}}} & a = a^* \end{cases}$$

for each $a \in \mathcal{A}$. In pure exploration bandits, $H_a$ is commonly used to characterize the complexity of identifying the optimal arm (e.g., Audibert et al. (2010)), which in our setting translates to identifying $Q^*_a$. The intuition behind $C_a$ is that it compares the difficulty of the two tasks we face: when $C_a \leq 1$ for some $a \neq a^*$, or $\Delta_{\mathrm{DR}}(a) \leq \Delta_{a,\mathrm{min}}$, it is more challenging to rule out $a$ as suboptimal than it is to identify $Q^*_a$.

### 2.3 ALGORITHMIC TOOLS

For each distribution $Q \in \mathcal{U}$, let $X_Q, \{X_Q^{(i)}\}_{i=1}^{\infty} \overset{iid}{\sim} Q$ be a sequence of independent data points. For each $(t, a, Q) \in \mathbb{N} \times \mathcal{A} \times \mathcal{U}$, we define the empirical mean

$$\hat{\mu}_t(a; Q) := \frac{1}{t} \sum_{i=1}^{t} r\left(a, X_Q^{(i)}\right)$$

Under a fixed sampling algorithm, let $n_t(Q) \coloneqq \sum_{s=1}^t \mathbb{I}\{Q_s = Q\}$ denote the number of times that $Q$ is played up to time $t$. The data received is then given by $X_t = X_{Q_t}^{(n_t(Q_t))}$.

## 3 Non-adaptive strategies

We begin by describing two simple non-adaptive strategies. In essence, both sample a fixed number times from each distribution in $\mathcal{U}$ and construct a proxy $\mu_T^o$ that is the natural empirical version of $\mu_{\mathrm{DR}}$. Proofs of the results are deferred to Appendix C.

### 3.1 Uniform exploration (UE)

The most straight-forward strategy is the idea of *uniform exploration (UE)* (Algorithm 1). As the name suggests, we sample the same number $n \in \mathbb{N}$ of times from each distribution, for a total of $T = nk$ samples, and form the empirical proxy

$$\mu_T^o(a) = \min_{Q \in \mathcal{U}} \hat{\mu}_n(a; Q)$$

---

**Algorithm 1** Uniform exploration (UE)

**Input:** Number of samples $n \in \mathbb{N}$
1: Sample $n$ times from each distribution $Q \in \mathcal{U}$
2: Construct $\mu_T^o(a) = \min_{Q \in \mathcal{U}} \hat{\mu}_n(a; Q)$
**Output:** $A_T^o = \mathrm{argmax}_{a \in \mathcal{A}} \mu_T^o(a)$

---

**Theorem 1** (UE regret). *Suppose that $n \geq \left(\frac{8}{\Delta_{\mathrm{DR,min}}}\right)^2 \log k$. Then, the UE algorithm attains the following simple regret bound:*

$$\mathbb{E}[\Delta_{\mathrm{DR}}(A_T^o)] \leq \sum_{a \in \mathcal{A}: \Delta_{\mathrm{DR}}(a) > 0} \Delta_{\mathrm{DR}}(a) \exp\left(-\frac{n}{2}\left[\Delta_{\mathrm{DR}}(a) - 8\sqrt{\frac{\log k}{n}}\right]^2\right)$$

Since our empirical proxy $\mu_T^o$ is not an unbiased estimate of $\mu_{\mathrm{DR}}$, we end up with an approximation error bounded by $\sqrt{\frac{\log k}{n}}$. The lower bound on $n$ is then required to apply tail bounds by ensuring that $\Delta_{\mathrm{DR}}(a) - 8\sqrt{\frac{\log k}{n}} \geq 0$ for all $a \in \mathcal{A}$ (see Appendix C.1).

*Remark* 2 (Small gaps). When $\Delta_{\mathrm{DR,min}}$ is really small, the lower bound condition on $n$ may be difficult to attain. This may be counterintuitive, for example, when all gaps are small, as we expect the problem to be easy. In such situations, an alternative guarantee is

$$\mathbb{E}[\Delta_{\mathrm{DR}}(A_T^o)] \leq \Delta + \sum_{a \in \mathcal{A}: \Delta_{\mathrm{DR}}(a) > \Delta} \Delta_{\mathrm{DR}}(a) \exp\left(-\frac{n}{2}\left[\Delta_{\mathrm{DR}}(a) - 8\sqrt{\frac{\log k}{n}}\right]^2\right)$$

for any $\Delta > 0$, provided that $n \geq \left(\frac{8}{\Delta}\right)^2 \log k$.

With some further manipulation, we can additionally obtain a distribution-independent regret bound.

**Corollary 1** (UE distribution-independent regret). *Suppose that $n \geq \left(\frac{8}{\Delta_{\mathrm{DR,min}}}\right)^2 \log k$. Then, the UE algorithm attains the following distribution-independent simple regret bound:*

$$\mathbb{E}[\Delta_{\mathrm{DR}}(A_T^o)] \lesssim \sqrt{\frac{k \log(kl)}{T}}$$

### 3.2 Non-uniform exploration (NUE)

A natural extension of the UE strategy is to sample a different number of times from each distribution. To address this, *non-uniform exploration (NUE)* (Algorithm 2) samples $n_Q \in \mathbb{N}$ times from

each distribution $Q \in \mathcal{U}$, for a total of $T = \sum_{Q \in \mathcal{U}} n_Q$ samples. Similarly, we define the proxy

$$\mu_T^{\mathrm{o}}(a) = \min_{Q \in \mathcal{U}} \hat{\mu}_{n_Q}(a; Q)$$

Here, we consider real-valued data in $\mathcal{X} \subset \mathbb{R}$ and define mean $\mu_Q := \mathbb{E}[X_Q]$ and variance $\sigma_Q^2 := \mathrm{Var}(X_Q)$ for each $Q \in \mathcal{U}$. We relax the boundedness assumption to rewards $r \in [0, M]$, for some $M > 0$. Additionally, let us sort the sample sizes in increasing order as follows: $0 =: n_{(0)} \le n_{(1)} \le \cdots \le n_{(k)}$ and let $Q_{(j)}$ denote the corresponding distribution in the $j$th position: $n_{Q_{(j)}} = n_{(j)}$. The regret bound presented will rely on the following variance quantities:

$$V_T := \sum_{j=1}^{k} \left( n_{(j)} - n_{(j-1)} \right) \mathbb{E}\left[ \max_{r \in \{j, \dots, k\}} \frac{1}{n_{(r)}^2} \left[ X_{Q_{(r)}} - \mu_{Q_{(r)}} \right]^2 \right]$$

$$\Sigma_T^2 := \mathbb{E}\left[ \max_{Q \in \mathcal{U}} \frac{1}{n_Q^2} \sum_{i=1}^{n_Q} \left( X_Q^{(i)} - \mu_Q \right)^2 \right]$$

$$\sigma_T^2 := \max_{Q \in \mathcal{U}} \frac{\sigma_Q^2}{n_Q}$$

Lastly, we make use of the quantity $G_T := 8M \left( \frac{4 \log k}{\min_{Q \in \mathcal{U}} n_Q} + L\sigma_T \sqrt{2 \log k} \right)$, which we note decreases with the $\{n_Q\}$.

---

**Algorithm 2** Non-uniform exploration (NUE)

---

**Input:** Number of samples $\{n_Q\}_{Q \in \mathcal{U}} \subset \mathbb{N}$ allocated to each distribution
  1: Sample $n_Q$ times from each distribution $Q \in \mathcal{U}$
  2: Construct $\mu_T^{\mathrm{o}}(a) = \min_{Q \in \mathcal{U}} \hat{\mu}_{n_Q}(a; Q)$
**Output:** $A_T^o = \mathrm{argmax}_{a \in \mathcal{A}} \mu_T^{\mathrm{o}}(a)$

---

**Theorem 2** (NUE regret). *Suppose that $r(a, \cdot) : \mathcal{X} \to [0, M]$ is $L$-Lipschitz for each $a \in \mathcal{A}$, and that $\Delta_{\mathrm{DR,min}} \ge G_T$. Then, the NUE algorithm attains the following simple regret bound:*

$$\mathbb{E}[\Delta_{\mathrm{DR}}(A_T^o)]$$

$$\le \sum_{a \in \mathcal{A}: \Delta_{\mathrm{DR}}(a) > 0} \Delta_{\mathrm{DR}}(a) \exp\left( -\frac{[\Delta_{\mathrm{DR}}(a) - G_T]^2}{16L^2 \left( 2\sigma_T^2 + \Sigma_T^2 + 6V_T \right) + \frac{2\sqrt{6}M}{\min_{Q \in \mathcal{U}} n_Q} [\Delta_{\mathrm{DR}}(a) - G_T]} \right)$$

As intuition suggests, the definitions imply that sampling more from distributions with higher variance yields better rates. On the other hand, due to the presence of $\min_{Q \in \mathcal{U}} n_Q$ in the bound, it may also be favorable to balance this principle with ensuring that no distribution is significantly undersampled.

### 3.3 UNIFORM V.S. NON-UNIFORM EXPLORATION

Consider bounded rewards $r \in [0, M]$, where $M > 0$. We can more generally express the probability of selecting a suboptimal arm $a \in \mathcal{A}$ for UE and NUE as follows (see Appendix C):

$$\underbrace{\exp\left( -\frac{n}{M^2} [\Delta_{\mathrm{DR}}(a) - B_n]^2 \right)}_{\text{UE}} \quad \text{v.s.} \quad \underbrace{\exp\left( -\frac{[\Delta_{\mathrm{DR}}(a) - B_n]^2}{\sigma_T^2 + \Sigma_T^2 + V_T + \frac{M}{\min_Q n_Q} [\Delta_{\mathrm{DR}}(a) - B_n]} \right)}_{\text{NUE}}$$

where we have omitted constants. Here, $B_n$ is a quantity that decreases with the sample size. To mirror the standard Hoeffding v.s. Bernstein discussion, consider a small-sample regime where $\Delta_{\mathrm{DR}}(a) \approx B_n$. The comparison then reduces to $\frac{M^2}{n}$ (for UE) v.s. $\sigma_T^2 + \Sigma_T^2 + V_T$ (for NUE), where the smaller term is better. Note that $M$ captures the range of the reward function $r$, while $\sigma_T^2, \Sigma_T^2$ and $V_T$ capture the variance of the distributions in $\mathcal{U}$. This shows that NUE can be better for two reasons:

(i) If the reward can take very large values but the data concentrates in a small region, then the variances can be much smaller compared to $M^2$.

(ii) If the learner allocates more samples to distributions with higher variance, then the decay can be much faster.

### 3.4 BOUNDS ON VARIANCE QUANTITIES

While the variance quantities introduced seem hard to control and lack interpretability, here we highlight some strategies and examples to mitigate this issue. Proofs of all results shown here are deferred to Appendix G.

#### 3.4.1 CRUDE BOUND

Note the variance hierarchy $\sigma_T^2 \leq \Sigma_T^2 \leq V_T$. To unify them, we can bound the max with a sum to get $V_T \leq \sum_{Q \in \mathcal{U}} \frac{\sigma_Q^2}{n_Q}$, which we can then substitute all three terms with. However, this results in a linear dependence on $k$ that we aim to avoid.

#### 3.4.2 BOUNDING $\Sigma_T^2$

Suppose that our data is bounded: $X_Q \in [0, 1]$ for each $Q \in \mathcal{U}$. Then we can establish the following upper bound:

$$\Sigma_T^2 \lesssim \sqrt{\frac{\log k}{\min_{Q \in \mathcal{U}} n_Q^3}} + \sigma_T^2$$

Since the first term on the right-hand side decays faster than $O\left(\frac{1}{\min_{Q \in \mathcal{U}} n_Q}\right)$, we can focus our attention on $\sigma_T^2$, which is a more interpretable quantity.

#### 3.4.3 BOUNDING $V_T$

The most formidable quantity is $V_T$, but we can readily relate it to $\Sigma_T^2$:

$$V_T \leq \min\left\{\max_{Q \in \mathcal{U}} n_Q, k\right\} \Sigma_T^2$$

In a setting where $k$ is not too large, this result shows that control over $\Sigma_T^2$, also ensures control over $V_T$.

For a more concrete example, suppose that $\mathcal{U} = \{Q_1, \ldots, Q_k\}$, where $Q_1, \ldots, Q_{k-1}$ share a common small variance $\sigma^2$ and $Q_k$ has a much larger variance $\nu^2 \gg \sigma^2$. In addition, suppose that $Q_1, \ldots, Q_{k-1}$ are supported in $[0, 1]$. Consider the NUE procedure with $n$ samples from each $Q_1, \ldots, Q_{k-1}$ and $m = T - n(k-1) \geq n$ samples (where $T \geq nk$ is the total number of samples) from $Q_k$. Intuitively, we would like for $m \gg n$ since $Q_k$ is harder to learn (i.e., has more variability). This can be reflected in the strong variance:

$$V_T \lesssim \frac{\sqrt{\log k} + \sigma^2}{n} + \frac{\nu^2}{T - nk}$$

The comparison with UE then becomes (we ignore $\sigma_T^2$ and $\Sigma_T^2$ since $V_T$ is the dominating term)

$$\underbrace{\frac{M^2 k}{T}}_{\text{UE}} \quad \text{v.s.} \quad \underbrace{\frac{\sqrt{\log k} + \sigma^2}{n} + \frac{\nu^2}{T - nk}}_{\text{NUE}}$$

again with the smaller term being better. This shows that the NUE decay can be much smaller when $\nu^2, M^2, k$ and $T$ are large relative to $\sigma^2$ and $n$. For example, consider $\sigma^2 = n = 1$, $\nu^2 = M^2 = k = C > 1$ and $T = C + 49$; that is, 1 sample is allocated to $Q_1, \ldots, Q_{k-1}$ and 50 samples to $Q_k$. Then, up to constants, the comparison becomes $C^2$ (for UE) v.s. $\sqrt{\log C} + C$ (for NUE). As $C$ grows, the UE bound becomes arbitrarily larger.

# 4 OPTIMISM

As opposed to the non-adaptive strategies covered thus far, the next algorithm we present makes sampling decisions as it interacts with the environment. For this analysis, we additionally operate under the following uniqueness assumption.

**Assumption 3** (Unique optima). $a^*$ and $Q_a^*$ are the *unique* optimal decision and the *unique* worst-case distribution for $a \in \mathcal{A}$, respectively.

As is standard in UCB-style algorithms, for some choice of parameter $\epsilon > 0$, we define *index*

$$\text{LCB}_t(Q; a, \epsilon) := \hat{\mu}_{n_t(Q)}(a; Q) - \sqrt{\frac{\epsilon}{n_t(Q)}} \quad \forall (t, a, Q) \in \mathbb{N} \times \mathcal{A} \times \mathcal{U}$$

which represents a *lower confidence bound (LCB)* on the true mean $\mu(a; Q)$. At a high-level, the *LCB-DR* strategy (Algorithm 3) iterates through each decision $a \in \mathcal{A}$ and performs a modified version of UCB-E Audibert et al. (2010) to identify $Q_a^*$. The modification takes advantage of the fact that data sampled in a previous round can be reused for the current one. In essence, we analyze UCB-E when each distribution starts the game with a certain number of pulls. Intuitively, if some distribution has already been played sufficiently many times, it will not be played again in this round, yielding an improved sample complexity.

For completeness, we initiate the procedure by sampling from each distribution once; that is, $n_k(Q) := 1$ for each $Q \in \mathcal{U}$. As a result, we define $T_0 := \bar{T}_0 := k$ to be the total number of samples gathered before the game starts. The inputs to the algorithm are a permutation $(a_1, \ldots, a_l)$ of $\mathcal{A}$, dictating the order in which decisions are iterated through, and index parameters $(\epsilon_1, \ldots, \epsilon_l)$ satisfying

$$\epsilon_j \geq \frac{25}{36} \Delta_{a_j, \min}^2 (u_{j-1} - 1) \tag{1}$$

where

$$u_0 := k \quad \text{and} \quad u_j := k(j+1) + \frac{72}{25} \sum_{r=1}^{j} \epsilon_r H_{a_r}$$

The procedure then works as follows: at each round $j \in [l]$,

1. Since we reuse samples from previous rounds, some distributions may already have enough samples by the start of the current round and, thus, may not be sampled from at all. We define the following set as a proxy for the arms that will be played in this round:

$$\mathcal{U}_j := \left\{ Q \in \mathcal{U} \setminus \left\{ Q_{a_j}^* \right\} : n_{\bar{T}_{j-1}}(Q) < \frac{36}{25} \epsilon_j \Delta_{a_j}^{-2}(Q) \right\} \cup \left\{ Q_{a_j}^* : n_{\bar{T}_{j-1}}\left( Q_{a_j}^* \right) < \frac{36}{25} \epsilon_j \Delta_{a_j, \min}^{-2} \right\}$$

Additionally, define

$$k_j := |\mathcal{U}_j| \, \mathbb{I}\left\{ Q_{a_j}^* \in \mathcal{U}_j \right\}, \quad \tilde{T}_j := \sum_{Q \in \mathcal{U}_j} n_{\bar{T}_{j-1}}(Q)$$

$$H_j := \Delta_{a_j, \min}^{-2} \mathbb{I}\left\{ Q_{a_j}^* \in \mathcal{U}_j \right\} + \sum_{Q \in \mathcal{U}_j \setminus \left\{ Q_{a_j}^* \right\}} \Delta_{a_j}^{-2}(Q)$$

2. Allocate

$$T_j := \frac{36}{25} \epsilon_j H_j - \tilde{T}_j + k_j$$

samples to this round and let $\bar{T}_j := \sum_{r=0}^{j} T_r$ denote the total number of samples obtained up to and including round $j \in [l]$.

3. For each $t = \bar{T}_{j-1} + 1, \ldots, \bar{T}_j$, sample

$$X_t \sim Q_t := \underset{Q \in \mathcal{U}}{\operatorname{argmin}} \, \text{LCB}_{t-1}(Q; a_j, \epsilon_j)$$

In essence, we play the modified UCB-E for $T_j$ rounds on expected rewards $\{\mu(a_j; Q)\}_{Q \in \mathcal{U}}$.

4. Define

$$\hat{Q}_j := \operatorname*{argmin}_{Q \in \mathcal{U}} \hat{\mu}_{n_{\bar{T}_j}(Q)}(a_j; Q) \quad \text{and} \quad \mu_T^o(a_j) := \hat{\mu}_{n_{\bar{T}_j}(\hat{Q}_j)}(a_j; \hat{Q}_j)$$

Intuitively, $\hat{Q}_j$ and $\mu_T^o$ are proxies for $Q_{a_j}^*$ and $\mu_{\mathrm{DR}}$, respectively.

Finally, after gathering $T := \sum_{j=0}^l T_j$ total samples, we maximize the proxy objective: $A_T^o := \operatorname{argmax}_{a \in \mathcal{A}} \mu_T^o(a)$. By analyzing the optimiality of the modified UCB-E algorithm (see Appendix D), we can then reach the following conclusion.

---

**Algorithm 3** LCB-DR

---

**Input:** Initial number of samples $T_0 = \bar{T}_0 = k$, permutation $(a_1, \ldots, a_l)$ of $\mathcal{A}$ and index parameters $(\epsilon_1, \ldots, \epsilon_l)$.
1: **for** $j = 1, \ldots, l$ **do**
2:     Define proxy set $\mathcal{U}_j$ and quantities $k_j$, $\tilde{T}_j$ and $H_j$.
3:     Allocate $T_j$ samples to this round.
4:     **for** $t = \bar{T}_{j-1} + 1, \ldots, \bar{T}_j$ **do**
5:         Sample data point $X_t \sim Q_t$.
6:     **end for**
7:     Define proxies $\hat{Q}_j$ and $\mu_T^o(a_j)$.
8: **end for**
**Output:** $A_T^o = \operatorname{argmax}_{a \in \mathcal{A}} \mu_T^o(a)$

---

**Theorem 3** (LCB-DR error probability). *Under Assumption 3 and the parameter lower bound* (1), *the LCB-DR algorithm attains the following error probability:*

$$\mathbb{P}(A_T^o \neq a^*) \leq 2k \sum_{j=1}^l u_j \exp\left(-\frac{2\left(C_{a_j}^2 \wedge 1\right)\epsilon_j}{25}\right)$$

Note that $\epsilon_j = \frac{25}{36}\frac{T_j + \tilde{T}_j - k_j}{H_j}$, so that the decay scales with $O\left(\frac{\left(C_{a_j}^2 \wedge 1\right)\left(T_j + \tilde{T}_j - k_j\right)}{H_j}\right)$. Intuitively, at each round $j \in [l]$, the sample complexity depends on the difficulty of identifying the worst-case distribution $Q_{a_j}^*$, which, as in PE-MAB, is controlled by the suboptimality gaps $\left\{\Delta_{a_j}(Q)\right\}_{Q \in \mathcal{U}}$.

*Remark* 3 (Improvement over UCB-E). We highlight the importance of using samples obtained in previous rounds: as opposed to the standard UCB-E analysis, we have the additional $\tilde{T}_j$ contribution, we only offset by $k_j \leq k$, and the complexity measure $H_j$ improves upon $H_{a_j}$ by only summing over a subset of $\mathcal{U}$.

*Remark* 4 (Unknown quantities). Note that the choice of $T_j$ requires knowledge of unknown quantities, such as $\mathcal{U}_j$ and $Q_{a_j}^*$. However, as shown in the statement of Theorem D.1, optimality is ensured provided that $T_j \geq \frac{36}{25}\epsilon_j H_j - \tilde{T}_j + k_j$, but the concentration bound in Appendix E.1 requires additional manipulation when substituting $\epsilon_j$ into Hoeffding's.

In addition, we emphasize that $\epsilon_j$ and the decisions $Q_t$ *do not* require knowledge of $\mathcal{U}_j$: the former relies on a lower bound and the latter optimizes over all of $\mathcal{U}$.

### 4.1 COMPARISON WITH UE

Focusing on the dominating terms, the probability of selecting a suboptimal arm $a_j \in \mathcal{A}$, that is in the $j$th permutation position for LCB-DR, is approximately $\approx \exp\left(-\frac{T\Delta_{\mathrm{DR}}^2(a_j)}{k}\right)$ for UE and $\approx \exp\left(-\frac{\left(C_{a_j}^2 \wedge 1\right)\left(T_j + \tilde{T}_j\right)}{H_j}\right)$ for LCB-DR. Extracting the quantity inside the exponential, we break it down into two cases:

- $\Delta_{\mathrm{DR}}\left(a_j\right) \le \Delta_{a_j,\min}$ (or $C_{a_j} \le 1$): intuitively, this means that it is more difficult to rule out $a_j$ as suboptimal than to identify $Q^*_{a_j}$. Then, the comparison reduces to $\frac{T}{k\Delta_{a_j,\min}^{-2}}$ (for UE) v.s. $\frac{T_j + \tilde{T}_j}{H_j}$ (for LCB-DR).

- $\Delta_{a_j,\min} \le \Delta_{\mathrm{DR}}\left(a_j\right)$ (or $C_{a_j} \ge 1$): intuitively, this means that it is more difficult to identify $Q^*_{a_j}$ than to rule out $a_j$ as suboptimal. Then, the comparison is between $\frac{T}{k\Delta_{\mathrm{DR}}^{-2}(a_j)}$ (for UE) v.s. $\frac{T_j + \tilde{T}_j}{H_j}$ (for LCB-DR).

Putting these together results in

$$\underbrace{\frac{T}{k \min\left\{\Delta_{a_j,\min}^{-2}, \Delta_{\mathrm{DR}}^{-2}\left(a_j\right)\right\}}}_{\text{UE}} \quad \text{v.s.} \quad \underbrace{\frac{T_j + \tilde{T}_j}{H_j}}_{\text{LCB-DR}}$$

where the larger term yields the better rate. When sample sizes are large relative to $l$, so that $T \approx T_j + \tilde{T}_j$, optimism is favorable when $H_j \le k \min\left\{\Delta_{a_j,\min}^{-2}, \Delta_{\mathrm{DR}}^{-2}\left(a_j\right)\right\}$. As in MAB, this is always the case when $\Delta_{a_j,\min}^{-2}$ is the smaller term; otherwise, it depends on the problem instance. Note that $H_j$ can be much smaller when $|\mathcal{U}_j| \ll k$.

## 5 EXTENDING TO INFINITE DECISION SETS

While the results discussed thus far only apply to finite decision sets $\mathcal{A}$, it is possible to extend to larger (possibly infinite) sets via standard covering arguments. Let $\bar{Q} := \frac{1}{k}\sum_{Q \in \mathcal{U}} Q$ be the uniform mixture and suppose that we have access to a finite $\frac{\epsilon}{k}$-cover $\mathcal{A}_\epsilon$ of $\left(\{r\left(a, \cdot\right)\}_{a \in \mathcal{A}}, L^2\left(\bar{Q}\right)\right)$ in the following sense: for all $a \in \mathcal{A}$, there exists a $\phi_a \in \mathcal{A}_\epsilon$ such that

$$\|r\left(a, \cdot\right) - r\left(\phi_a, \cdot\right)\|_{L^2\left(\bar{Q}\right)} := \sqrt{\mathbb{E}_{X \sim \bar{Q}}\left[\left(r\left(a, X\right) - r\left(\phi_a, X\right)\right)^2\right]} \le \frac{\epsilon}{k}$$

The idea is that the regret under the finite set $\mathcal{A}_\epsilon$ is close to the regret under $\mathcal{A}$, so that a learner can play the game dynamics on the former.

**Lemma 1** (Controlling regret using a cover). *Let* $\Delta_{\mathrm{DR}}\left(a; \mathcal{A}_\epsilon\right) := \max_{a^* \in \mathcal{A}_\epsilon} \mu_{\mathrm{DR}}\left(a^*\right) - \mu_{\mathrm{DR}}\left(a\right)$ *denote the suboptimality gap with respect to* $\mathcal{A}_\epsilon$. *Then,* $\Delta_{\mathrm{DR}}\left(a\right) \le \Delta_{\mathrm{DR}}\left(a; \mathcal{A}_\epsilon\right) + \epsilon$ *for all* $a \in \mathcal{A}$.

This result admits a straightforward proof, which we defer to Appendix F. In addition, in Appendix F.1, we specialize this result to the binary classification setting and present a distribution-independent bound for classes of finite VC dimension.

## 6 DISCUSSION

In this work, we delve into the problem of DRO within the MDL framework, an area of growing popularity in high-stakes machine learning applications. Rooted in empirical process theory and inspired by the PE-MAB literature, we offer novel insight into the key strategies of uniform and non-uniform exploration via distribution-dependent bounds. By scaling with instance-specific quantities, our proposed bounds decay much faster, with respect to sample sizes, than existing ones. We additionally devise an optimistic method, LCB-DR, that shows improvements over its non-adaptive counterparts, paralleling classical findings in the MAB setting.

While LCB-DR exhibits favorable rates, we reiterate that tuning certain parameters involves estimating unknown quantities. This raises the question of whether there exists a more astute way to select such quantities with minimal prior information. Moreover, the procedure requires specifying the order to play the actions in. Although the absence of any problem knowledge might preclude exploiting this sequence effectively, perhaps some preliminary understanding of the distributions allows potential advantages (e.g., start with actions that explore as much as possible, so that $\mathcal{U}_j$ is small in future iterations).

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

# A  EXPECTATION OF EMPIRICAL PROCESS MAXIMUM

Let $\mathcal{U} \subset \mathcal{P}(\mathcal{X})$ be a finite set of distributions over a data space $\mathcal{X}$, with $2 \leq k \coloneqq |\mathcal{U}| < \infty$. For each distribution $Q \in \mathcal{U}$, we have an associated sample size $n_Q \in \mathbb{N}$ and define $T \coloneqq \sum_{Q \in \mathcal{U}} n_Q$. When $\mathcal{X} \subset \mathbb{R}$, we additionally denote the variance of each distribution by $\sigma_Q^2 \coloneqq \mathrm{Var}(Q)$ and define $\sigma_T \coloneqq \max_{Q \in \mathcal{U}} \frac{\sigma_Q}{\sqrt{n_Q}}$.

In the development that follows, we will work with independent $\mathcal{X}$-valued random variables $(X_Q)_{Q \in \mathcal{U}}, X \coloneqq \left(X_Q^{(i)}\right)_{Q \in \mathcal{U}, i \in [n_Q]}$, where $X_Q, \left(X_Q^{(i)}\right)_{i \in [n_Q]} \overset{iid}{\sim} Q$ for each $Q \in \mathcal{U}$. For a collection of functions $\{f_Q : \mathcal{X} \to [-1, 1]\}_{Q \in \mathcal{U}}$, such that each $f_Q(X_Q)$ is centered, our primary goal will be to bound the following quantity:

$$\mathbb{E}\left[\max_{Q \in \mathcal{U}} \left| \frac{1}{n_Q} \sum_{i=1}^{n_Q} f_Q\left(X_Q^{(i)}\right) \right|\right]$$

In particular, we will show the following bounds.

**Theorem A.1.** *Let $\{f_Q : \mathcal{X} \to [-1, 1]\}_{Q \in \mathcal{U}}$ be a collection of functions such that $\mathbb{E}[f_Q(X_Q)] = 0$ for each $Q \in \mathcal{U}$. Then,*

$$\mathbb{E}\left[\max_{Q \in \mathcal{U}} \left| \frac{1}{n_Q} \sum_{i=1}^{n_Q} f_Q\left(X_Q^{(i)}\right) \right|\right] \leq 4\sqrt{\frac{\log k}{\min_{Q \in \mathcal{U}} n_Q}}$$

*Moreover, if $\mathcal{X} \subset \mathbb{R}$ and each function $f_Q$ is $L$-Lipschitz, then*

$$\mathbb{E}\left[\max_{Q \in \mathcal{U}} \left| \frac{1}{n_Q} \sum_{i=1}^{n_Q} f_Q\left(X_Q^{(i)}\right) \right|\right] \leq \frac{16 \log k}{\min_{Q \in \mathcal{U}} n_Q} + 4L\sigma_T\sqrt{2 \log k}$$

We note that the first bound can be directly obtained by a high-probability bound via Hoeffding's inequality, along with a union bound, and a subsequent integration of the tails. The second bound (Theorem A.3) requires a more careful analysis and, in the process of deriving it, we additionally show the first result (Corollary A.2).

The proof will follow in two parts: first, in Section A.1, we use symmetrization to bound the quantity of interest with a notion of Rademacher complexity, and subsequently derive bounds on this complexity in Section A.2.

## A.1 SYMMETRIZATION

A standard approach to bound empirical process maxima is via symmetrization. We begin by defining the Rademacher complexity variant of a class of functions $\{h_Q : \mathcal{X} \to \mathbb{R}\}_{Q \in \mathcal{U}}$:

$$\mathfrak{R}_T \left( \{h_Q\}_{Q \in \mathcal{U}} \right) := \mathbb{E} \left[ \max_{Q \in \mathcal{U}} \left| \frac{1}{n_Q} \sum_{i=1}^{n_Q} \epsilon_i h_Q \left( X_Q^{(i)} \right) \right| \right]$$

where $\epsilon_1, \ldots, \epsilon_{\max_{Q \in \mathcal{U}} n_Q} \overset{iid}{\sim}$ Rad (i.e., they are each uniform on $\{-1, 1\}$) are independent from $X$. Note that we place no assumptions on $h_Q (X_Q)$ being centered. We begin by stating an auxiliary lemma.

**Lemma A.1.** *For random variable $Z \in \mathcal{Z}$ and function class $\mathcal{F} \subset \mathbb{R}^{\mathcal{Z}}$, we have that*

$$\sup_{f \in \mathcal{F}} |\mathbb{E} [f (Z)]| \leq \mathbb{E} \left[ \sup_{f \in \mathcal{F}} |f (Z)| \right]$$

*Proof.* For any $f \in \mathcal{F}$,

$$|\mathbb{E} [f (Z)]| \leq \mathbb{E} [|f (Z)|] \leq \mathbb{E} \left[ \sup_{g \in \mathcal{F}} |g (Z)| \right]$$

The claim then follows by taking the supremum over $f \in \mathcal{F}$ on the left-hand side. ∎

The proof of the following result is virtually the same as that of (Wainwright, 2019, Theorem 4.10), with minor modifications, and we present it here for completeness.

**Theorem A.2** (Symmetrization). *For any collection of functions $\{h_Q : \mathcal{X} \to \mathbb{R}\}_{Q \in \mathcal{U}}$, we have that*

$$\mathbb{E} \left[ \max_{Q \in \mathcal{U}} \left| \frac{1}{n_Q} \sum_{i=1}^{n_Q} \left\{ h_Q \left( X_Q^{(i)} \right) - \mathbb{E} [h_Q (X_Q)] \right\} \right| \right] \leq 2 \mathfrak{R}_T \left( \{h_Q\}_{Q \in \mathcal{U}} \right)$$

*Proof.* Let $Y := \left( Y_Q^{(i)} \right)_{Q \in \mathcal{U}, i \in [n_Q]}$ be an independent copy of $X$ and let $P$ denote their common distribution. Then,

$$\mathbb{E} \left[ \max_{Q \in \mathcal{U}} \left| \frac{1}{n_Q} \sum_{i=1}^{n_Q} \left\{ h_Q \left( X_Q^{(i)} \right) - \mathbb{E} [h_Q (X_Q)] \right\} \right| \right] = \mathbb{E} \left[ \max_{Q \in \mathcal{U}} \left| \frac{1}{n_Q} \sum_{i=1}^{n_Q} \left\{ h_Q \left( X_Q^{(i)} \right) - \mathbb{E} \left[ h_Q \left( Y_Q^{(i)} \right) \right] \right\} \right| \right]$$

$$= \int_{\mathcal{X}^T} \max_{Q \in \mathcal{U}} \left| \mathbb{E} \left[ \frac{1}{n_Q} \sum_{i=1}^{n_Q} \left[ h_Q \left( x_Q^i \right) - h_Q \left( Y_Q^{(i)} \right) \right] \right] \right| dP (\mathbf{x})$$

$$=: (*_1)$$

Here, we view $\mathbf{x} := \left( x_Q^i \right)_{Q \in \mathcal{U}, i \in [n_Q]} \in \mathcal{X}^T$ as a $T$-dimensional vector. For each such vector, define function class

$$\mathcal{F}_{\mathbf{x}} := \left\{ \mathbf{y} \mapsto \frac{1}{n_Q} \sum_{i=1}^{n_Q} \left[ h_Q \left( x_Q^i \right) - h_Q \left( y_Q^i \right) \right] : Q \in \mathcal{U} \right\} \subset \{\mathcal{X}^T \to \mathbb{R}\}$$

We can then apply Lemma A.1 to obtain

$$(*_1) = \int_{\mathcal{X}^T} \max_{f \in \mathcal{F}_{\mathbf{x}}} |\mathbb{E} [f (Y)]| \, dP (\mathbf{x})$$

$$\leq \int_{\mathcal{X}^T} \mathbb{E} \left[ \max_{f \in \mathcal{F}_{\mathbf{x}}} |f (Y)| \right] dP (\mathbf{x}) \qquad \text{Lem. A.1}$$

$$= \int_{\mathcal{X}^T} \mathbb{E} \left[ \max_{Q \in \mathcal{U}} \left| \frac{1}{n_Q} \sum_{i=1}^{n_Q} \left[ h_Q \left( x_Q^i \right) - h_Q \left( Y_Q^{(i)} \right) \right] \right| \right] dP (\mathbf{x})$$

$$= \mathbb{E} \left[ \max_{Q \in \mathcal{U}} \left| \frac{1}{n_Q} \sum_{i=1}^{n_Q} \left[ h_Q \left( X_Q^{(i)} \right) - h_Q \left( Y_Q^{(i)} \right) \right] \right| \right]$$

$$=: (*_2)$$

Next, define $n := \max_{Q \in \mathcal{U}} n_Q$ and let $\tilde{\epsilon}_1, \ldots, \tilde{\epsilon}_n \in \{-1, 1\}$ be fixed quantities. From symmetry and independence, we have that

$$\left( h_Q \left( X_Q^{(i)} \right) - h_Q \left( Y_Q^{(i)} \right) \right)_{Q \in \mathcal{U}, i \in [n_Q]} \overset{d}{=} \left( \tilde{\epsilon}_i \left[ h_Q \left( X_Q^{(i)} \right) - h_Q \left( Y_Q^{(i)} \right) \right] \right)_{Q \in \mathcal{U}, i \in [n_Q]}$$

Hence, if we define Rademacher variables $\epsilon^n \overset{iid}{\sim} \text{Rad}$ that are independent from $X$ and $Y$, we can conclude that

$$(*_2) = \frac{1}{2^n} \sum_{\tilde{\epsilon}^n \in \{-1,1\}^n} \mathbb{E} \left[ \max_{Q \in \mathcal{U}} \left| \frac{1}{n_Q} \sum_{i=1}^{n_Q} \tilde{\epsilon}_i \left[ h_Q \left( X_Q^{(i)} \right) - h_Q \left( Y_Q^{(i)} \right) \right] \right| \right]$$

$$= \mathbb{E} \left[ \max_{Q \in \mathcal{U}} \left| \frac{1}{n_Q} \sum_{i=1}^{n_Q} \epsilon_i \left[ h_Q \left( X_Q^{(i)} \right) - h_Q \left( Y_Q^{(i)} \right) \right] \right| \right]$$

$$\leq \mathbb{E} \left[ \max_{Q \in \mathcal{U}} \left\{ \left| \frac{1}{n_Q} \sum_{i=1}^{n_Q} \epsilon_i h_Q \left( X_Q^{(i)} \right) \right| + \left| \frac{1}{n_Q} \sum_{i=1}^{n_Q} \epsilon_i h_Q \left( Y_Q^{(i)} \right) \right| \right\} \right]$$

$$\leq \mathbb{E} \left[ \max_{Q \in \mathcal{U}} \left\{ \left| \frac{1}{n_Q} \sum_{i=1}^{n_Q} \epsilon_i h_Q \left( X_Q^{(i)} \right) \right| \right\} + \max_{Q \in \mathcal{U}} \left\{ \left| \frac{1}{n_Q} \sum_{i=1}^{n_Q} \epsilon_i h_Q \left( Y_Q^{(i)} \right) \right| \right\} \right]$$

$$= 2 \mathbb{E} \left[ \max_{Q \in \mathcal{U}} \left| \frac{1}{n_Q} \sum_{i=1}^{n_Q} \epsilon_i h_Q \left( X_Q^{(i)} \right) \right| \right]$$

$$= 2 \, \mathfrak{R}_T \left( \{h_Q\}_{Q \in \mathcal{U}} \right)$$

∎

## A.2 Bounds on the Rademacher complexity

For the symmetrization trick to be useful, we need to bound $\mathfrak{R}_T \left( \{h_Q\}_{Q \in \mathcal{U}} \right)$. To this end, we begin by defining the Rademacher complexity of a set $\Theta \subset \mathbb{R}^n$:

$$\hat{\mathfrak{R}} (\Theta) := \mathbb{E} \left[ \sup_{\theta \in \Theta} |\langle \epsilon^n, \theta \rangle| \right]$$

where $\epsilon^n = (\epsilon_1, \ldots, \epsilon_n) \overset{iid}{\sim} \text{Rad}$. The process $\{\langle \epsilon^n, \theta \rangle\}_{\theta \in \Theta}$ is sub-Gaussian and, for finite $\Theta$, the Rademacher complexity admits a particularly simple bound, shown next. For a deeper dive into the field, see, e.g., (Wainwright, 2019, Chapter 5).

**Lemma A.2** (Bounding the Rademacher complexity of a finite set). *Let $\Theta \subset \mathbb{R}^n$ satisfy $2 \leq |\Theta| < \infty$. Then,*

$$\hat{\mathfrak{R}} (\Theta) \leq 2 D_\Theta \sqrt{\log |\Theta|}$$

*where $D_\Theta := \max_{\theta \in \Theta} \|\theta\|_2$.*

*Proof.* Note that since each $\epsilon_i$ is 1-sub-Gaussian,

$$\mathbb{E} \left[ e^{\lambda \langle \epsilon^n, \theta \rangle} \right] = \prod_{i=1}^{n} \mathbb{E} \left[ e^{\lambda \epsilon_i \theta_i} \right] \leq \prod_{i=1}^{n} e^{\frac{\lambda^2 \theta_i^2}{2}} = e^{\frac{\lambda^2 \|\theta\|_2^2}{2}} \leq e^{\frac{\lambda^2 D_\Theta^2}{2}}$$

for any $\theta \in \Theta$ and $\lambda \in \mathbb{R}$. That is, $\langle \epsilon^n, \theta \rangle$ is a centered $D_\Theta$-sub-Gaussian variable and we can, thus, apply the standard maximal inequality (e.g., (Boucheron et al., 2013 - 2013, Theorem 2.5)) to obtain the claim. ∎

We can relate both notions of Rademacher complexity introduced thus far to conclude the following result.

**Corollary A.1.** *For a collection of functions* $\{h_Q : \mathcal{X} \to \mathbb{R}\}_{Q \in \mathcal{U}}$, *define the random variable*

$$D\left(\{h_Q\}_{Q \in \mathcal{U}}\right) := \max_{Q \in \mathcal{U}} \sqrt{\sum_{i=1}^{n_Q} \left(\frac{h_Q\left(X_Q^{(i)}\right)}{n_Q}\right)^2}$$

*Then, we have that*

$$\mathfrak{R}_T\left(\{h_Q\}_{Q \in \mathcal{U}}\right) \leq 2\sqrt{\log k}\,\mathbb{E}\left[D\left(\{h_Q\}_{Q \in \mathcal{U}}\right)\right]$$

*Proof.* Fix $\mathbf{x} := \left(x_Q^i\right)_{Q \in \mathcal{U}, i \in [n_Q]} \in \mathcal{X}^T$. Let $n := \max_{Q \in \mathcal{U}} n_Q$ and define vectors $\theta_Q^{\mathbf{x}} \in \mathbb{R}^n$ by

$$\left[\theta_Q^{\mathbf{x}}\right]_i := \begin{cases} \frac{h_Q\left(x_Q^i\right)}{n_Q} & i \leq n_Q \\ 0 & \text{otherwise} \end{cases} \quad \forall i \in [n], Q \in \mathcal{U}$$

and define the set of all such vectors $\Theta^{\mathbf{x}} := \left\{\theta_Q^{\mathbf{x}} : Q \in \mathcal{U}\right\}$, so that $|\Theta^{\mathbf{x}}| = k \geq 2$. Then, note that

$$\hat{\mathfrak{R}}\left(\Theta^{\mathbf{x}}\right) = \mathbb{E}\left[\max_{\theta \in \Theta^{\mathbf{x}}} |\langle \epsilon^n, \theta \rangle|\right] = \mathbb{E}\left[\max_{Q \in \mathcal{U}} \left|\frac{1}{n_Q}\sum_{i=1}^{n_Q} \epsilon_i h_Q\left(x_Q^i\right)\right|\right]$$

Moreover, since $D_{\Theta^{\mathbf{x}}} = \max_{Q \in \mathcal{U}} \sqrt{\sum_{i=1}^{n_Q}\left(\frac{h_Q\left(x_Q^i\right)}{n_Q}\right)^2}$, Lemma A.2 yields

$$\mathfrak{R}_T\left(\{h_Q\}_{Q \in \mathcal{U}}\right) = \mathbb{E}\left[\hat{\mathfrak{R}}\left(\Theta^X\right)\right] \leq \mathbb{E}\left[2D_{\Theta^{\mathbf{x}}}\sqrt{\log|\Theta^X|}\right] = 2\sqrt{\log k}\,\mathbb{E}\left[D\left(\{h_Q\}_{Q \in \mathcal{U}}\right)\right]$$

∎

We can then readily obtain the first bound of interest.

**Corollary A.2.** *Let* $\{f_Q : \mathcal{X} \to [-1, 1]\}_{Q \in \mathcal{U}}$ *be a collection of functions such that* $\mathbb{E}\left[f_Q\left(X_Q\right)\right] = 0$ *for each* $Q \in \mathcal{U}$. *Then,*

$$\mathbb{E}\left[\max_{Q \in \mathcal{U}} \left|\frac{1}{n_Q}\sum_{i=1}^{n_Q} f_Q\left(X_Q^{(i)}\right)\right|\right] \leq 4\sqrt{\frac{\log k}{\min_{Q \in \mathcal{U}} n_Q}}$$

*Proof.* Since each $f_Q \in [-1, 1]$, we have that

$$D\left(\{f_Q\}_{Q \in \mathcal{U}}\right) \leq \sqrt{\max_{Q \in \mathcal{U}} \frac{1}{n_Q}} = \sqrt{\frac{1}{\min_{Q \in \mathcal{U}} n_Q}}$$

Hence, combining Theorem A.2 and Corollary A.1 yields

$$\mathbb{E}\left[\max_{Q \in \mathcal{U}} \left|\frac{1}{n_Q}\sum_{i=1}^{n_Q} f_Q\left(X_Q^{(i)}\right)\right|\right] \leq 2\,\mathfrak{R}_T\left(\{f_Q\}_{Q \in \mathcal{U}}\right) \leq 4\sqrt{\log k}\,\mathbb{E}\left[D\left(\{f_Q\}_{Q \in \mathcal{U}}\right)\right] \leq 4\sqrt{\frac{\log k}{\min_{Q \in \mathcal{U}} n_Q}}$$

∎

To obtain the second bound, we require a more refined analysis. We begin by introducing two simple auxiliary lemmas.

**Lemma A.3.** *Let* $b, c > 0$ *and suppose that* $x^2 \leq bx + c$. *Then,* $x \leq b + \sqrt{c}$.

*Proof.* Define quadratic $p(z) := z^2 - bz - c$, so that $p(x) \leq 0$. Since $p(0) = -c < 0$, consider its roots $r_1 < 0 < r_2$. Then, $p$ is positive on $(r_2, \infty)$ and, thus,

$$x \leq r_2 = \frac{b + \sqrt{b^2 + 4c}}{2} \leq b + \sqrt{c}$$

∎

**Lemma A.4** (Variance of Lipschitz functions). *Let $Z \in \mathcal{Z} \subset \mathbb{R}$ be a random variable, and suppose that $f : \mathcal{Z} \to \mathbb{R}$ is L-Lipschitz. Then,*

$$\mathrm{Var}\left(f\left(Z\right)\right) \le 2L^2 \mathrm{Var}\left(Z\right)$$

*Proof.* Let $Z'$ be an independent copy of $Z$. Then,

$$
\begin{aligned}
\mathrm{Var}\left(f\left(Z\right)\right) &= \mathbb{E}\left[\left(f\left(Z\right) - \mathbb{E}\left[f\left(Z'\right)\right]\right)^2\right] \\
&= \mathbb{E}\left[\mathbb{E}\left[f\left(Z\right) - f\left(Z'\right)|Z\right]^2\right] \\
&\le \mathbb{E}\left[\left(f\left(Z\right) - f\left(Z'\right)\right)^2\right] && \text{Jensen's} \\
&\le L^2 \mathbb{E}\left[\left(Z - Z'\right)^2\right] && \text{Lipschitzness} \\
&= 2L^2 \left\{\mathrm{Var}\left(Z\right) + \mathbb{E}\left[\left(Z - \mathbb{E}\left[Z\right]\right)\left(\mathbb{E}\left[Z\right] - Z'\right)\right]\right\} && Z \overset{(d)}{=} Z' \\
&= 2L^2 \mathrm{Var}\left(Z\right) && Z \perp\!\!\!\perp Z'
\end{aligned}
$$

∎

Borrowing ideas from (Giné & Nickl, 2021, Corollary 3.5.7), we then conclude the second target bound.

**Theorem A.3.** *Suppose that $\mathcal{X} \subset \mathbb{R}$. Let $\{f_Q : \mathcal{X} \to [-1, 1]\}_{Q \in \mathcal{U}}$ be a collection of functions such that $\mathbb{E}\left[f_Q\left(X_Q\right)\right] = 0$ and $f_Q$ is L-Lipschitz for each $Q \in \mathcal{U}$. Then,*

$$\mathbb{E}\left[\max_{Q \in \mathcal{U}} \left|\frac{1}{n_Q} \sum_{i=1}^{n_Q} f_Q\left(X_Q^{(i)}\right)\right|\right] \le \frac{16 \log k}{\min_{Q \in \mathcal{U}} n_Q} + 4L\sigma_T\sqrt{2 \log k}$$

*Proof.* We begin with the following observation: from Jensen's, we obtain

$$C := \sqrt{\log k}\,\mathbb{E}\left[D\left(\{f_Q\}_{Q \in \mathcal{U}}\right)\right] \le \sqrt{\left(\log k\right)\mathbb{E}\left[D\left(\{f_Q\}_{Q \in \mathcal{U}}\right)^2\right]}$$

Next, we bound the expectation on the right-hand side:

$$
\begin{aligned}
\mathbb{E}\left[D\left(\{f_Q\}_{Q \in \mathcal{U}}\right)^2\right] &= \mathbb{E}\left[\max_{Q \in \mathcal{U}} \sum_{i=1}^{n_Q} \left(\frac{f_Q\left(X_Q^{(i)}\right)}{n_Q}\right)^2\right] \\
&= \mathbb{E}\left[\max_{Q \in \mathcal{U}} \sum_{i=1}^{n_Q} \left\{\left(\frac{f_Q\left(X_Q^{(i)}\right)}{n_Q}\right)^2 - \mathbb{E}\left[\left(\frac{f_Q\left(X_Q\right)}{n_Q}\right)^2\right] + \mathbb{E}\left[\left(\frac{f_Q\left(X_Q\right)}{n_Q}\right)^2\right]\right\}\right] \\
&\le \underbrace{\max_{Q \in \mathcal{U}} \left\{\frac{\mathbb{E}\left[f_Q^2\left(X_Q\right)\right]}{n_Q}\right\}}_{=:(*_1)} + \underbrace{\mathbb{E}\left[\max_{Q \in \mathcal{U}} \left|\frac{1}{n_Q}\sum_{i=1}^{n_Q}\left\{\frac{f_Q^2\left(X_Q^{(i)}\right)}{n_Q} - \mathbb{E}\left[\frac{f_Q^2\left(X_Q\right)}{n_Q}\right]\right\}\right|\right]}_{=:(*_2)}
\end{aligned}
$$

From Lemma A.4 and the fact that $\mathbb{E}\left[f_Q\left(X_Q\right)\right] = 0$, we know that

$$(*_1) = \max_{Q \in \mathcal{U}} \frac{\mathrm{Var}\left(f_Q\left(X_Q\right)\right)}{n_Q} \le 2L^2 \max_{Q \in \mathcal{U}} \frac{\sigma_Q^2}{n_Q} = 2L^2\sigma_T^2$$

As for $(*_2)$, we can apply Theorem A.2 on functions $h_Q(x) := \frac{f_Q^2(x)}{n_Q}$ to conclude that

$$
\begin{aligned}
(*_2) &\leq 2\,\mathfrak{R}_T\left(\{h_Q\}_{Q\in\mathcal{U}}\right) && \text{Thm. A.2}\\
&\leq 4\sqrt{\log k}\,\mathbb{E}\left[D\left(\{h_Q\}_{Q\in\mathcal{U}}\right)\right] && \text{Cor. A.1}\\
&= 4\sqrt{\log k}\,\mathbb{E}\left[\max_{Q\in\mathcal{U}}\sqrt{\sum_{i=1}^{n_Q}\left(\frac{f_Q\left(X_Q^{(i)}\right)}{n_Q}\right)^4}\right]\\
&\leq 4\sqrt{\log k}\,\mathbb{E}\left[\max_{Q\in\mathcal{U}}\left\{\frac{1}{n_Q}\sqrt{\sum_{i=1}^{n_Q}\left(\frac{f_Q\left(X_Q^{(i)}\right)}{n_Q}\right)^2}\right\}\right] && f_Q^4 \leq f_Q^2\\
&\leq 4\sqrt{\log k}\max_{Q\in\mathcal{U}}\left\{\frac{1}{n_Q}\right\}\mathbb{E}\left[D\left(\{f_Q\}_{Q\in\mathcal{U}}\right)\right]\\
&= \frac{4}{\min_{Q\in\mathcal{U}} n_Q}C
\end{aligned}
$$

In other words, we have that

$$
C^2 \leq (\log k)\,\mathbb{E}\left[D\left(\{f_Q\}_{Q\in\mathcal{U}}\right)^2\right] \leq \frac{4\log k}{\min_{Q\in\mathcal{U}} n_Q}C + 2L^2\sigma_T^2\log k
$$

Then, Lemma A.3 implies that

$$
C \leq \frac{4\log k}{\min_{Q\in\mathcal{U}} n_Q} + L\sigma_T\sqrt{2\log k}
$$

Combining this with Theorem A.2 and Corollary A.1, we conclude that

$$
\mathbb{E}\left[\max_{Q\in\mathcal{U}}\left|\frac{1}{n_Q}\sum_{i=1}^{n_Q}f_Q\left(X_Q^{(i)}\right)\right|\right] \leq 2\,\mathfrak{R}_T\left(\{f_Q\}_{Q\in\mathcal{U}}\right) \leq 4C \leq \frac{16\log k}{\min_{Q\in\mathcal{U}} n_Q} + 4L\sigma_T\sqrt{2\log k}
$$

∎

## B  EMPIRICAL PROCESS CONCENTRATION INEQUALITIES

Again, suppose that $\mathcal{U} \subset \mathcal{P}(\mathcal{X})$ is a collection of $k$ distributions, and define independent variables $X := \left(X_Q^{(i)}\right)_{Q\in\mathcal{U},i\in[n_Q]}$, where $n_Q \in \mathbb{N}$ and $\left(X_Q^{(i)}\right)_{i\in[n_Q]} \overset{iid}{\sim} Q$ for each $Q \in \mathcal{U}$. Our object of interest in this section is the random variable

$$
Z_f := \min_{Q\in\mathcal{U}}\frac{1}{n_Q}\sum_{i=1}^{n_Q}f\left(X_Q^{(i)}\right)
$$

for a function $f : \mathcal{X} \to \mathbb{R}$. As will become clear later, our primary goal will be to obtain concentration inequalities on $Z_{f,g} := Z_f - Z_g$.

### B.1  MCDIARMID

To obtain the UE regret bound, we will apply a very simple concentration inequality, called McDiarmid's inequality (e.g., see (Boucheron et al., 2013 - 2013, Theorem 6.2)). Here, we specialize to

$$
Z_f = \min_{Q\in\mathcal{U}}\frac{1}{n}\sum_{i=1}^{n}f\left(X_Q^{(i)}\right)
$$

Let us define the function $\Phi_f : \left(\mathcal{X}^k\right)^n \to [0,1]$ by $\Phi_f\left(\mathbf{x}_1, \ldots, \mathbf{x}_n\right) \coloneqq \min_{Q \in \mathcal{U}} \frac{1}{n} \sum_{i=1}^n f\left(x_Q^i\right)$, where each $\mathbf{x}_i = \left(x_Q^i\right)_{Q \in \mathcal{U}} \in \mathcal{X}^k$. Then, we can write $Z_f = \Phi_f(X)$, where we view $X$ as $n$ vectors of dimension $k$. Next, we show that $\Phi_f$ satisfies the bounded differences property when $f$ is bounded.

**Proposition B.1** (Bounded differences). *Suppose that $f : \mathcal{X} \to [0,1]$. Then,*

$$\max_{i \in [n]} \sup_{\mathbf{x}_1, \ldots, \mathbf{x}_n, \mathbf{y} \in \mathcal{X}^k} \left|\Phi_f\left(\mathbf{x}_1, \ldots, \mathbf{x}_n\right) - \Phi_f\left(\mathbf{x}_1, \ldots, \mathbf{x}_{i-1}, \mathbf{y}, \mathbf{x}_{i+1}, \ldots, \mathbf{x}_n\right)\right| \leq \frac{1}{n}$$

*Proof.* Let us begin with a simple observation: for real-valued functions $g, h : \mathcal{Z} \to \mathbb{R}$, where $\mathcal{Z}$ is any domain, we have that

$$\inf_{z' \in \mathcal{Z}} g\left(z'\right) - \inf_{z \in \mathcal{Z}} h\left(z\right) = \sup_{z \in \mathcal{Z}}\left\{\inf_{z' \in \mathcal{Z}} g\left(z'\right) - h\left(z\right)\right\} \leq \sup_{z \in \mathcal{Z}}\left\{g\left(z\right) - h\left(z\right)\right\} \leq \sup_{z \in \mathcal{Z}}\left|g\left(z\right) - h\left(z\right)\right|$$

By symmetry, it then follows that $\left|\inf_{z' \in \mathcal{Z}} g\left(z'\right) - \inf_{z \in \mathcal{Z}} h\left(z\right)\right| \leq \sup_{z \in \mathcal{Z}}\left|g\left(z\right) - h\left(z\right)\right|$. Next, fix any index $i \in [n]$ and inputs $\mathbf{x}_1, \ldots, \mathbf{x}_n, \mathbf{y} \coloneqq \left(y_Q\right)_{Q \in \mathcal{U}} \in \mathcal{X}^k$, and define vectors $\mathbf{x} \coloneqq \left(\mathbf{x}_1, \ldots, \mathbf{x}_n\right)$ and $\mathbf{x}' \coloneqq \left(\mathbf{x}_1, \ldots, \mathbf{x}_{i-1}, \mathbf{y}, \mathbf{x}_{i+1}, \ldots, \mathbf{x}_n\right)$. Then, from our initial observation, we know that

$$\begin{aligned}
\left|\Phi_f\left(\mathbf{x}\right) - \Phi_f\left(\mathbf{x}'\right)\right| &= \frac{1}{n}\left|\min_{Q' \in \mathcal{U}}\left\{\sum_{j=1}^n f\left(x_{Q'}^j\right)\right\} - \min_{Q \in \mathcal{U}}\left\{f\left(y_Q\right) + \sum_{j \in [n]: j \neq i} f\left(x_Q^j\right)\right\}\right| \\
&\leq \frac{1}{n}\max_{Q \in \mathcal{U}}\left|\sum_{j=1}^n f\left(x_Q^j\right) - \left[f\left(y_Q\right) + \sum_{j \in [n]: j \neq i} f\left(x_Q^j\right)\right]\right| \\
&\leq \frac{1}{n}\max_{Q \in \mathcal{U}}\left|f\left(x_Q^i\right) - f\left(y_Q\right)\right| \\
&\leq \frac{1}{n}
\end{aligned}$$

$\blacksquare$

When the inequality in Proposition B.1 holds, we say that $\Phi_f$ satisfies the *bounded differences property* with constant parameter $\frac{1}{n}$. This immediately implies the next claim.

**Corollary B.1.** *For any two functions $f, g : \mathcal{X} \to [0,1]$, the function $\Phi_f - \Phi_g$ satisfies the bounded differences property with constant parameter $\frac{2}{n}$.*

*Proof.* Using the same variables $\mathbf{x}$ and $\mathbf{x}'$ as in the proof of Proposition B.1, we obtain

$$\left|\left[\Phi_f\left(\mathbf{x}\right) - \Phi_g\left(\mathbf{x}\right)\right] - \left[\Phi_f\left(\mathbf{x}'\right) - \Phi_g\left(\mathbf{x}'\right)\right]\right| \leq \left|\Phi_f\left(\mathbf{x}\right) - \Phi_f\left(\mathbf{x}'\right)\right| + \left|\Phi_g\left(\mathbf{x}\right) - \Phi_g\left(\mathbf{x}'\right)\right| \leq \frac{2}{n}$$

$\blacksquare$

Via McDiarmid's, this property then directly yields the following concentration result.

**Corollary B.2.** *Let $f, g : \mathcal{X} \to [0,1]$. Then,*

$$\mathbb{P}\left(Z_{f,g} - \mathbb{E}\left[Z_{f,g}\right] \geq t\right) \leq \exp\left(-\frac{nt^2}{2}\right) \quad \forall t \geq 0$$

*Proof.* Since $Z_{f,g} = \left(\Phi_f - \Phi_g\right)(X)$ and $X$ has independent components, we simply apply Corollary B.1 and McDiarmid's. $\blacksquare$

## B.2 BERNSTEIN

In contrast to McDiarmid's inequality, our next goal is to derive a more involved bound that additionally scales with the variance. To this end, we sort the sample sizes: $0 =: n_{(0)} \le n_{(1)} \le \cdots \le n_{(k)}$ and let $Q_{(j)} \in \mathcal{U}$ be such that $n_{Q_{(j)}} = n_{(j)}$. Our analysis then relies on the following:

$$
V_T := \sum_{j=1}^{k} \left( n_{(j)} - n_{(j-1)} \right) \mathbb{E} \left[ \max_{r \in \{j, \ldots, k\}} \frac{1}{n_{(r)}^2} \left[ X_{Q_{(r)}} - \mu_{Q_{(r)}} \right]^2 \right]
$$

$$
\Sigma_T^2 := \mathbb{E} \left[ \max_{Q \in \mathcal{U}} \frac{1}{n_Q^2} \sum_{i=1}^{n_Q} \left( X_Q^{(i)} - \mu_Q \right)^2 \right]
$$

$$
\sigma_T^2 := \max_{Q \in \mathcal{U}} \frac{\sigma_Q^2}{n_Q}
$$

**Theorem B.1.** *Suppose that $\mathcal{X} \subset \mathbb{R}$ and $f, g : \mathcal{X} \to [0, M]$ are L-Lipschitz. Then,*

$$
\mathbb{P} \left( Z_{f,g} - \mathbb{E} \left[ Z_{f,g} \right] \ge t \right) \le \exp \left( -\frac{t^2}{16 L^2 \left( 2\sigma_T^2 + \Sigma_T^2 + 6 V_T \right) + \frac{2\sqrt{6} M t}{\min_{Q \in \mathcal{U}} n_Q}} \right) \quad \forall t \ge 0
$$

### B.2.1 PRELIMINARIES

To prove Theorem B.1, we must first state some standard results and definitions from the theory of concentration of measure. We do not prove most results stated, and refer to Boucheron et al. (2013 - 2013) for further reference.

We say that a random variable $X \in \mathbb{R}$ is *sub-gamma on the right tail* with parameters $\nu, c > 0$ if

$$
\log \mathbb{E} \left[ e^{\lambda(X - \mathbb{E}[X])} \right] \le \frac{\nu^2 \lambda^2}{2(1 - c\lambda)} \quad \forall \lambda \in \left[ 0, \frac{1}{c} \right)
$$

We denote the class of such variables by $\Gamma_+ (\nu, c)$. Due to the decaying tail, we get the following concentration bound.

**Proposition B.2** (Sub-gamma concentration). *Let $X \in \Gamma_+ (\nu, c)$. Then,*

$$
\mathbb{P} \left( X - \mathbb{E} \left[ X \right] \ge t \right) \le \exp \left( -\frac{t^2}{2(\nu^2 + ct)} \right) \quad \forall t \ge 0
$$

*Proof.* See (Boucheron et al., 2013 - 2013, Section 2.4). ∎

Next, we introduce the notion of self-bounding functions: we say that a nonnegative function $f : \mathcal{X}^n \to \mathbb{R}_+$ has the *self-bounding property* if there exists functions $\left\{ f_i : \mathcal{X}^{n-1} \to \mathbb{R} \right\}_{i \in [n]}$ such that

$$
f(\mathbf{x}) - f_i (\mathbf{x}_{\backslash i}) \in [0, 1] \quad \text{and} \quad \sum_{i=1}^{n} \left[ f(\mathbf{x}) - f_i (\mathbf{x}_{\backslash i}) \right] \le f(\mathbf{x})
$$

for all $i \in [n]$ and $\mathbf{x} \in \mathcal{X}^n$, where we define $\mathbf{x}_{\backslash i} := (x_1, \ldots, x_{i-1}, x_{i+1}, \ldots, x_n)$. A simple observation about such functions is that they are closed under convex combinations.

**Lemma B.1** (Convex combination of self-bounding functions). *Suppose that $f$ and $g$ satisfy the self-bounding property and let $\alpha \in [0, 1]$. Then, $\alpha f + (1 - \alpha) g$ also satisfies the self-bounding property.*

*Proof.* Let $\{f_i\}$ and $\{g_i\}$ be the functions satisfying the self-bounding property, and define $h := \alpha f + (1 - \alpha) g$ and $h_i := \alpha f_i + (1 - \alpha) g_i$. Then, for any $i \in [n]$ and $\mathbf{x} \in \mathcal{X}^n$,

$$
h(\mathbf{x}) - h_i (\mathbf{x}_{\backslash i}) = \alpha \left[ f(\mathbf{x}) - f_i (\mathbf{x}_{\backslash i}) \right] + (1 - \alpha) \left[ g(\mathbf{x}) - g_i (\mathbf{x}_{\backslash i}) \right] \in [0, 1]
$$

and

$$\sum_{i=1}^{n} \left[ h\left(\mathbf{x}\right) - h_i\left(\mathbf{x}_{\setminus i}\right) \right] = \alpha \sum_{i=1}^{n} \left[ f\left(\mathbf{x}\right) - f_i\left(\mathbf{x}_{\setminus i}\right) \right] + (1-\alpha)\sum_{i=1}^{n} \left[ g\left(\mathbf{x}\right) - g_i\left(\mathbf{x}_{\setminus i}\right) \right]$$
$$\leq \alpha f\left(\mathbf{x}\right) + (1-\alpha) g\left(\mathbf{x}\right)$$
$$= h\left(\mathbf{x}\right)$$

■

The reason for introducing such functions is that they possess a favorable bound on their cumulant-generating function (cgf).

**Proposition B.3** (Cgf of self-bounding functions). *Suppose that $f : \mathcal{X}^n \to \mathbb{R}_+$ has the self-bounding property and let $X^n = (X_1, \dots, X_n)$ be independent random variables. Then,*

$$\log \mathbb{E}\left[ e^{\lambda f(X^n)} \right] \leq \left(e^{\lambda} - 1\right) \mathbb{E}\left[ f\left(X^n\right) \right] \quad \forall \lambda \in \mathbb{R}$$

*Proof.* See (Boucheron et al., 2013 - 2013, Theorem 6.12). ■

The last tool we need employs symmetrization once again. For the next result and the development that follows, we omit the parentheses in $a_+^2 := (a_+)^2$; that is, we take the positive part before squaring.

**Proposition B.4** (Exponential Efron-Stein). *Suppose that $X^n = (X_1, \dots, X_n)$ are independent random variables and let $W^n = (W_1, \dots, W_n)$ be independent copies of them. Given a nonnegative function $f : \mathcal{X}^n \to \mathbb{R}_+$, define variables $Z := f(X^n)$ and its symmetrized counterpart*

$$Z_i' := f\left(X_1, \dots, X_{i-1}, W_i, X_{i+1}, \dots, X_n\right) \quad \forall i \in [n]$$

*Additionally, let*

$$V^+ := \sum_{i=1}^{n} \mathbb{E}\left[ \left(Z - Z_i'\right)_+^2 \Big| X^n \right]$$

*Then, we have that*

$$\log \mathbb{E}\left[ e^{\lambda(Z - \mathbb{E}[Z])} \right] \leq \frac{\theta \lambda}{1 - \theta \lambda} \log \mathbb{E}\left[ e^{\frac{\lambda V^+}{\theta}} \right]$$

*for any $\theta, \lambda > 0$ such that $\theta \lambda < 1$.*

*Proof.* See (Boucheron et al., 2013 - 2013, Theorem 6.16). ■

*Proof of Theorem B.1.* To conclude our main result, we begin with a more general setup: let $X := \left( X_Q^{(i)} \right)_{Q \in \mathcal{U}, i \in [n]}$, where $n \in \mathbb{N}$, be a collection of independent $\mathcal{X}$-valued random variables, and let $X^{(i)} := \left( X_Q^{(i)} \right)_{Q \in \mathcal{U}}$ for each $i \in [n]$. We de not impose any assumptions on their distributions. Our random variables of interest will be

$$Z_f := \min_{Q \in \mathcal{U}} \sum_{i=1}^{n} f_Q\left( X_Q^{(i)} \right) \quad \text{and} \quad Z_{f,g} := Z_f - Z_g$$

for collections of functions $f = \left\{ f_Q : \mathcal{X} \to \left[0, \frac{b}{\sqrt{6}}\right] \right\}_{Q \in \mathcal{U}}$ and $g = \left\{ g_Q : \mathcal{X} \to \left[0, \frac{b}{\sqrt{6}}\right] \right\}_{Q \in \mathcal{U}}$, where $b > 0$. Define

$$\mu_{f,i,Q} := \mathbb{E}\left[ f_Q\left( X_Q^{(i)} \right) \right] \quad \text{and} \quad \sigma_{f,i,Q}^2 := \mathrm{Var}\left( f_Q\left( X_Q^{(i)} \right) \right)$$

Similarly, consider the variance variants:

$$V_f := \sum_{i=1}^{n} \mathbb{E}\left[\max_{Q \in \mathcal{U}}\left[f_Q\left(X_Q^{(i)}\right) - \mu_{f,i,Q}\right]^2\right]$$

$$\Sigma_f^2 := \mathbb{E}\left[\max_{Q \in \mathcal{U}}\sum_{i=1}^{n}\left[f_Q\left(X_Q^{(i)}\right) - \mu_{f,i,Q}\right]^2\right]$$

$$\sigma_f^2 := \max_{Q \in \mathcal{U}}\sum_{i=1}^{n}\sigma_{f,i,Q}^2$$

Following the analysis of (Boucheron et al., 2013 - 2013, Theorem 12.2), we will use the tools provided and proceed in 5 steps:

1. Upper bound $V^+$.

2. Apply exponential Efron-Stein along with the bound on $V^+$.

3. Show the self-boundedness of certain functions and apply the cgf bound.

4. Show that $Z_{f,g}$ is sub-gamma and apply the tail bound.

5. Specialize the analysis to the original setting.

### B.2.2 BOUNDING $V^+$

For each pair $(i, Q) \in [n] \times \mathcal{U}$, let $W_Q^{(i)}$ be an independent copy of $X_Q^{(i)}$ and define $W^{(i)} := \left(W_Q^{(i)}\right)_{Q \in \mathcal{U}}$. Moreover, define

$$Y_i := \left(X^{(1)}, \ldots, X^{(i-1)}, W^{(i)}, X^{(i+1)}, \ldots, X^{(n)}\right) \quad \forall i \in [n]$$

and function $\Phi_{f,g} : \left(\mathcal{X}^k\right)^n \to \mathbb{R}$ by

$$\Phi_{f,g}\left(\mathbf{x}_1, \ldots, \mathbf{x}_n\right) := \min_{Q \in \mathcal{U}}\sum_{i=1}^{n} f_Q\left(x_Q^i\right) - \min_{Q' \in \mathcal{U}}\sum_{i=1}^{n} g_{Q'}\left(x_{Q'}^i\right)$$

where $\mathbf{x}_i = \left(x_Q^i\right)_{Q \in \mathcal{U}} \in \mathcal{X}^k$ for each $i \in [n]$. In what follows, we will use the more compact notation $\mathbf{x} = (\mathbf{x}_1, \ldots, \mathbf{x}_n)$. Note that $Z_{f,g} = \Phi_{f,g}(X)$ and

$$Z_i' := \Phi_{f,g}(Y_i)$$

$$= \min_{Q \in \mathcal{U}}\left\{f_Q\left(W_Q^{(i)}\right) + \sum_{j \in [n]: j \neq i} f_Q\left(X_Q^{(j)}\right)\right\} - \min_{Q' \in \mathcal{U}}\left\{g_{Q'}\left(W_{Q'}^{(i)}\right) + \sum_{j \in [n]: j \neq i} g_{Q'}\left(X_{Q'}^{(j)}\right)\right\}$$

Given functions $h = \{h_Q : \mathcal{X} \to \mathbb{R}\}_{Q \in \mathcal{U}}$, define minimizer $\hat{Q}_h : \left(\mathcal{X}^k\right)^n \to \mathcal{U}$ by

$$\hat{Q}_h(\mathbf{x}) := \operatorname*{argmin}_{Q \in \mathcal{U}}\sum_{i=1}^{n} h_Q\left(x_Q^i\right)$$

so that

$$\Phi_{f,g}(\mathbf{x}) = \sum_{i=1}^{n} f_{\hat{Q}_f(\mathbf{x})}\left(x_{\hat{Q}_f(\mathbf{x})}^i\right) - \sum_{i=1}^{n} g_{\hat{Q}_g(\mathbf{x})}\left(x_{\hat{Q}_g(\mathbf{x})}^i\right)$$

and

$$\sum_{i=1}^{n} f_{\hat{Q}_f(\mathbf{x})}\left(x_{\hat{Q}_f(\mathbf{x})}^i\right) - \sum_{i=1}^{n} g_Q\left(x_Q^i\right) \leq \Phi_{f,g}(\mathbf{x}) \leq \sum_{i=1}^{n} f_{Q'}\left(x_{Q'}^i\right) - \sum_{i=1}^{n} g_{\hat{Q}_g(\mathbf{x})}\left(x_{\hat{Q}_g(\mathbf{x})}^i\right)$$

for any $\mathbf{x} \in (\mathcal{X}^k)^n$ and $Q, Q' \in \mathcal{U}$. Choosing $Q = \hat{Q}_g(X)$ and $Q' = \hat{Q}_f(Y_i)$ below then yields

$$Z_{f,g} - Z'_i = \Phi_{f,g}(X) - \Phi_{f,g}(Y_i)$$

$$\leq \sum_{j=1}^n f_{\hat{Q}_f(Y_i)}\left(X^{(j)}_{\hat{Q}_f(Y_i)}\right) - \sum_{j=1}^n g_{\hat{Q}_g(X)}\left(X^{(j)}_{\hat{Q}_g(X)}\right)$$

$$- \left[f_{\hat{Q}_f(Y_i)}\left(W^{(i)}_{\hat{Q}_f(Y_i)}\right) + \sum_{j\in[n]:j\neq i} f_{\hat{Q}_f(Y_i)}\left(X^{(j)}_{\hat{Q}_f(Y_i)}\right)\right]$$

$$+ \left[g_{\hat{Q}_g(X)}\left(W^{(i)}_{\hat{Q}_g(X)}\right) + \sum_{j\in[n]:j\neq i} g_{\hat{Q}_g(X)}\left(X^{(j)}_{\hat{Q}_g(X)}\right)\right]$$

$$= f_{\hat{Q}_f(Y_i)}\left(X^{(i)}_{\hat{Q}_f(Y_i)}\right) - f_{\hat{Q}_f(Y_i)}\left(W^{(i)}_{\hat{Q}_f(Y_i)}\right) + g_{\hat{Q}_g(X)}\left(W^{(i)}_{\hat{Q}_g(X)}\right) - g_{\hat{Q}_g(X)}\left(X^{(i)}_{\hat{Q}_g(X)}\right)$$

Then,

$$(Z_{f,g} - Z'_i)^2_+$$

$$\leq \left[f_{\hat{Q}_f(Y_i)}\left(X^{(i)}_{\hat{Q}_f(Y_i)}\right) - f_{\hat{Q}_f(Y_i)}\left(W^{(i)}_{\hat{Q}_f(Y_i)}\right) + g_{\hat{Q}_g(X)}\left(W^{(i)}_{\hat{Q}_g(X)}\right) - g_{\hat{Q}_g(X)}\left(X^{(i)}_{\hat{Q}_g(X)}\right)\right]^2$$

$$\leq 2\left[f_{\hat{Q}_f(Y_i)}\left(X^{(i)}_{\hat{Q}_f(Y_i)}\right) - f_{\hat{Q}_f(Y_i)}\left(W^{(i)}_{\hat{Q}_f(Y_i)}\right)\right]^2 + 2\left[g_{\hat{Q}_g(X)}\left(X^{(i)}_{\hat{Q}_g(X)}\right) - g_{\hat{Q}_g(X)}\left(W^{(i)}_{\hat{Q}_g(X)}\right)\right]^2$$
$$\text{(B.1)}$$

Recall that our goal is to bound $V^+ = \sum_{i=1}^n \mathbb{E}\left[(Z_{f,g} - Z'_i)^2_+ \Big| X\right]$. We begin with the second term: by adding and subtracting $\mu_{g,i,\hat{Q}_g(X)}$, expanding the square and noting that the cross term is 0 under the conditional expectation, we get that

$$\sum_{i=1}^n \mathbb{E}\left[\left[g_{\hat{Q}_g(X)}\left(X^{(i)}_{\hat{Q}_g(X)}\right) - g_{\hat{Q}_g(X)}\left(W^{(i)}_{\hat{Q}_g(X)}\right)\right]^2 \Big| X\right]$$

$$= \sum_{i=1}^n \left\{\left[g_{\hat{Q}_g(X)}\left(X^{(i)}_{\hat{Q}_g(X)}\right) - \mu_{g,i,\hat{Q}_g(X)}\right]^2 + \mathbb{E}\left[\left[g_{\hat{Q}_g(X)}\left(W^{(i)}_{\hat{Q}_g(X)}\right) - \mu_{g,i,\hat{Q}_g(X)}\right]^2 \Big| X\right]\right\}$$

$$\leq \underbrace{\max_{Q\in\mathcal{U}}\left\{\sum_{i=1}^n \left[g_Q\left(X^{(i)}_Q\right) - \mu_{g,i,Q}\right]^2\right\}}_{=:\Gamma_g} + \underbrace{\max_{Q\in\mathcal{U}}\left\{\sum_{i=1}^n \mathbb{E}\left[\left[g_Q\left(W^{(i)}_Q\right) - \mu_{g,i,Q}\right]^2\right]\right\}}_{=\sigma_g^2}$$

Note that we were able to upper bound via a maximization outside of the sum since the $Q$ indices were fixed w.r.t. $i$. The first term in (B.1) is not so readily bounded due to the dependence of $Y_i$ on $i$. Hence, we rely on a weaker approach: for each $i \in [n]$, we have that

$$\left[f_{\hat{Q}_f(Y_i)}\left(X^{(i)}_{\hat{Q}_f(Y_i)}\right) - f_{\hat{Q}_f(Y_i)}\left(W^{(i)}_{\hat{Q}_f(Y_i)}\right)\right]^2$$

$$\leq 2\left[f_{\hat{Q}_f(Y_i)}\left(X^{(i)}_{\hat{Q}_f(Y_i)}\right) - \mu_{f,i,\hat{Q}_f(Y_i)}\right]^2 + 2\left[f_{\hat{Q}_f(Y_i)}\left(W^{(i)}_{\hat{Q}_f(Y_i)}\right) - \mu_{f,i,\hat{Q}_f(Y_i)}\right]^2$$

$$\leq 2\max_{Q\in\mathcal{U}}\left\{\left[f_Q\left(X^{(i)}_Q\right) - \mu_{f,i,Q}\right]^2\right\} + 2\max_{Q\in\mathcal{U}}\left\{\left[f_Q\left(W^{(i)}_Q\right) - \mu_{f,i,Q}\right]^2\right\}$$

Summing and taking conditional expectations then yields

$$\sum_{i=1}^n \mathbb{E}\left[\left[f_{\hat{Q}_f(Y_i)}\left(X^{(i)}_{\hat{Q}_f(Y_i)}\right) - f_{\hat{Q}_f(Y_i)}\left(W^{(i)}_{\hat{Q}_f(Y_i)}\right)\right]^2 \Big| X\right]$$

$$\leq 2\underbrace{\sum_{i=1}^n \max_{Q\in\mathcal{U}}\left\{\left[f_Q\left(X^{(i)}_Q\right) - \mu_{f,i,Q}\right]^2\right\}}_{=:T_f} + 2\underbrace{\sum_{i=1}^n \mathbb{E}\left[\max_{Q\in\mathcal{U}}\left\{\left[f_Q\left(W^{(i)}_Q\right) - \mu_{f,i,Q}\right]^2\right\}\right]}_{=V_f}$$

Finally, by putting everything together, we can obtain the upper bound

$$V^+ \leq 2 \left( \Gamma_g + \sigma_g^2 \right) + 4 \left( T_f + V_f \right)$$

where $\mathbb{E}\left[\Gamma_g\right] = \Sigma_g^2$ and $\mathbb{E}\left[T_f\right] = V_f$.

### B.2.3 Efron-Stein

Next, we apply exponential Efron-Stein (Proposition B.4): for $\lambda \in \left[0, b^{-1}\right)$, we have that

$$\begin{aligned}
\log \mathbb{E}\left[e^{\lambda(Z_{f,g} - \mathbb{E}[Z_{f,g}])}\right] &\leq \frac{b\lambda}{1 - b\lambda} \log \mathbb{E}\left[e^{\lambda b^{-1} V^+}\right] \\
&\leq \frac{b\lambda}{1 - b\lambda} \log \mathbb{E}\left[e^{\lambda b^{-1}\left[2\left(\Gamma_g + \sigma_g^2\right) + 4\left(T_f + V_f\right)\right]}\right] \\
&= \frac{b\lambda}{1 - b\lambda} \left\{ \log \mathbb{E}\left[e^{b\lambda\left[\frac{1}{3}\left(6b^{-2}\Gamma_g\right) + \frac{2}{3}\left(6b^{-2}T_f\right)\right]}\right] + \lambda b^{-1}\left(2\sigma_g^2 + 4V_f\right) \right\}
\end{aligned}$$
(B.2)

### B.2.4 Self-boundedness

To bound the cgf of $\frac{1}{3}\left(6b^{-2}\Gamma_g\right) + \frac{2}{3}\left(6b^{-2}T_f\right)$, we will show the self-boundedness of

$$h^{(1)}\left(\mathbf{x}\right) := 6b^{-2} \max_{Q \in \mathcal{U}} \sum_{i=1}^n \left[g_Q\left(x_Q^i\right) - \mu_{g,i,Q}\right]^2 \quad \text{and} \quad h^{(2)}\left(\mathbf{x}\right) := 6b^{-2} \sum_{i=1}^n \max_{Q \in \mathcal{U}} \left[f_Q\left(x_Q^i\right) - \mu_{f,i,Q}\right]^2$$

so that the function $\frac{1}{3}h^{(1)} + \frac{2}{3}h^{(2)}$ is also self-bounded by Lemma B.1 and we can thus bound the cgf of $\left(\frac{1}{3}h^{(1)} + \frac{2}{3}h^{(2)}\right)(X) = \frac{1}{3}\left(6b^{-2}\Gamma_g\right) + \frac{2}{3}\left(6b^{-2}T_f\right)$ using Proposition B.3. We begin by showing that $h^{(1)}$ is self-bounded: let

$$h_i^{(1)}\left(\mathbf{x}_{\backslash i}\right) := 6b^{-2} \max_{Q \in \mathcal{U}} \sum_{j \in [n]: j \neq i} \left[g_Q\left(x_Q^j\right) - \mu_{g,j,Q}\right]^2 \quad \forall i \in [n]$$

and define the maximizing distribution in $h^{(1)}$:

$$\tilde{Q}\left(\mathbf{x}\right) := \underset{Q \in \mathcal{U}}{\operatorname{argmax}} \sum_{i=1}^n \left[g_Q\left(x_Q^i\right) - \mu_{g,i,Q}\right]^2$$

Fix some $\mathbf{x} \in \left(\mathcal{X}^k\right)^n$ and $i \in [n]$. Clearly, we have that $h^{(1)}\left(\mathbf{x}\right) \geq h_i^{(1)}\left(\mathbf{x}_{\backslash i}\right)$. Moreover,

$$\begin{aligned}
h^{(1)}\left(\mathbf{x}\right) - h_i^{(1)}\left(\mathbf{x}_{\backslash i}\right) &= 6b^{-2}\left[\sum_{j=1}^n \left[g_{\tilde{Q}(\mathbf{x})}\left(x_{\tilde{Q}(\mathbf{x})}^j\right) - \mu_{g,i,\tilde{Q}(\mathbf{x})}\right]^2 - \max_{Q \in \mathcal{U}}\left\{\sum_{j \in [n]: j \neq i}\left[g_Q\left(x_Q^j\right) - \mu_{g,j,Q}\right]^2\right\}\right] \\
&\leq 6b^{-2}\left[g_{\tilde{Q}(\mathbf{x})}\left(x_{\tilde{Q}(\mathbf{x})}^i\right) - \mu_{g,i,\tilde{Q}(\mathbf{x})}\right]^2 \\
&\leq 1
\end{aligned}$$

where the last line follows from our assumption that $g_Q \in \left[0, \frac{b}{\sqrt{6}}\right]$. We can add up the bounds to get

$$\sum_{i=1}^n \left[h^{(1)}\left(\mathbf{x}\right) - h_i^{(1)}\left(\mathbf{x}_{\backslash i}\right)\right] \leq 6b^{-2} \sum_{i=1}^n \left[g_{\tilde{Q}(\mathbf{x})}\left(x_{\tilde{Q}(\mathbf{x})}^i\right) - \mu_{g,i,\tilde{Q}(\mathbf{x})}\right]^2 = h^{(1)}\left(\mathbf{x}\right)$$

Together, these show that $h^{(1)}$ is self-bounded. To show the same for $h^{(2)}$, consider the functions

$$h_i^{(2)}\left(\mathbf{x}_{\backslash i}\right) := 6b^{-2} \sum_{j \in [n]: j \neq i} \max_{Q \in \mathcal{U}} \left[f_Q\left(x_Q^j\right) - \mu_{f,j,Q}\right]^2$$

Again, we have that $h^{(2)}(\mathbf{x}) \geq h_i^{(2)}(\mathbf{x}_{\setminus i})$ and

$$h^{(2)}(\mathbf{x}) - h_i^{(2)}(\mathbf{x}_{\setminus i}) = 6b^{-2} \max_{Q \in \mathcal{U}} \left[ f_Q(x_Q^i) - \mu_{f,i,Q} \right]^2 \leq 1$$

$$\sum_{i=1}^n \left[ h^{(2)}(\mathbf{x}) - h_i^{(2)}(\mathbf{x}_{\setminus i}) \right] = h^{(2)}(\mathbf{x})$$

That is, $h^{(2)}$ is also self-bounded. As a result, Proposition B.3 implies that

$$\log \mathbb{E}\left[ e^{b\lambda\left[\frac{1}{3}\left(6b^{-2}\Gamma_g\right) + \frac{2}{3}\left(6b^{-2}T_f\right)\right]} \right] \leq \left(e^{b\lambda} - 1\right) \mathbb{E}\left[ \frac{1}{3}\left(6b^{-2}\Gamma_g\right) + \frac{2}{3}\left(6b^{-2}T_f\right) \right]$$

$$= \left(e^{b\lambda} - 1\right) b^{-2} \left(2\Sigma_g^2 + 4V_f\right)$$

$$\leq \lambda b^{-1}\left(4\Sigma_g^2 + 8V_f\right) \tag{B.3}$$

provided that $\lambda \in \left[0, b^{-1}\right)$, where in the last line we have used the inequality $e^x \leq 1 + 2x$ for $x \leq 1$.

### B.2.5 SUB-GAMMA TAIL

Finally, we can combine Equations (B.2) and (B.3) to get that

$$\log \mathbb{E}\left[ e^{\lambda(Z_{f,g} - \mathbb{E}[Z_{f,g}])} \right] \leq \frac{\lambda^2}{1 - b\lambda}\left(2\sigma_g^2 + 4\Sigma_g^2 + 12V_f\right) = \frac{\left(4\sigma_g^2 + 8\Sigma_g^2 + 24V_f\right)\lambda^2}{2\left(1 - b\lambda\right)}$$

for all $\lambda \in \left[0, b^{-1}\right)$. That is, $Z_{f,g} \in \Gamma_+\left(\sqrt{4\sigma_g^2 + 8\Sigma_g^2 + 24V_f}, b\right)$, which we know from Proposition B.2 yields the tail bound

$$\mathbb{P}\left(Z_{f,g} - \mathbb{E}\left[Z_{f,g}\right] \geq t\right) \leq \exp\left(-\frac{t^2}{2\left(4\sigma_g^2 + 8\Sigma_g^2 + 24V_f + bt\right)}\right) \quad \forall t \geq 0 \tag{B.4}$$

### B.2.6 ORIGINAL SETTING

Recall that our original variables of interest live in some set $\mathcal{X}_0 \subset \mathbb{R}$, and that sample sizes $n_Q$ may vary. Let $n := \max_{Q \in \mathcal{U}} n_Q$ and consider the space $\mathcal{X} = \mathcal{X}_0 \cup \{x_0\}$ for the setup of this proof, where $x_0 \notin \mathcal{X}_0$. Suppose that $\left(X_Q^{(i)}\right)_{i \in [n_Q]} \overset{iid}{\sim} Q$ and $X_Q^{(n_Q+1)} = \cdots = X_Q^{(n)} = x_0$ almost surely. Let $f : \mathcal{X}_0 \to \mathbb{R}$ be the $L$-Lipschitz function from the statement of Theorem B.1, and consider its extension $\tilde{f} : \mathcal{X} \to \mathbb{R}$ given by

$$\tilde{f}(x) := \begin{cases} f(x) & x \in \mathcal{X}_0 \\ 0 & x = x_0 \end{cases}$$

We apply the analysis above to the functions $f_Q := \frac{\tilde{f}}{n_Q}$, ensuring that

$$Z_f = \min_{Q \in \mathcal{U}} \frac{1}{n_Q} \sum_{i=1}^{n_Q} f\left(X_Q^{(i)}\right)$$

where the variables follow the appropriate distributions, as in the original goal. Note that $f_Q \in \left[0, \frac{M}{n_Q}\right]$, so that we can set $b = \frac{\sqrt{6}M}{\min_{Q \in \mathcal{U}} n_Q}$. We analogously define everything for $g$. Next, we apply Lemma A.4 under the Lipschitzness assumption to obtain

$$\sigma_g^2 = \max_{Q \in \mathcal{U}} \left\{ n_Q \operatorname{Var}\left(\frac{g\left(X_Q\right)}{n_Q}\right) \right\} \leq 2L^2 \max_{Q \in \mathcal{U}} \frac{\sigma_Q^2}{n_Q} = 2L^2 \sigma_T^2$$

For each $Q \in \mathcal{U}$, let $X_Q \sim Q$ be independent from $\left( X_Q^{(i)} \right)_{i \in [n_Q]}$. Then,

$$\Sigma_g^2 = \mathbb{E}\left[ \max_{Q \in \mathcal{U}} \frac{1}{n_Q^2} \sum_{i=1}^{n_Q} \left[ g\left(X_Q^{(i)}\right) - \mathbb{E}\left[g\left(X_Q\right)\right] \right]^2 \right]$$

$$= \mathbb{E}\left[ \max_{Q \in \mathcal{U}} \frac{1}{n_Q^2} \sum_{i=1}^{n_Q} \mathbb{E}\left[ g\left(X_Q^{(i)}\right) - g\left(X_Q\right) \Big| X_Q^{(i)} \right]^2 \right]$$

$$\leq L^2 \mathbb{E}\left[ \max_{Q \in \mathcal{U}} \frac{1}{n_Q^2} \sum_{i=1}^{n_Q} \mathbb{E}\left[ \left(X_Q^{(i)} - X_Q\right)^2 \Big| X_Q^{(i)} \right] \right] \qquad \text{Lipschitzness + Jensen's}$$

$$= L^2 \mathbb{E}\left[ \max_{Q \in \mathcal{U}} \frac{1}{n_Q^2} \sum_{i=1}^{n_Q} \left[ \left(X_Q^{(i)} - \mu_Q\right)^2 + \sigma_Q^2 \right] \right] \qquad \mathbb{E}\left[ \left(X_Q^{(i)} - \mu_Q\right)(\mu_Q - X_Q) \Big| X_Q^{(i)} \right] = 0$$

$$\leq L^2 \left\{ \mathbb{E}\left[ \max_{Q \in \mathcal{U}} \frac{1}{n_Q^2} \sum_{i=1}^{n_Q} \left(X_Q^{(i)} - \mu_Q\right)^2 \right] + \max_{Q \in \mathcal{U}} \frac{\sigma_Q^2}{n_Q} \right\}$$

$$= L^2 \left( \Sigma_T^2 + \sigma_T^2 \right)$$

It remains to bound $V_f$: recall that $0 = n_{(0)} \leq n_{(1)} \leq \cdots \leq n_{(k)}$ and $n_{(j)} = n_{Q_{(j)}}$, so that

$$V_f = \sum_{i=1}^{n} \mathbb{E}\left[ \max_{Q \in \mathcal{U}} \left[ f_Q\left(X_Q^{(i)}\right) - \mu_{f,i,Q} \right]^2 \right]$$

$$= \sum_{j=1}^{k} \left( n_{(j)} - n_{(j-1)} \right) \mathbb{E}\left[ \max_{r \in \{j,\ldots,k\}} \frac{1}{n_{(r)}^2} \left[ f\left(X_{Q_{(r)}}\right) - \mathbb{E}\left[f\left(X_{Q_{(r)}}\right)\right] \right]^2 \right]$$

With a similar symmetrization trick, we can further bound each expectation in the sum: let $X_Q'$ be an independent copy of $X_Q$. Then,

$$\mathbb{E}\left[ \max_{r \in \{j,\ldots,k\}} \frac{1}{n_{(r)}^2} \left[ f\left(X_{Q_{(r)}}\right) - \mathbb{E}\left[f\left(X_{Q_{(r)}}\right)\right] \right]^2 \right] = \mathbb{E}\left[ \max_{r \in \{j,\ldots,k\}} \frac{1}{n_{(r)}^2} \mathbb{E}\left[ f\left(X_{Q_{(r)}}\right) - f\left(X_{Q_{(r)}}'\right) \Big| X_{Q_{(r)}} \right]^2 \right]$$

$$\overset{(1)}{\leq} L^2 \mathbb{E}\left[ \max_{r \in \{j,\ldots,k\}} \frac{1}{n_{(r)}^2} \mathbb{E}\left[ \left(X_{Q_{(r)}} - X_{Q_{(r)}}'\right)^2 \Big| X_{Q_{(r)}} \right] \right]$$

$$\overset{(2)}{\leq} L^2 \mathbb{E}\left[ \max_{r \in \{j,\ldots,k\}} \frac{1}{n_{(r)}^2} \left\{ \left[ X_{Q_{(r)}} - \mu_{Q_{(r)}} \right]^2 + \sigma_{Q_{(r)}}^2 \right\} \right]$$

$$\leq 2L^2 \mathbb{E}\left[ \max_{r \in \{j,\ldots,k\}} \frac{1}{n_{(r)}^2} \left[ X_{Q_{(r)}} - \mu_{Q_{(r)}} \right]^2 \right]$$

where, in (1), we have applied Lipschitzness and Jensen's and, in (2), we note again that the cross term cancels when expanding the square. Hence, we get that

$$V_f \leq 2L^2 \sum_{j=1}^{k} \left( n_{(j)} - n_{(j-1)} \right) \mathbb{E}\left[ \max_{r \in \{j,\ldots,k\}} \frac{1}{n_{(r)}^2} \left[ X_{Q_{(r)}} - \mu_{Q_{(r)}} \right]^2 \right] = 2L^2 V_T$$

Plugging these values back into the bound (B.4) then yields the claim.

$$\blacksquare$$

## C  PROOFS OF SECTION 3

Recall our non-adaptive proxy objective

$$\mu_T^o(a) = \min_{Q \in \mathcal{U}} \frac{1}{n_Q} \sum_{i=1}^{n_Q} r\left(a, X_Q^{(i)}\right)$$

where, for UE, $n_Q = n$ for all $Q \in \mathcal{U}$. For $a \in \mathcal{A}$, define generalization gaps

$$D_a := \mu_{\mathrm{DR}}(a) - \mu_T^{\mathrm{o}}(a) = \min_{Q \in \mathcal{U}} \mu(a; Q) - \min_{Q' \in \mathcal{U}} \hat{\mu}_{n_{Q'}}(a; Q')$$

Using the same argument as in the proof of Proposition B.1, we note that

$$|D_a| \leq \max_{Q \in \mathcal{U}} \left| \mu(a; Q) - \hat{\mu}_{n_Q}(a; Q) \right| = \max_{Q \in \mathcal{U}} \left| \frac{1}{n_Q} \sum_{i=1}^{n_Q} \left[ \mathbb{E}\left[ r(a, X_Q) \right] - r\left( a, X_Q^{(i)} \right) \right] \right| =: U_a$$

Then from the theory of Appendix A, we can conclude the following bounds.

**Theorem C.1.** *For rewards bounded in* $[0, 1]$*, we have that for any* $a \in \mathcal{A}$*,*

$$\mathbb{E}[U_a] \leq 4\sqrt{\frac{\log k}{\min_{Q \in \mathcal{U}} n_Q}}$$

*Additionally, when* $\mathcal{X} \subset \mathbb{R}$ *and* $r(a, \cdot)$ *is L-Lipschitz for each* $a \in \mathcal{A}$*, it follows that*

$$\mathbb{E}[U_a] \leq \frac{16 \log k}{\min_{Q \in \mathcal{U}} n_Q} + 4L\sigma_T \sqrt{2 \log k}$$

*Proof.* We apply Theorem A.1 on functions $f_Q(x) := \mathbb{E}[r(a, X_Q)] - r(a, x)$. Note that $f_Q \in [-1, 1]$ since $r \in [0, 1]$. Moreover, if $r(a, \cdot)$ is $L$-Lipschitz, then so is $f_Q$, as we only add a constant to it. ∎

Let $\mathbb{E}[U_a] \leq B$ be any of the bounds from Theorem C.1. Then, we get that

$$\mathbb{E}[\mu_T^{\mathrm{o}}(a^*) - \mu_T^{\mathrm{o}}(a)] = \Delta_{\mathrm{DR}}(a) + \mathbb{E}[\mu_{\mathrm{DR}}(a) - \mu_T^{\mathrm{o}}(a)] - \mathbb{E}\left[\mu_{\mathrm{DR}}^* - \mu_T^{\mathrm{o}}(a^*)\right]$$

$$= \Delta_{\mathrm{DR}}(a) + \mathbb{E}[D_a] - \mathbb{E}[D_{a^*}]$$

$$\geq \Delta_{\mathrm{DR}}(a) - |\mathbb{E}[D_a]| - |\mathbb{E}[D_{a^*}]|$$

$$\geq \Delta_{\mathrm{DR}}(a) - \mathbb{E}[|D_a|] - \mathbb{E}[|D_{a^*}|]$$

$$\geq \Delta_{\mathrm{DR}}(a) - 2\mathbb{E}[U_a]$$

$$\geq \Delta_{\mathrm{DR}}(a) - 2B$$

for all $a \in \mathcal{A}$. Hence,

$$\mathbb{P}(A_T^{\mathrm{o}} = a) \leq \mathbb{P}(\mu_T^{\mathrm{o}}(a) \geq \mu_T^{\mathrm{o}}(a^*))$$

$$= \mathbb{P}(\mu_T^{\mathrm{o}}(a) - \mu_T^{\mathrm{o}}(a^*) - \mathbb{E}[\mu_T^{\mathrm{o}}(a) - \mu_T^{\mathrm{o}}(a^*)] \geq \mathbb{E}[\mu_T^{\mathrm{o}}(a^*) - \mu_T^{\mathrm{o}}(a)])$$

$$\leq \mathbb{P}(\mu_T^{\mathrm{o}}(a) - \mu_T^{\mathrm{o}}(a^*) - \mathbb{E}[\mu_T^{\mathrm{o}}(a) - \mu_T^{\mathrm{o}}(a^*)] \geq \Delta_{\mathrm{DR}}(a) - 2B) \quad \text{(C.1)}$$

What remains is to apply the concentration inequalities of Appendix B.

## C.1 Proof of Theorem 1

Here, we use the UE proxy $\mu_T^{\mathrm{o}}(a) = \min_{Q \in \mathcal{U}} \frac{1}{n} \sum_{i=1}^n r\left( a, X_Q^{(i)} \right)$. We can then obtain the following concentration inequality.

**Corollary C.1** (UE concentration inequality)**.** *We have that*

$$\mathbb{P}(\mu_T^o(a) - \mu_T^o(a') - \mathbb{E}[\mu_T^o(a) - \mu_T^o(a')] \geq t) \leq \exp\left(-\frac{nt^2}{2}\right)$$

*for all* $t \geq 0$ *and* $a, a' \in \mathcal{A}$*.*

*Proof.* Note that in the notation of Appendix B.1, $Z_{r(a, \cdot)} = \mu_T^{\mathrm{o}}(a)$. Since $r(a, \cdot) \in [0, 1]$ for each $a \in \mathcal{A}$, the claim follows by applying Corollary B.2. ∎

Next, note that under the assumption $n \geq \left(\frac{8}{\Delta_{\mathrm{DR,min}}}\right)^2 \log k$, we get that $\Delta_{\mathrm{DR}}(a) \geq 8\sqrt{\frac{\log k}{n}}$ for all $a \in \mathcal{A}$ with a positive gap. Hence, for all such $a$, plugging in the bound $B = 4\sqrt{\frac{\log k}{n}}$ into Equation (C.1) yields

$$\mathbb{P}(A_T^o = a) \leq \mathbb{P}\left(\mu_T^o(a) - \mu_T^o(a^*) - \mathbb{E}[\mu_T^o(a) - \mu_T^o(a^*)] \geq \Delta_{\mathrm{DR}}(a) - 8\sqrt{\frac{\log k}{n}}\right) \quad \text{Eq. (C.1)}$$

$$\leq \exp\left(-\frac{n}{2}\left[\Delta_{\mathrm{DR}}(a) - 8\sqrt{\frac{\log k}{n}}\right]^2\right) \qquad \text{Cor. C.1}$$

This directly yields the desired regret bound:

$$\mathbb{E}[\Delta_{\mathrm{DR}}(A_T^o)] = \sum_{a \in \mathcal{A} : \Delta_{\mathrm{DR}}(a) > 0} \Delta_{\mathrm{DR}}(a)\,\mathbb{P}(A_T^o = a)$$

$$\leq \sum_{a \in \mathcal{A} : \Delta_{\mathrm{DR}}(a) > 0} \Delta_{\mathrm{DR}}(a) \exp\left(-\frac{n}{2}\left[\Delta_{\mathrm{DR}}(a) - 8\sqrt{\frac{\log k}{n}}\right]^2\right)$$

Note that we can scale the rewards to instead operate under $r \in [0, M]$. This in turn yields the bound

$$\mathbb{E}[\Delta_{\mathrm{DR}}(A_T^o)] \leq \sum_{a \in \mathcal{A} : \Delta_{\mathrm{DR}}(a) > 0} \Delta_{\mathrm{DR}}(a) \exp\left(-\frac{n}{2M^2}\left[\Delta_{\mathrm{DR}}(a) - 8M\sqrt{\frac{\log k}{n}}\right]^2\right)$$

### C.2 PROOF OF COROLLARY 1

An alternative way of writing the UE regret bound is as follows:

$$\mathbb{E}[\Delta_{\mathrm{DR}}(A_T^o)] = \sum_{a \in \mathcal{A} : \Delta_{\mathrm{DR}}(a) \leq \Delta} \Delta_{\mathrm{DR}}(a)\,\mathbb{P}(A_T^o = a) + \sum_{a \in \mathcal{A} : \Delta_{\mathrm{DR}}(a) > \Delta} \Delta_{\mathrm{DR}}(a)\,\mathbb{P}(A_T^o = a)$$

$$\leq \Delta + \sum_{a \in \mathcal{A} : \Delta_{\mathrm{DR}}(a) > \Delta} \Delta_{\mathrm{DR}}(a) \exp\left(-\frac{n}{2}\left[\Delta_{\mathrm{DR}}(a) - 8\sqrt{\frac{\log k}{n}}\right]^2\right)$$

for any $\Delta \geq 0$. In other words,

$$\mathbb{E}[\Delta_{\mathrm{DR}}(A_T^o)] \leq \inf_{\Delta \geq 0}\left\{\Delta + \sum_{a \in \mathcal{A} : \Delta_{\mathrm{DR}}(a) > \Delta} \Delta_{\mathrm{DR}}(a) \exp\left(-\frac{n}{2}\left[\Delta_{\mathrm{DR}}(a) - 8\sqrt{\frac{\log k}{n}}\right]^2\right)\right\}$$
$$\text{(C.2)}$$

Next, we introduce a simple technical lemma.

**Lemma C.1.** *Let $\alpha, \beta > 0$. Then, the function $f(x) := x \exp\left(-\alpha(x-\beta)^2\right)$ is decreasing for $x \geq \frac{1}{2}\left(\beta + \sqrt{\beta^2 + \frac{2}{\alpha}}\right)$.*

*Proof.* Notice that

$$f'(x) = \exp\left(-\alpha(x-\beta)^2\right) - 2\alpha x(x-\beta)\exp\left(-\alpha(x-\beta)^2\right)$$

$$= [1 - 2\alpha x(x-\beta)]\exp\left(-\alpha(x-\beta)^2\right)$$

Now, note that the function $x \mapsto 2\alpha x(x-\beta) - 1$ is quadratic, convex and has roots $\frac{1}{2}\left(\beta + \sqrt{\beta^2 + \frac{2}{\alpha}}\right)$ and $\frac{1}{2}\left(\beta - \sqrt{\beta^2 + \frac{2}{\alpha}}\right)$. Since the former is larger, it follows that the quadratic is nonnegative for larger values. In other words, $f'(x) \leq 0$ whenever $x \geq \frac{1}{2}\left(\beta + \sqrt{\beta^2 + \frac{2}{\alpha}}\right)$. $\blacksquare$

As a result, we can show the following inequality.

**Lemma C.2.** *Provided that $l \geq 2$ and $\Delta_{\mathrm{DR}}(a) \geq \frac{8\sqrt{\log k} + \sqrt{2\log l}}{\sqrt{n}}$, we have that*

$$\Delta_{\mathrm{DR}}(a) \exp\left(-\frac{n}{2}\left[\Delta_{\mathrm{DR}}(a) - 8\sqrt{\frac{\log k}{n}}\right]^2\right) \leq \frac{8\sqrt{\log k} + \sqrt{2\log l}}{l\sqrt{n}}$$

*Proof.* Note that the left-hand side of the claim is of the form $f(\Delta_{\mathrm{DR}}(a))$, where $f$ is defined as in Lemma C.1 with $\alpha := \frac{n}{2}$ and $\beta := 8\sqrt{\frac{\log k}{n}}$, so that we know it is decreasing for $x \geq K$, where

$$
\begin{aligned}
K &:= \frac{1}{2}\left(\beta + \sqrt{\beta^2 + \frac{2}{\alpha}}\right) \\
&= \frac{1}{2}\left[8\sqrt{\frac{\log k}{n}} + \sqrt{\frac{64\log k}{n} + \frac{4}{n}}\right] \\
&= \frac{8\sqrt{\log k} + \sqrt{64\log k + 4}}{2\sqrt{n}} \\
&\leq \frac{8\sqrt{\log k} + 1}{\sqrt{n}} \qquad\qquad \sqrt{a+b} \leq \sqrt{a} + \sqrt{b} \\
&\leq \frac{8\sqrt{\log k} + \sqrt{2\log l}}{\sqrt{n}} \qquad\qquad \sqrt{2\log l} \geq 1
\end{aligned}
$$

The result then follows by plugging in $\frac{8\sqrt{\log k} + \sqrt{2\log l}}{\sqrt{n}}$ into $f$ to get the right-hand side of the claim. ∎

Finally, we can set $\Delta := \frac{8\sqrt{\log k} + \sqrt{2\log l}}{\sqrt{n}}$ in Equation (C.2) and apply Lemma C.2 to obtain

$$
\begin{aligned}
\mathbb{E}\left[\Delta_{\mathrm{DR}}(A_T^{\mathrm{o}})\right] &\leq \frac{8\sqrt{\log k} + \sqrt{2\log l}}{\sqrt{n}} + \left|\{a \in \mathcal{A} : \Delta_{\mathrm{DR}}(a) > \Delta\}\right| \frac{8\sqrt{\log k} + \sqrt{2\log l}}{l\sqrt{n}} \\
&\leq \frac{16\sqrt{\log k} + 2\sqrt{2\log l}}{\sqrt{n}} \\
&\lesssim \sqrt{\frac{\log(kl)}{n}}
\end{aligned}
$$

where in the last line we have used the fact that $\sqrt{a} + \sqrt{b} \leq \sqrt{2(a+b)}$. Substituting $n = \frac{T}{k}$ then yields the result.

## C.3 PROOF OF THEOREM 2

Returning to the general NUE proxy $\mu_T^{\mathrm{o}}(a) = \min_{Q \in \mathcal{U}} \frac{1}{n_Q} \sum_{i=1}^{n_Q} r\left(a, X_Q^{(i)}\right)$, let us further assume that $\mathcal{X} \subset \mathbb{R}$. Then, we conclude the following result.

**Corollary C.2** (NUE concentration inequality). *Suppose that $r(a, \cdot) : \mathcal{X} \to [0, M]$ is $L$-Lipschitz for each $a \in \mathcal{A}$. Then, we have that*

$$\mathbb{P}\left(\mu_T^o(a) - \mu_T^o(a') - \mathbb{E}\left[\mu_T^o(a) - \mu_T^o(a')\right] \geq t\right) \leq \exp\left(-\frac{t^2}{16L^2\left(2\sigma_T^2 + \Sigma_T^2 + 6V_T\right) + \frac{2\sqrt{6}Mt}{\min_{Q \in \mathcal{U}} n_Q}}\right)$$

*for all $t \geq 0$ and $a, a' \in \mathcal{A}$.*

*Proof.* Once again, using the definitions of Appendix B, we get that $Z_{r(a,\cdot)} = \mu_T^{\mathrm{o}}(a)$. Since $r(a, \cdot) \in [0, M]$ is $L$-Lipschitz for each $a \in \mathcal{A}$, the claim follows by applying Theorem B.1. ∎

Note that we can scale all quantities in Theorem C.1 by $M$ to work with rewards in $[0, M]$ instead of $[0, 1]$. Then, as in the UE analysis, provided that $\Delta_{\mathrm{DR,min}} \geq G_T = 8M \left( \frac{4 \log k}{\min_{Q \in \mathcal{U}} n_Q} + L\sigma_T \sqrt{2 \log k} \right)$, we can plug $B = \frac{16 \log k}{\min_{Q \in \mathcal{U}} n_Q} + 4L\sigma_T \sqrt{2 \log k}$ into Equation (C.1) to conclude that

$$\mathbb{P}\left(A_T^{\mathrm{o}} = a\right) \leq \mathbb{P}\left(\mu_T^{\mathrm{o}}(a) - \mu_T^{\mathrm{o}}(a^*) - \mathbb{E}\left[\mu_T^{\mathrm{o}}(a) - \mu_T^{\mathrm{o}}(a^*)\right] \geq \Delta_{\mathrm{DR}}(a) - G_T\right) \qquad \text{Eq. (C.1)}$$

$$\leq \exp\left(-\frac{\left[\Delta_{\mathrm{DR}}(a) - G_T\right]^2}{16L^2\left(2\sigma_T^2 + \Sigma_T^2 + 6V_T\right) + \frac{2\sqrt{6}M}{\min_{Q \in \mathcal{U}} n_Q}\left[\Delta_{\mathrm{DR}}(a) - G_T\right]}\right) \quad \text{Cor. C.1}$$

for all $a \in \mathcal{A}$ with positive gap. This in turn yields the regret bound

$$\mathbb{E}\left[\Delta_{\mathrm{DR}}\left(A_T^{\mathrm{o}}\right)\right] = \sum_{a \in \mathcal{A}: \Delta_{\mathrm{DR}}(a) > 0} \Delta_{\mathrm{DR}}(a)\,\mathbb{P}\left(A_T^{\mathrm{o}} = a\right)$$

$$\leq \sum_{a \in \mathcal{A}: \Delta_{\mathrm{DR}}(a) > 0} \Delta_{\mathrm{DR}}(a) \exp\left(-\frac{\left[\Delta_{\mathrm{DR}}(a) - G_T\right]^2}{16L^2\left(2\sigma_T^2 + \Sigma_T^2 + 6V_T\right) + \frac{2\sqrt{6}M}{\min_{Q \in \mathcal{U}} n_Q}\left[\Delta_{\mathrm{DR}}(a) - G_T\right]}\right)$$

## D  MODIFIED UCB-E

Our goal is to perform a minimization variant of UCB-E Audibert et al. (2010) for $T$ rounds on the set of "arms" $\mathcal{U}$. Since we will analyze all random variables under a fixed high-probability event, we treat all quantities here as deterministic. In particular, we work with $\mu(Q), \hat{\mu}_t(Q) \in [0, 1]$ for each $Q \in \mathcal{U}$ and $t \in \{n_0(Q), \ldots, n_0(Q) + T\}$, where $n_0(Q) \geq 1$ is the number of pulls from arm $Q \in \mathcal{U}$ that we start the game with. We assume a unique optimal arm $Q^* := \mathrm{argmin}_{Q \in \mathcal{U}} \mu(Q)$, with $\mu^* := \mu(Q^*)$, and define suboptimality gaps $\Delta(Q) := \mu(Q) - \mu^*$ and $\Delta_{\min} := \min_{Q \in \mathcal{U} \setminus \{Q^*\}} \Delta(Q)$. For some choice of plays $\{Q_t\}_{t=1}^T$, let

$$n_t(Q) := n_0(Q) + \sum_{s=1}^t \mathbb{I}\{Q_s = Q\}$$

denote the number of times distribution $Q$ has been played at time $t \in [T]$. Additionally, we define the following subset of arms:

$$\mathcal{U}_0 := \left\{Q \in \mathcal{U} \setminus \{Q^*\} : n_0(Q) < \frac{36}{25}\epsilon\Delta^{-2}(Q)\right\} \cup \left\{Q^* : n_0(Q^*) < \frac{36}{25}\epsilon\Delta_{\min}^{-2}\right\}$$

along with its cardinality (provided that it contains $Q^*$) $k_0 := |\mathcal{U}_0| \,\mathbb{I}\{Q^* \in \mathcal{U}_0\}$, total initial sample size $\tilde{T}_0 := \sum_{Q \in \mathcal{U}_0} n_0(Q)$ and the complexity notion it defines:

$$H_0 := \Delta_{\min}^{-2}\mathbb{I}\{Q^* \in \mathcal{U}_0\} + \sum_{Q \in \mathcal{U}_0 \setminus \{Q^*\}} \Delta^{-2}(Q)$$

The intuition is that $\mathcal{U}_0$ is a proxy for the set of arms played:

$$\mathcal{U}' := \{Q \in \mathcal{U} : n_T(Q) > n_0(Q)\}$$

The UCB-E algorithm works by defining indices (adjusted here for lower confidence bounds)

$$\mathrm{LCB}_t(Q; \epsilon) := \hat{\mu}_{n_t(Q)}(Q) - \sqrt{\frac{\epsilon}{n_t(Q)}} \quad \forall Q \in \mathcal{U}$$

given a parameter $\epsilon > 0$ and, at each time step $t \in [T]$, playing

$$Q_t := \mathrm{argmin}_{Q \in \mathcal{U}} \mathrm{LCB}_{t-1}(Q; \epsilon)$$

After $T$ rounds, we output

$$\hat{Q} := \mathrm{argmin}_{Q \in \mathcal{U}} \hat{\mu}_{n_T(Q)}(Q)$$

**Theorem D.1** (Modified UCB-E optimality). *Suppose that*

$$|\mu(Q) - \hat{\mu}_t(Q)| < \frac{1}{5}\sqrt{\frac{\epsilon}{t}}$$

*for all $Q \in \mathcal{U}$ and $t \in \{n_0(Q), \ldots, n_0(Q) + T\}$, and that*

$$\epsilon \geq \frac{25}{36}\Delta_{min}^2 [n_0(Q^*) - 1]$$

$$T \geq \frac{36}{25}\epsilon H_0 - \tilde{T}_0 + k_0$$

*Then, it follows that $\hat{Q} = Q^*$ and*

$$\frac{1}{5}\sqrt{\frac{\epsilon}{n_T(Q^*)}} \leq \frac{\Delta_{min}}{2}$$

*Proof.* First, notice that for any $t \in \{0, \ldots, T\}$ and $Q \in \mathcal{U}$, we have by assumption that

$$\left|\mu(Q) - \hat{\mu}_{n_t(Q)}(Q)\right| < \frac{1}{5}\sqrt{\frac{\epsilon}{n_t(Q)}} \tag{D.1}$$

since $n_t(Q) \in \{n_0(Q), \ldots, n_0(Q) + T\}$. All we need to do is show that, for any $Q \in \mathcal{U} \backslash \{Q^*\}$,

$$n_T(Q) \geq \frac{4}{25}\epsilon\Delta^{-2}(Q) \quad \text{and} \quad n_T(Q^*) \geq \frac{4}{25}\epsilon\Delta_{min}^{-2} \tag{D.2}$$

since this implies that

$$\frac{1}{5}\sqrt{\frac{\epsilon}{n_T(Q)}} \leq \frac{\Delta(Q)}{2} \quad \text{and} \quad \frac{1}{5}\sqrt{\frac{\epsilon}{n_T(Q^*)}} \leq \frac{\Delta_{min}}{2} \leq \frac{\Delta(Q)}{2}$$

The second inequality is one of our desired results. To obtain the other, we observe that

$$\hat{\mu}_{n_T(Q)}(Q) - \hat{\mu}_{n_T(Q^*)}(Q^*) = \hat{\mu}_{n_T(Q)}(Q) - \mu(Q) + \Delta(Q) + \mu^* - \hat{\mu}_{n_T(Q^*)}(Q^*)$$

$$> \Delta(Q) - \frac{1}{5}\sqrt{\frac{\epsilon}{n_T(Q)}} - \frac{1}{5}\sqrt{\frac{\epsilon}{n_T(Q^*)}} \qquad \text{Eq. (D.1)}$$

$$\geq \Delta(Q) - \frac{\Delta(Q)}{2} - \frac{\Delta(Q)}{2}$$

$$= 0$$

Since this holds for all $Q \in \mathcal{U} \backslash \{Q^*\}$, it follows that $\hat{Q} = Q^*$. To show (D.2), we break into two cases.

## D.1 CASE 1: $Q^* \notin \mathcal{U}_0$

First, suppose that $Q^* \notin \mathcal{U}_0$ and note that

$$n_T(Q) \geq \frac{36}{25}\epsilon\Delta^{-2}(Q) > \frac{4}{25}\epsilon\Delta^{-2}(Q) \quad \text{and} \quad n_T(Q^*) \geq n_0(Q^*) \geq \frac{36}{25}\epsilon\Delta_{min}^{-2} > \frac{4}{25}\epsilon\Delta_{min}^{-2}$$

for any $Q \notin \mathcal{U}_0 \cup \{Q^*\}$ by definition. To show the first inequality for $\mathcal{U}_0$, we observe that $k_0 = 0$ and $H_0 = \sum_{Q \in \mathcal{U}_0} \Delta^{-2}(Q)$ and make the following claim, that applies in both cases.

**Lemma D.1.** *Fix $t \in [T]$. If $Q_t = Q \neq Q^*$, then*

$$n_{t-1}(Q) < \frac{36}{25}\epsilon\Delta^{-2}(Q)$$

*Proof.* We have that

$$\mu^* > \hat{\mu}_{n_{t-1}(Q^*)}(Q^*) - \frac{1}{5}\sqrt{\frac{\epsilon}{n_{t-1}(Q^*)}} \qquad \text{Eq. (D.1)}$$

$$\geq \text{LCB}_{t-1}(Q^*; \epsilon)$$
$$\geq \text{LCB}_{t-1}(Q; \epsilon) \qquad\qquad Q_t = Q$$

$$= \hat{\mu}_{n_{t-1}(Q)}(Q) - \sqrt{\frac{\epsilon}{n_{t-1}(Q)}}$$

$$> \mu(Q) - \frac{6}{5}\sqrt{\frac{\epsilon}{n_{t-1}(Q)}} \qquad \text{Eq. (D.1)}$$

Rearranging then yields the claim. $\blacksquare$

In other words, once $n_t(Q) \geq \frac{36}{25}\epsilon\Delta^{-2}(Q)$, arm $Q \neq Q^*$ will no longer be played after round $t$. This means that any arm outside of $\mathcal{U}_0 \cup \{Q^*\}$ will not be played at all. In addition, if $Q^*$ is not played in the first

$$T' := \sum_{Q \in \mathcal{U}_0} \left[ \frac{36}{25}\epsilon\Delta^{-2}(Q) - n_0(Q) \right] = \frac{36}{25}\epsilon H_0 - \tilde{T}_0 + k_0$$

rounds, then the plays will distributed within $\mathcal{U}_0$, resulting in

$$n_T(Q) \geq n_{T'}(Q) = \frac{36}{25}\epsilon\Delta^{-2}(Q) > \frac{4}{25}\epsilon\Delta^{-2}(Q) \quad \forall Q \in \mathcal{U}_0$$

where the first inequality uses the assumption that $T \geq T'$. When $Q^*$ is played, we get the following result.

**Proposition D.1.** *Suppose that $Q^*$ is played in some round. Then,*

$$n_T(Q) \geq \frac{4}{25}\epsilon\Delta^{-2}(Q) \quad \forall Q \in \mathcal{U}_0$$

*Proof.* Let $Q \in \mathcal{U}_0$ and let $t \in [T]$ be any round such that $Q_t = Q^*$. Then,

$$\mu(Q) - \frac{4}{5}\sqrt{\frac{\epsilon}{n_T(Q)}} \geq \mu(Q) - \frac{4}{5}\sqrt{\frac{\epsilon}{n_{t-1}(Q)}}$$

$$> \text{LCB}_{t-1}(Q; \epsilon) \qquad\qquad \text{Eq. (D.1)}$$
$$\geq \text{LCB}_{t-1}(Q^*; \epsilon)$$

$$> \mu^* - \frac{6}{5}\sqrt{\frac{\epsilon}{n_{t-1}(Q^*)}} \qquad\qquad \text{Eq. (D.1)}$$

$$\geq \mu^* - \frac{6}{5}\sqrt{\frac{\epsilon}{n_0(Q^*)}}$$

$$\geq \mu^* - \Delta_{\min} \qquad\qquad n_0(Q^*) \geq \frac{36}{25}\epsilon\Delta_{\min}^{-2}$$

$$\geq \mu^* - \Delta(Q)$$

The claim then follows by rearranging the terms. $\blacksquare$

## D.2 CASE 2: $Q^* \in \mathcal{U}_0$

Next, we note that

$$k_0 = |\mathcal{U}_0| \quad \text{and} \quad H_0 = \Delta_{\min}^{-2} + \sum_{Q \in \mathcal{U}_0 \setminus \{Q^*\}} \Delta^{-2}(Q)$$

As a direct consequence of Lemma D.1, we can conclude that our proxy set $\mathcal{U}_0$ indeed contains the arms played.

**Corollary D.1.** $\mathcal{U}' \subset \mathcal{U}_0$.

*Proof.* Fix $Q \in \mathcal{U}' \setminus \{Q^*\}$ and let $t \in [T]$ denote any round in which $Q_t = Q$. From Lemma D.1 we then get that $n_0(Q) \leq n_{t-1}(Q) < \frac{36}{25}\epsilon\Delta^{-2}(Q)$. ∎

Next, we show that suboptimal arms in the proxy set do not have too many samples by the end of the procedure.

**Proposition D.2.**

$$n_T(Q) < \frac{36}{25}\epsilon\Delta^{-2}(Q) + 1 \quad \forall Q \in \mathcal{U}_0 \setminus \{Q^*\}$$

*Proof.* If $Q \in \mathcal{U}_0 \setminus (\mathcal{U}' \cup \{Q^*\})$, then

$$n_T(Q) = n_0(Q) < \frac{36}{25}\epsilon\Delta^{-2}(Q) < \frac{36}{25}\epsilon\Delta^{-2}(Q) + 1$$

Otherwise, fix any $Q \in \mathcal{U}' \setminus \{Q^*\}$ and let $t \in [T]$ be the largest time step such that $Q_t = Q$ (i.e., the last round in which $Q$ is played). Lemma D.1 then implies that

$$n_T(Q) = n_{T-1}(Q) = \cdots = n_t(Q) = n_{t-1}(Q) + 1 < \frac{36}{25}\epsilon\Delta^{-2}(Q) + 1$$

∎

This, in turn, implies that the optimal arm has sufficiently many samples and, in fact, is in $\mathcal{U}'$.

**Proposition D.3.**

$$n_T(Q^*) > \frac{36}{25}\epsilon\Delta_{min}^{-2} + 1$$

*Proof.* We have that

$$
\begin{aligned}
n_T(Q^*) &= T + n_0(Q^*) - \sum_{Q \in \mathcal{U}' \setminus \{Q^*\}} [n_T(Q) - n_0(Q)] \\
&= T + n_0(Q^*) - \sum_{Q \in \mathcal{U}_0 \setminus \{Q^*\}} [n_T(Q) - n_0(Q)] && \text{Cor. D.1} \\
&= T + \tilde{T}_0 - \sum_{Q \in \mathcal{U}_0 \setminus \{Q^*\}} n_T(Q) \\
&> T + \tilde{T}_0 - \sum_{Q \in \mathcal{U}_0 \setminus \{Q^*\}} \left[\frac{36}{25}\epsilon\Delta^{-2}(Q) + 1\right] && \text{Prop. D.2} \\
&= T + \tilde{T}_0 - \frac{36}{25}\epsilon\left(H_0 - \Delta_{\min}^{-2}\right) - k_0 + 1 \\
&\geq \frac{36}{25}\epsilon\Delta_{\min}^{-2} + 1
\end{aligned}
$$

where the last line follows from our lower bound assumption on $T$. ∎

**Corollary D.2.** *We have that $Q^* \in \mathcal{U}'$.*

*Proof.* This immediately follows from Proposition D.3 and our lower bound assumption on $\epsilon$:

$$n_T(Q^*) > \frac{36}{25}\epsilon\Delta_{\min}^{-2} + 1 \geq n_0(Q^*)$$

∎

We are then able to show that, by the end of the game, every arm has sufficiently many samples.

**Proposition D.4.**

$$n_T(Q) \geq \frac{4}{25}\epsilon\Delta^{-2}(Q) \quad \forall Q \in \mathcal{U}\backslash\{Q^*\}$$

*Proof.* Let $Q \in \mathcal{U}\backslash\{Q^*\}$. Since $Q^* \in \mathcal{U}'$ by Corollary D.2, let $t \in [T]$ be the last round such that $Q_t = Q^*$. Then,

$$
\begin{aligned}
\mu(Q) - \frac{4}{5}\sqrt{\frac{\epsilon}{n_T(Q)}} &\geq \mu(Q) - \frac{4}{5}\sqrt{\frac{\epsilon}{n_{t-1}(Q)}} \\
&> \mathrm{LCB}_{t-1}(Q;\epsilon) && \text{Eq. (D.1)} \\
&\geq \mathrm{LCB}_{t-1}(Q^*;\epsilon) \\
&> \mu^* - \frac{6}{5}\sqrt{\frac{\epsilon}{n_{t-1}(Q^*)}} && \text{Eq. (D.1)} \\
&= \mu^* - \frac{6}{5}\sqrt{\frac{\epsilon}{n_T(Q^*)-1}} && n_T(Q^*) = n_t(Q^*) = n_{t-1}(Q^*)+1 \\
&> \mu^* - \Delta(Q) && \text{Prop. D.3 and } \Delta_{\min} \leq \Delta(Q)
\end{aligned}
$$

The claim then follows by rearranging the terms. ∎

Let $Q \in \mathcal{U}\backslash\{Q^*\}$. From Propositions D.3 and D.4, we can thus conclude inequalities (D.2)

$$n_T(Q) \geq \frac{4}{25}\epsilon\Delta^{-2}(Q) \quad \text{and} \quad n_T(Q^*) \geq \frac{36}{25}\epsilon\Delta_{\min}^{-2} + 1 > \frac{4}{25}\epsilon\Delta_{\min}^{-2}$$

∎

# E  PROOF OF THEOREM 3

Suppose that we are operating under permutation $(a_1, \ldots, a_l)$ and parameters $(\epsilon_1, \ldots, \epsilon_l)$ satisfying the bound (1). To show our desired result, we will define a high-probability event, under which the modified UCB-E analysis ensures the correctness of LCB-DR's decision.

## E.1  CONCENTRATION INEQUALITY

From the boundedness of $r \in [0,1]$, Hoeffding's inequality implies that

$$\mathbb{P}\left(|\mu(a;Q) - \hat{\mu}_t(a;Q)| < \frac{1}{5}\sqrt{\frac{\epsilon}{t}}\right) \geq 1 - 2\exp\left(-\frac{2\epsilon}{25}\right)$$

for all $a \in \mathcal{A}, Q \in \mathcal{U}, t \in \mathbb{N}$ and $\epsilon \geq 0$. Fix some $j \in [l]$. Then, taking union bounds yields

$$\mathbb{P}\left(\bigcap_{Q \in \mathcal{U}} \bigcap_{t \in [u_j]} \left\{|\mu(a_j;Q) - \hat{\mu}_t(a_j;Q)| < \frac{C_{a_j} \wedge 1}{5}\sqrt{\frac{\epsilon_j}{t}}\right\}\right)$$
$$\geq 1 - 2ku_j\exp\left(-\frac{2\left(C_{a_j}^2 \wedge 1\right)\epsilon_j}{25}\right)$$

We then define the high-probability event of interest:

$$A_j := \bigcap_{Q \in \mathcal{U}} \bigcap_{t \in [u_j]} \left\{|\mu(a_j;Q) - \hat{\mu}_t(a_j;Q)| < \frac{C_{a_j} \wedge 1}{5}\sqrt{\frac{\epsilon_j}{t}}\right\}$$

### E.2 Modified UCB-E analysis

Here, we apply the UCB-E analysis of Appendix D. Note that

$$\bar{T}_j = \sum_{r=0}^{j} T_r = k + \sum_{r=1}^{j} \left[ \frac{36}{25} \epsilon_r \underbrace{H_r}_{\leq 2H_{a_r}} \underbrace{-\tilde{T}_r}_{\leq 0} + \underbrace{k_r}_{\leq k} \right] \leq u_j$$

Hence, $n_{\bar{T}_{j-1}}(Q) + T_j \leq \bar{T}_j \leq u_j$, for any $Q \in \mathcal{U}$ and, thus, under event $A_j$,

$$|\mu(a_j; Q) - \hat{\mu}_t(a_j; Q)| < \frac{C_{a_j} \wedge 1}{5} \sqrt{\frac{\epsilon_j}{t}} \leq \frac{1}{5} \sqrt{\frac{\epsilon_j}{t}}$$

for all $Q \in \mathcal{U}$ and $t \in \left\{ n_{\bar{T}_{j-1}}(Q), \ldots, n_{\bar{T}_{j-1}}(Q) + T_j \right\}$. Moreover, since $\bar{T}_0 = u_0$, we have from the lower bound (1) on $(\epsilon_1, \ldots, \epsilon_l)$ that

$$\epsilon_j \geq \frac{25}{36} \Delta_{a_j,\min}^2 (u_{j-1} - 1) \geq \frac{25}{36} \Delta_{a_j,\min}^2 (\bar{T}_{j-1} - 1) \geq \frac{25}{36} \Delta_{a_j,\min}^2 \left( n_{\bar{T}_{j-1}} \left( Q_{a_j}^* \right) - 1 \right)$$

for all $j \in [l]$. We can then conclude the following result.

**Theorem E.1.** *For any $j \in [l]$, under event $A_j$, it follows that $\hat{Q}_j = Q_{a_j}^*$ and*

$$|\mu_{\mathrm{DR}}(a_j) - \mu_T^o(a_j)| < \begin{cases} \dfrac{\Delta_{\mathrm{DR}}(a_j)}{2} & a_j \neq a^* \\ \dfrac{\Delta_{\mathrm{DR,min}}}{2} & a_j = a^* \end{cases}$$

*Proof.* If we set $T = T_j, \epsilon = \epsilon_j, n_0 = n_{\bar{T}_{j-1}}, \mu = \mu(a_j; \cdot)$ and $\hat{\mu}_t = \hat{\mu}_t(a_j; \cdot)$ in the setup of Appendix D, then we can immediately see that $\hat{Q}_j = Q_{a_j}^*$ by Theorem D.1, as its assumptions are satisfied under $A_j$. Moreover, we have that

$$
\begin{aligned}
|\mu_{\mathrm{DR}}(a_j) - \mu_T^o(a_j)| &= \left| \mu\left(a_j, Q_{a_j}^*\right) - \hat{\mu}_{n_{\bar{T}_j}\left(Q_{a_j}^*\right)}\left(a_j, Q_{a_j}^*\right) \right| & \hat{Q}_j = Q_{a_j}^* \\
&< \frac{C_{a_j} \wedge 1}{5} \sqrt{\frac{\epsilon_j}{n_{\bar{T}_j}\left(Q_{a_j}^*\right)}} & \text{event } A_j \text{ and } n_{\bar{T}_j} \leq \bar{T}_j \leq u_j \\
&\leq C_{a_j} \frac{\Delta_{a_j,\min}}{2} & \text{Thm. D.1} \\
&= \begin{cases} \dfrac{\Delta_{\mathrm{DR}}(a_j)}{2} & a_j \neq a^* \\ \dfrac{\Delta_{\mathrm{DR,min}}}{2} & a_j = a^* \end{cases}
\end{aligned}
$$

∎

### E.3 LCB-DR correctness

Under the event $\bigcap_{j=1}^{l} A_j$, we know that

$$
\begin{aligned}
\mu_T^o(a^*) - \mu_T^o(a) &= \mu_T^o(a^*) - \mu_{\mathrm{DR}}^* + \Delta_{\mathrm{DR}}(a) + \mu_{\mathrm{DR}}(a) - \mu_T^o(a) \\
&> \Delta_{\mathrm{DR}}(a) - \frac{\Delta_{\mathrm{DR,min}}}{2} - \frac{\Delta_{\mathrm{DR}}(a)}{2} & \text{Thm. E.1} \\
&\geq 0 & \Delta_{\mathrm{DR,min}} \leq \Delta_{\mathrm{DR}}(a)
\end{aligned}
$$

for every $a \neq a^*$. That is, $A_T^o = \operatorname{argmax}_{a \in \mathcal{A}} \mu_T^o(a) = a^*$ and, thus, $\mathbb{P}(A_T^o = a^*) \geq \mathbb{P}\left(\bigcap_{j=1}^{l} A_j\right)$. The result then follows from a union bound on the high-probability events $\{A_j\}_{j=1}^{l}$.

# F EXTENDING TO INFINITE DECISION SETS

Let $\bar{Q} := \frac{1}{k}\sum_{Q \in \mathcal{U}} Q$ be the uniform mixture and suppose that we have access to a finite $\frac{\epsilon}{k}$-cover $\mathcal{A}_\epsilon$ of $\left(\{r(a, \cdot)\}_{a \in \mathcal{A}}, L^2(\bar{Q})\right)$ in the following sense: for all $a \in \mathcal{A}$, there exists a $\phi_a \in \mathcal{A}_\epsilon$ such that

$$\|r(a, \cdot) - r(\phi_a, \cdot)\|_{L^2(\bar{Q})} := \sqrt{\mathbb{E}_{X \sim \bar{Q}}\left[(r(a, X) - r(\phi_a, X))^2\right]} \leq \frac{\epsilon}{k}$$

The idea is that a learner can play the game dynamics on the finite set $\mathcal{A}_\epsilon$ to control the gap $\Delta_{\mathrm{DR}}(\cdot; \mathcal{A}_\epsilon)$, where we made the underlying decision set explicit in the notation, and this ensures control of the original objective. We can relate this gap to the quantity of interest by noting that for any $a \in \mathcal{A}$,

$$\Delta_{\mathrm{DR}}(a; \mathcal{A}) = \max_{a^* \in \mathcal{A}} \mu_{\mathrm{DR}}(a^*) - \mu_{\mathrm{DR}}(a)$$

$$= \max_{a^* \in \mathcal{A}} \mu_{\mathrm{DR}}(a^*) - \max_{a_\epsilon^* \in \mathcal{A}_\epsilon} \mu_{\mathrm{DR}}(a_\epsilon^*) + \Delta_{\mathrm{DR}}(a; \mathcal{A}_\epsilon)$$

$$= \max_{a^* \in \mathcal{A}} \left\{\mu_{\mathrm{DR}}(a^*) - \max_{a_\epsilon^* \in \mathcal{A}_\epsilon} \mu_{\mathrm{DR}}(a_\epsilon^*)\right\} + \Delta_{\mathrm{DR}}(a; \mathcal{A}_\epsilon)$$

We can bound the error term as follows: for any $a \in \mathcal{A}$,

$$\mu_{\mathrm{DR}}(a) - \max_{a_\epsilon^* \in \mathcal{A}_\epsilon} \mu_{\mathrm{DR}}(a_\epsilon^*) \leq \mu_{\mathrm{DR}}(a) - \mu_{\mathrm{DR}}(\phi_a)$$

$$= \min_{Q \in \mathcal{U}} \mathbb{E}_{X \sim Q}[r(a, X)] - \min_{Q \in \mathcal{U}} \mathbb{E}_{X \sim Q}[r(\phi_a, X)]$$

$$\leq \max_{Q \in \mathcal{U}} \mathbb{E}_{X \sim Q}[r(a, X) - r(\phi_a, X)]$$

$$\leq \sum_{Q \in \mathcal{U}} \mathbb{E}_{X \sim Q}[r(a, X) - r(\phi_a, X)]$$

$$= k \underbrace{\mathbb{E}_{X \sim \bar{Q}}[r(a, X) - r(\phi_a, X)]}_{(*)}$$

$$\leq k \|r(a, \cdot) - r(\phi_a, \cdot)\|_{L^2(\bar{Q})}$$

$$\leq \epsilon$$

That is,

$$\Delta_{\mathrm{DR}}(\cdot; \mathcal{A}) \leq \Delta_{\mathrm{DR}}(\cdot; \mathcal{A}_\epsilon) + \epsilon$$

## F.1 BINARY CLASSIFICATION

A special case is the binary classification setting:

- The data are pairs $(X, Y) \in \mathcal{X} \times \{0, 1\}$.
- Decisions are binary-valued functions $a: \mathcal{X} \to \{0, 1\}$ and $\mathrm{VC}(\mathcal{A}) = d < \infty$.
- The reward function is $r(a, (x, y)) = \mathbb{I}\{a(x) = y\}$, so that
$$\mathbb{E}_{(X,Y) \sim Q}[r(a, (X, Y))] = \mathbb{P}_{(X,Y) \sim Q}(a(X) = Y)$$

Suppose that we have a finite $\sqrt{\frac{\epsilon}{k}}$-cover $\mathcal{A}_\epsilon$ of $(\mathcal{A}, L^2(\bar{Q}_\mathcal{X}))$, where $\bar{Q}_\mathcal{X}$ is the marginal distribution of $\bar{Q}$ over $\mathcal{X}$ (recall that now the $Q$'s are distributions over pairs $(X, Y) \in \mathcal{X} \times \{0, 1\}$): for any $a \in \mathcal{A}$, there exists a $\phi_a \in \mathcal{A}_\epsilon$ such that

$$\|a - \phi_a\|_{L^2(\bar{Q}_\mathcal{X})} := \sqrt{\mathbb{E}_{X \sim \bar{Q}_\mathcal{X}}\left[(a(X) - \phi_a(X))^2\right]} \leq \sqrt{\frac{\epsilon}{k}}$$

From Dudley (see e.g. (van Handel, 2014, Theorem 7.16)), we know that there exists such a cover of size

$$|\mathcal{A}_\epsilon| \lesssim \left(\frac{k}{\epsilon}\right)^{Cd}$$

for some universal constant $C$. Then from the more general derivation, note that

$$(*) = \mathbb{E}_{(X,Y)\sim\bar{Q}} \left[ \mathbb{I}\{a(X) = Y\} - \mathbb{I}\{\phi_a(X) = Y\} \right]$$

$$= \mathbb{E}_{(X,Y)\sim\bar{Q}} \left[ \mathbb{I}\{\phi_a(X) \neq Y\} - \mathbb{I}\{a(X) \neq Y\} \right]$$

$$= \mathbb{E}_{(X,Y)\sim\bar{Q}} \left[ (\phi_a(X) - Y)^2 - (a(X) - Y)^2 \right]$$

$$= \mathbb{E}_{(X,Y)\sim\bar{Q}} \left[ (\phi_a(X) - a(X))(\phi_a(X) + a(X) - 2Y) \right]$$

$$= \mathbb{E}_{(X,Y)\sim\bar{Q}} \left[ (\phi_a(X) - a(X))^2 + 2(\phi_a(X) - a(X))(a(X) - Y) \right]$$

$$\overset{(1)}{\leq} \|a - \phi_a\|^2_{L^2(\bar{Q}_{\mathcal{X}})}$$

$$\leq \frac{\epsilon}{k}$$

where for $(1)$, we note that

- If $a(X) = 1$, then $\phi_a(X) - a(X) \leq 0$ and $a(X) - Y \geq 0$.
- If $a(X) = 0$, then $\phi_a(X) - a(X) \geq 0$ and $a(X) - Y \leq 0$.

In both cases $(\phi_a(X) - a(X))(a(X) - Y) \leq 0$.

For example, if we use the distribution-independent regret of Corollary 1, this shows that the output $A^{\mathrm{o}}_T$ of UE on $\mathcal{A}_\epsilon$ guarantees

$$\mathbb{E}[\Delta_{\mathrm{DR}}(A^{\mathrm{o}};\mathcal{A})] \leq \mathbb{E}[\Delta_{\mathrm{DR}}(A^{\mathrm{o}};\mathcal{A}_\epsilon)] + \epsilon$$

$$\lesssim \sqrt{\frac{k\log(k|\mathcal{A}_\epsilon|)}{T}} + \epsilon$$

$$\lesssim \sqrt{\frac{k\left(\log k + d\log\frac{k}{\epsilon}\right)}{T}} + \epsilon$$

$$\lesssim \sqrt{\frac{k\left(\log k + d\log(kT)\right)}{T}}$$

where we chose $\epsilon = \frac{1}{\sqrt{T}}$ for the last line.

## G   UE v.s. NUE

Here, we will prove the bounds stated in Section 3.4. For convenience, we present the variance quantities again below:

$$V_T = \sum_{j=1}^{k} \left(n_{(j)} - n_{(j-1)}\right) \mathbb{E}\left[\max_{r\in\{j,\dots,k\}} \frac{1}{n_{(r)}^2} \left[X_{Q_{(r)}} - \mu_{Q_{(r)}}\right]^2\right]$$

$$\Sigma_T^2 = \mathbb{E}\left[\max_{Q\in\mathcal{U}} \frac{1}{n_Q^2} \sum_{i=1}^{n_Q} \left(X_Q^{(i)} - \mu_Q\right)^2\right]$$

$$\sigma_T^2 = \max_{Q\in\mathcal{U}} \frac{\sigma_Q^2}{n_Q}$$

We begin by proving the bound on $\Sigma_T^2$.

**Lemma G.1.** *Suppose that our data is bounded: $X_Q \in [0,1]$. Then,*

$$\Sigma_T^2 \leq 8\sqrt{\frac{2\log(2k)}{\min_{Q\in\mathcal{U}} n_Q^3}} + \sigma_T^2$$

*Proof.* Recall that

$$\Sigma_T^2 = \mathbb{E}\left[\max_{Q\in\mathcal{U}} \frac{1}{n_Q^2} \sum_{i=1}^{n_Q} Y_{i,Q}^2\right]$$

where we define $Y_{i,Q} := X_Q^{(i)} - \mu_Q \in [-1, 1]$ and note that $\mathbb{E}\left[Y_{i,Q}^2\right] = \sigma_Q^2$. Let us begin by noting that

$$\Sigma_T^2 \leq \mathbb{E}\left[\max_{Q \in \mathcal{U}} \frac{1}{n_Q^2} \sum_{i=1}^{n_Q} \left(Y_{i,Q}^2 - \mathbb{E}\left[Y_{i,Q}^2\right]\right)\right] + \sigma_T^2$$

For a one-sided symmetrization argument, let $Z_{i,Q}$ be independent copies of the $Y_{i,Q}$ and let $\epsilon^n \overset{iid}{\sim}$ Rad be independent from them, where $n := \max_{Q \in \mathcal{U}} n_Q$. Then, we can bound the first quantity in the upper bound as follows:

$$\mathbb{E}\left[\max_{Q \in \mathcal{U}} \frac{1}{n_Q^2} \sum_{i=1}^{n_Q} \left(Y_{i,Q}^2 - \mathbb{E}\left[Y_{i,Q}^2\right]\right)\right] = \mathbb{E}\left[\max_{Q \in \mathcal{U}} \mathbb{E}\left[\frac{1}{n_Q^2} \sum_{i=1}^{n_Q} \left(Y_{i,Q}^2 - Z_{i,Q}^2\right)\middle| Y\right]\right]$$

$$\leq \mathbb{E}\left[\max_{Q \in \mathcal{U}} \frac{1}{n_Q^2} \sum_{i=1}^{n_Q} \left(Y_{i,Q}^2 - Z_{i,Q}^2\right)\right]$$

$$= \mathbb{E}\left[\max_{Q \in \mathcal{U}} \frac{1}{n_Q^2} \sum_{i=1}^{n_Q} \epsilon_i \left(Y_{i,Q}^2 - Z_{i,Q}^2\right)\right]$$

$$\leq \mathbb{E}\left[\max_{Q \in \mathcal{U}} \frac{1}{n_Q^2} \sum_{i=1}^{n_Q} \epsilon_i Y_{i,Q}^2\right] + \mathbb{E}\left[\max_{Q \in \mathcal{U}} \frac{1}{n_Q^2} \sum_{i=1}^{n_Q} -\epsilon_i Z_{i,Q}^2\right]$$

$$= 2\mathbb{E}\left[\max_{Q \in \mathcal{U}} \frac{1}{n_Q^2} \sum_{i=1}^{n_Q} \epsilon_i Y_{i,Q}^2\right]$$

where $Y$ denotes the collection of all $Y_{i,Q}$'s. In the next lemma, we bound the last quantity above.

**Lemma G.2** (Contraction). *We have that*

$$\mathbb{E}\left[\max_{Q \in \mathcal{U}} \frac{1}{n_Q^2} \sum_{i=1}^{n_Q} \epsilon_i Y_{i,Q}^2\right] \leq \mathbb{E}\left[\max_{Q \in \mathcal{U}} C_Q \sum_{i=1}^{n_Q} \epsilon_i Y_{i,Q}\right]$$

*where $C_Q := \frac{2}{n_Q \cdot \min_{Q' \in \mathcal{U}} n_{Q'}}$.*

*Proof of Lemma G.2.* Fix an index $j \in [n]$, where $n := \max_{Q \in \mathcal{U}} n_Q$. For each $Q \in \mathcal{U}$, let us additionally define dummy variables $Y_{n_Q+1,Q}, \ldots, Y_{n,Q} := 0$, so that

$$\mathbb{E}\left[\max_{Q \in \mathcal{U}} \frac{1}{n_Q^2} \sum_{i=1}^{n_Q} \epsilon_i Y_{i,Q}^2\right] = \mathbb{E}\left[\max_{Q \in \mathcal{U}} \frac{1}{n_Q^2} \sum_{i=1}^{n} \epsilon_i Y_{i,Q}^2\right]$$

In what follows, we use $\mathbb{E}_{\epsilon_j}$ to denote an expectation only w.r.t. $\epsilon_j$, while all other random variables remain fixed (that is, conditioned on all other variables due to independence). Note that

$$\mathbb{E}_{\epsilon_j}\left[\max_{Q \in \mathcal{U}} \left\{\frac{1}{n_Q^2} \sum_{i=1}^{j} \epsilon_i Y_{i,Q}^2 + C_Q \sum_{i=j+1}^{n} \epsilon_i Y_{i,Q}\right\}\right]$$

$$= \frac{1}{2} \max_{Q,Q' \in \mathcal{U}} \left\{\frac{1}{n_Q^2} \sum_{i=1}^{j-1} \epsilon_i Y_{i,Q}^2 + \frac{Y_{j,Q}^2}{n_Q^2} + C_Q \sum_{i=j+1}^{n} \epsilon_i Y_{i,Q}\right.$$

$$\left. + \frac{1}{n_{Q'}^2} \sum_{i=1}^{j-1} \epsilon_i Y_{i,Q'}^2 - \frac{Y_{j,Q'}^2}{n_{Q'}^2} + C_{Q'} \sum_{i=j+1}^{n} \epsilon_i Y_{i,Q'}\right\}$$

Next, note that

$$\frac{Y_{j,Q}^2}{n_Q^2} - \frac{Y_{j,Q'}^2}{n_{Q'}^2} = \left(\frac{Y_{j,Q}}{n_Q} + \frac{Y_{j,Q'}}{n_{Q'}}\right)\left(\frac{Y_{j,Q}}{n_Q} - \frac{Y_{j,Q'}}{n_{Q'}}\right)$$

$$\leq \left(\frac{1}{n_Q} + \frac{1}{n_{Q'}}\right)\left|\frac{Y_{j,Q}}{n_Q} - \frac{Y_{j,Q'}}{n_{Q'}}\right|$$

$$\leq |C_Q Y_{j,Q} - C_{Q'} Y_{j,Q'}|$$

Hence,

$$\mathbb{E}_{\epsilon_j}\left[\max_{Q\in\mathcal{U}}\left\{\frac{1}{n_Q^2}\sum_{i=1}^{j}\epsilon_i Y_{i,Q}^2 + C_Q\sum_{i=j+1}^{n}\epsilon_i Y_{i,Q}\right\}\right]$$

$$\leq \frac{1}{2}\max_{Q,Q'\in\mathcal{U}}\left\{\frac{1}{n_Q^2}\sum_{i=1}^{j-1}\epsilon_i Y_{i,Q}^2 + C_Q\sum_{i=j+1}^{n}\epsilon_i Y_{i,Q}\right.$$

$$\left.+ \frac{1}{n_{Q'}^2}\sum_{i=1}^{j-1}\epsilon_i Y_{i,Q'}^2 + C_{Q'}\sum_{i=j+1}^{n}\epsilon_i Y_{i,Q'} + |C_Q Y_{j,Q} - C_{Q'} Y_{j,Q'}|\right\}$$

$$= \frac{1}{2}\max_{Q,Q'\in\mathcal{U}}\left\{\frac{1}{n_Q^2}\sum_{i=1}^{j-1}\epsilon_i Y_{i,Q}^2 + C_Q\sum_{i=j+1}^{n}\epsilon_i Y_{i,Q}\right.$$

$$\left.+ \frac{1}{n_{Q'}^2}\sum_{i=1}^{j-1}\epsilon_i Y_{i,Q'}^2 + C_{Q'}\sum_{i=j+1}^{n}\epsilon_i Y_{i,Q'} + C_Q Y_{j,Q} - C_{Q'} Y_{j,Q'}\right\}$$

$$= \mathbb{E}_{\epsilon_j}\left[\max_{Q\in\mathcal{U}}\left\{\frac{1}{n_Q^2}\sum_{i=1}^{j-1}\epsilon_i Y_{i,Q}^2 + C_Q\sum_{i=j}^{n}\epsilon_i Y_{i,Q}\right\}\right]$$

From independence, we can thus integrate iteratively starting at $j = n$ to conclude that

$$\mathbb{E}\left[\max_{Q\in\mathcal{U}}\frac{1}{n_Q^2}\sum_{i=1}^{n}\epsilon_i Y_{i,Q}^2\right] \leq \mathbb{E}\left[\max_{Q\in\mathcal{U}} C_Q\sum_{i=1}^{n}\epsilon_i Y_{i,Q}\right] = \mathbb{E}\left[\max_{Q\in\mathcal{U}} C_Q\sum_{i=1}^{n_Q}\epsilon_i Y_{i,Q}\right]$$

∎

Again using symmetrization, let $Z_{i,Q}$ be independent copies of the $Y_{i,Q}$ and independent from $\epsilon^n$. Since $Y_{i,Q}$ are centered, we have that

$$\mathbb{E}\left[\max_{Q\in\mathcal{U}} C_Q\sum_{i=1}^{n_Q}\epsilon_i Y_{i,Q}\right] = \mathbb{E}\left[\max_{Q\in\mathcal{U}}\mathbb{E}\left[C_Q\sum_{i=1}^{n_Q}\epsilon_i\left(Y_{i,Q} - Z_{i,Q}\right)\Big|\epsilon^n, Y\right]\right]$$

$$\leq \mathbb{E}\left[\max_{Q\in\mathcal{U}} C_Q\sum_{i=1}^{n_Q}\epsilon_i\left(Y_{i,Q} - Z_{i,Q}\right)\right]$$

$$= \mathbb{E}\left[\max_{Q\in\mathcal{U}} C_Q\sum_{i=1}^{n_Q}\left(Y_{i,Q} - Z_{i,Q}\right)\right]$$

$$\leq 2\mathbb{E}\left[\max_{Q\in\mathcal{U}} C_Q\left|\sum_{i=1}^{n_Q} Y_{i,Q}\right|\right]$$

Next, we bound this expectation using Hoeffding's inequality. We begin with a high-probability bound:

$$\mathbb{P}\left(\max_{Q\in\mathcal{U}} C_Q\left|\sum_{i=1}^{n_Q} Y_{i,Q}\right| \geq t\right) \leq \sum_{Q\in\mathcal{U}}\mathbb{P}\left(C_Q\left|\sum_{i=1}^{n_Q} Y_{i,Q}\right| \geq t\right)$$

$$\leq 2\sum_{Q\in\mathcal{U}}\exp\left(-\frac{2t^2}{C_Q^2 n_Q}\right)$$

$$= 2\sum_{Q\in\mathcal{U}}\exp\left(-\frac{t^2 n_Q \min_{Q'\in\mathcal{U}} n_{Q'}^2}{2}\right)$$

$$\leq 2k\exp\left(-\frac{t^2 \min_{Q\in\mathcal{U}} n_Q^3}{2}\right)$$

We can subsequently integrate the tails to obtain the in-expectation bound

$$\mathbb{E}\left[\max_{Q\in\mathcal{U}} C_Q \left|\sum_{i=1}^{n_Q} Y_{i,Q}\right|\right] \leq 2\sqrt{\frac{2\log(2k)}{\min_{Q\in\mathcal{U}} n_Q^3}}$$

Combining all bounds presented thus far finally yields

$$\Sigma_T^2 \leq 8\sqrt{\frac{2\log(2k)}{\min_{Q\in\mathcal{U}} n_Q^3}} + \sigma_T^2$$

∎

Next, we show how $V_T$ relates to $\Sigma_T^2$.

**Lemma G.3.** *We have that*

$$V_T \leq \min\left\{\max_{Q\in\mathcal{U}} n_Q, k\right\} \Sigma_T^2$$

*Proof.* Let $n := \max_{Q\in\mathcal{U}} n_Q$ and note that we can equivalently express

$$V_T = \sum_{i=1}^{n} \mathbb{E}\left[\max_{Q\in\mathcal{U}:n_Q\geq i} \frac{1}{n_Q^2}\left(X_Q^{(i)} - \mu_Q\right)^2\right]$$

From this, we see that

$$V_T \leq n\mathbb{E}\left[\max_{i\in[n]} \max_{Q\in\mathcal{U}:n_Q\geq i} \frac{1}{n_Q^2}\left(X_Q^{(i)} - \mu_Q\right)^2\right]$$

$$= n\mathbb{E}\left[\max_{Q\in\mathcal{U}} \max_{i\in[n_Q]} \frac{1}{n_Q^2}\left(X_Q^{(i)} - \mu_Q\right)^2\right]$$

$$\leq n\mathbb{E}\left[\max_{Q\in\mathcal{U}} \sum_{i=1}^{n_Q} \frac{1}{n_Q^2}\left(X_Q^{(i)} - \mu_Q\right)^2\right]$$

$$= n\Sigma_T^2$$

Alternatively, we can begin by bounding the max by a sum in $V_T$:

$$V_T \leq \mathbb{E}\left[\sum_{i=1}^{n} \sum_{Q\in\mathcal{U}:n_Q\geq i} \frac{1}{n_Q^2}\left(X_Q^{(i)} - \mu_Q\right)^2\right]$$

$$= \mathbb{E}\left[\sum_{Q\in\mathcal{U}} \sum_{i=1}^{n_Q} \frac{1}{n_Q^2}\left(X_Q^{(i)} - \mu_Q\right)^2\right]$$

$$\leq k\mathbb{E}\left[\max_{Q\in\mathcal{U}} \sum_{i=1}^{n_Q} \frac{1}{n_Q^2}\left(X_Q^{(i)} - \mu_Q\right)^2\right]$$

$$= k\Sigma_T^2$$

∎

Finally, we prove the upper bound on $V_T$ stated in the example of Section 3.4.3.

**Lemma G.4.** *Let $\mathcal{U} = \{Q_1, \ldots, Q_k\}$, where*

- *$Q_1, \ldots, Q_{k-1}$ share a common variance $\sigma^2$ and are supported in $[0, 1]$.*

- *$Q_k$ has variance $\nu^2$.*

- *We sample $n$ times from each $Q_1, \ldots, Q_{k-1}$ and $m = T - n(k-1) \geq n$ times from $Q_k$, for a total of $T \geq nk$ samples.*

*Then,*

$$V_T \leq \frac{\sqrt{2\log(k-1)} + \sigma^2}{n} + \frac{\nu^2}{T - n(k-1)}$$

*Proof.* Note that

$$V_T = n\mathbb{E}\left[\max\left\{\max_{j\in[k-1]}\left\{\frac{1}{n^2}\left(X_{Q_j} - \mu_{Q_j}\right)^2\right\}, \frac{1}{m^2}\left(X_{Q_k} - \mu_{Q_k}\right)^2\right\}\right] + \frac{(m-n)\nu^2}{m^2}$$

$$\leq \frac{1}{n}\mathbb{E}\left[\max_{j\in[k-1]}\left\{\left(X_{Q_j} - \mu_{Q_j}\right)^2\right\}\right] + \frac{n\nu^2}{m^2} + \frac{(m-n)\nu^2}{m^2}$$

$$= \frac{1}{n}\underbrace{\mathbb{E}\left[\max_{j\in[k-1]}\left\{\left(X_{Q_j} - \mu_{Q_j}\right)^2\right\}\right]}_{(*)} + \frac{\nu^2}{m}$$

Since $\left|\left(X_{Q_j} - \mu_{Q_j}\right)^2 - \sigma^2\right| \leq 1$ for all $j \in [k-1]$, we then have that

$$(*) = \mathbb{E}\left[\max_{j\in[k-1]}\left\{\left(X_{Q_j} - \mu_{Q_j}\right)^2 - \sigma^2\right\}\right] + \sigma^2 \leq \sqrt{2\log(k-1)} + \sigma^2$$

∎

