# OpenReview forum: "Distribution-Dependent Rates for Multi-Distribution Learning"
_ICLR.cc/2025/Conference — Submitted to ICLR 2025_

### Official Review · Reviewer_eYFo · 2024-11-02

**Soundness:** 3
**Presentation:** 4
**Contribution:** 2
**Rating:** 5
**Confidence:** 4

**Summary:**

This paper presents innovative strategies within the framework of Multi-Distribution Learning (MDL), with the primary objective of identifying the best-performed distribution. It is informed by principles from Distributionally Robust Optimization (DRO) and multi-armed bandit theory, proposing both non-adaptive and adaptive methodologies. The non-adaptive techniques, namely Uniform Exploration (UE) and Non-Uniform Exploration (NUE), yield both distribution-independent and distribution-dependent bounds. Furthermore, the paper introduces an adaptive method in the interactive environment, LCB-DR, which further optimizes performance by employing optimistic sampling strategies analogous to the Upper Confidence Bound for Exploration (UCB-E) utilized in multi-armed bandit scenarios.

**Strengths:**

The paper is well-written and clearly conveys the problem setting and the conclusion to the reader. This work provides a distribution-dependent bound with analysis, which has not been reported in the literature. The distribution-dependent bound enjoys an exponential decay which can be compared to the probability of identification failure in the Best-arm Identification. Furthermore, this paper does not limit itself to non-adaptive exploring but extends to an adaptive exploring strategy, which is the UCB-E algorithm.

**Weaknesses:**

Since the author mentions that MDL draws inspiration from multi-armed bandits, I have found that identifying the best-performed distribution can be viewed as an analogy to identifying the best arm (BAI) in MAB. In lines 199-203, the author also mentions a connection between this work and BAI, which is a $H_a$ term; It would be better if the author could draw more comparisons between MDL and BAI. Since several works in BAI can achieve instance-independent bound \cite{audibert2010best, chen2017towards}. How does the objective upper bound guarantee relate to the existing bound shown in the BAI literature? I believe it is important to show whether the given bound is tight when reducing the problem setting to the existing work.

**Questions:**

- What is the definition of $a^\star$ in lines 199-203.

- What is $l$ appearing in the RHS of Eq in lines 263-266?

- Is $M$ a known parameter or an unknown parameter to the agent?

- In lines 320-321, why $\Delta_{\text{DR}}(a) \approx B_n$ induces the comparison to be $\tfrac{M^2}{n}$ v.s. $\sigma^2_T + \Sigma^2_T + V_T$? Both exponential terms should be $1$ when the exponential factor becomes $0$.

**Details Of Ethics Concerns:**

There are no ethical concerns.

---

> ### Author Response · Authors · 2024-11-15
>
> $\newcommand{\Uj}{\mathcal{U}_j}$
> Thank you for the feedback. To answer your first point, let us detail how the algorithms presented mirror their BAI counterparts:
> - The UE strategy in the BAI literature similarly samples $n$ times from each arm $a$ and ultimately picks the one with best empirical reward $\mu_T^\text{o}(a) \coloneqq \frac{1}{n}\sum_{i=1}^n r(X_a^{(i)})$. Since this is an unbiased estimate of the true mean, the regret analysis is a straightforward application of Hoeffding's inequality and yields the rate $\sum_a \Delta(a)\exp(-n\Delta^2(a))$, ignoring constants. In the MDL setting, there are two variables: the decision $a$ and the distributions $Q$ (serving the role of the ''arm'' in the BAI context). In essence, we apply UE on the latter to construct empirical proxy $\mu_T^\text{o}(a) = \min_Q \frac{1}{n}\sum_{i=1}^n r(a,X_Q^{(i)})$. Now, the analysis is more involved since $\mu_T^\text{o}$ is no longer an unbiased estimate of the true objective function, due to the minimization. For this reason, we instead rely on tools from empirical process theory and obtain the analogous regret of $\sum_a \Delta(a)\exp(-n[\Delta(a)-\sqrt{\log k / n}]^2)$. For more details on the BAI UE, we refer to Chapter 33 of Lattimore and Szepesvári - Bandit Algorithms.
>
> - Our LCB-DR algorithm essentially uses the UCB-E strategy from BAI to find the worst-performing distribution for each decision $a$. Hence, let us take an iteration $j$ of LCB-DR and compare it to UCB-E for $k$ arms over $T$ rounds. Focusing on the exponentially decaying term, the latter guarantees a rate of $\exp(-(T-k)/H)$, where $H = \sum_a \Delta^{-2}(a)$ captures the instance complexity. In LCB-DR, our goal is to apply the same idea on the distributions $Q\in\mathcal{U}$. However, by round $j$, we have already collected data in the first $j-1$ rounds and would like to make use of it, in addition to the $T_j$ samples collected in this round. Intuitively, if some distribution has already been played enough times, the learner can identify it as suboptimal. This is what the set $\Uj\subset\mathcal{U}$ captures: it is a guess of the distributions that the learner is still unsure about and will thus end up playing on round $j$ (note that we do not constrain the decision to be from $\Uj$, so this quantity can be unknown). In Appendix D, we derive a new analysis of UCB-E for the setting where the learner starts the game with some samples, as is the case here. Corollary D.1 shows that $\Uj$ indeed contains the arms played. The regret bound shows that the learner essentially plays UCB-E only on arms $\Uj$ and, thus, the resulting rate scales with $H_j=\sum_{Q\in \Uj}\Delta_{a_j}^{-2}(Q)$, which can be much smaller than $H$ (in MDL, this translates to summing over all of $\mathcal{U}$) as it only sums over a subset of $\mathcal{U}$. In addition, the total number of pre-existing samples $\tilde T_j$ from $\Uj$ also contributes to the rate and we only offset by the size $k_j$ of $\Uj$. Putting these together, the LCB-DR regret for this round scales with $\exp(-(C_{a_j}^2\wedge 1)(T_j+\tilde T_j-k_j)/H_j)$, where $C_{a_j}$ is a complexity measure specific to MDL (see Section 4.1 for some intuition).
>
> With regards to tightness of these bounds, it remains an open question. Translating results from BAI to MDL is not very straightforward, for the reasons mentioned above, and our work does not focus on the lower bounds. It would be interesting to see such guarantees in follow-up works.
>
> As for the questions posed:
> - We apologize for the omission. Here, $a^* = \text{arg}\max_{a\in\mathcal{A}} \mu_\text{DR}(a)$ is the optimal distributionally robust decision, which is assumed to be unique for the LCB-DR algorithm (its analysis is the only one that uses $C_a$). More generally, we can define
> $$
> C_a=\begin{cases}
> \frac{\Delta_\text{DR}(a)}{\Delta_{a,\text{min}}} & a \not\in \text{arg}\max_{a\in\mathcal{A}} \mu_\text{DR}(a) \\\\
> \frac{\Delta_{\text{DR,min}}}{\Delta_{a,\text{min}}} & \text{o.w.}
> \end{cases}
> $$
>
> - $l=|\mathcal{A}|$ is the number of decisions available to the learner, as defined in Assumption 1.
>
> - $M$ does not have to be known to the learner.
>
> - Perhaps a more appropriate characterization would be $\Delta_\text{DR}(a)-B_n$ being suitably small. The idea is to mirror the Hoeffding v.s. Bernstein difference: for a random variable bounded in $[0,M]$ and with variance $\sigma^2$, the bounds are $\text{exp}(-t^2/M^2)$ v.s. $\text{exp}(-t^2/(\sigma^2+Mt))$, up to constants. For suitably small $t$, the $\sigma^2$ term dominates the sum $\sigma^2+Mt$ and the comparison boils down to $M^2$ v.s. $\sigma^2$, where the smaller term is better (note that here, the latter is always better). Similarly, when $\Delta_\text{DR}(a)-B_n$ is small relative to $\sigma_T^2+\Sigma_T^2+V_T$ (which occurs in small sample regimes), the latter dominates the denominator of the NUE rate and the UE v.s. NUE comparison reduces to $M^2/n$ v.s. $\sigma_T^2+\Sigma_T^2+V_T$.

---

> > ### Comment · Reviewer_eYFo · 2024-11-27
> >
> > I like the response by the author and their explanation is clear to me. I will keep my score.

---

### Official Review · Reviewer_ExHf · 2024-11-04

**Soundness:** 3
**Presentation:** 3
**Contribution:** 3
**Rating:** 6
**Confidence:** 2

**Summary:**

The authors study distribution-dependent guarantees in the multi-distribution learning framework. They prove that distribution-dependent bounds are tighter than distribution-independent bounds. Specifically, they derive finite sample bounds under uniform and non-uniform exploration and propose an algorithm that improves over non-adaptive counterparts.

**Strengths:**

The authors clearly compared the finite sample bounds under uniform exploration against that under non-uniform exploration, and they highlighted where non-uniform exploration could have gains.

**Weaknesses:**

The authors compared their proposed algorithm against uniform sampling, but not non-uniform sampling. Non-uniform sampling benefits from varied sampled sizes and would be a stronger baseline to compare against.

It would be nice to provide experimental results, even in very simple set ups, to showcase the strength of their proposed algorithm. The main results of this work are in theoretical results and there are substantial theoretical contributions, and thus I understand the experimental results may not be necessary.

**Questions:**

How does this work relates to active learning, where one would also take an adaptive strategy? What are the barriers that prevent active learning algorithm from being applied to the problem setting studied in this work?

Are there relevant lower bounds available? If so, how do the upper bounds proven in this work compare to the lower bounds?

---

> ### Author Response · Authors · 2024-11-16
>
> Thank you for the thoughtful feedback. Below, we address the identified weaknesses and respond to your specific questions.
>
> **Weaknesses**
> - **(UCB v.s. NUE)** Our core interest is in comparing adaptive v.s. non-adaptive strategies. While NUE incorporates a more nuanced approach than UE by exploiting the variance of different distributions, its advantage arises from a fundamentally different principle than LCB-DR, which leverages optimism by collecting data in an adaptive manner. The comparison between UE and LCB-DR directly examines the impact of adaptivity in terms of suboptimality gaps, providing insights into the trade-offs between simple exploration and more sophisticated adaptive strategies. Switching to NUE does not add any insight in this direction. In essence, NUE and LCB-DR offer complementary benefits relative to UE, and we opted to evaluate each one separately.
> - **(Experimental results)** While experiments are indeed interesting to evaluate our methods, the central goal of this work is to provide a fundamentally different type of analysis for MDL algorithms. Our distribution-dependent approach examines how regret decays relative to instance-specific parameters (e.g., suboptimality gaps), offering a perspective that previous analyses did not address. This type of theoretical insight is challenging to illustrate through experiments, as the distribution-dependent rates of existing algorithms are not known.
>
> **Questions**
> - Active learning (AL) shares some similarities with MDL in that both involve actively collecting data. However, there are two fundamental differences between the two:
>   - AL is specific to supervised learning, where the data is i.i.d. from a single distribution; the learner's goal is then to select which points to label, given access to covariates. In contrast, MDL can also address unsupervised tasks and involves multiple distributions; the learner's goal is to decide which distributions to sample from. This distinction means that, in MDL, the data can be tailored to fundamentally different tasks (when distributions are very dissimilar), leading to sampling strategies that differ significantly from those in AL.
>   - The ultimate goal of AL is to produce a decision with good expected performance under the single data-generating distribution. In MDL, the objective is instead to obtain uniformly good performance across a collection of distributions. As a result, the way the learner uses the collected data to make decisions can vary significantly between the two settings.
>
>   Due to these fundamental differences in the scope of both settings, ideas from AL are not so directly transferable to MDL.
>
> - As far as we are aware, there are no lower bounds for distribution-dependent rates.

---

> > ### Comment · Reviewer_ExHf · 2024-11-26
> >
> > I thank the authors for the response. I will keep my score and defer to other reviewers who are more familiar with this research area.

---

### Official Review · Reviewer_2Ve4 · 2024-11-04

**Soundness:** 3
**Presentation:** 2
**Contribution:** 3
**Rating:** 6
**Confidence:** 2

**Summary:**

This paper studies multiple distribution learning (MDL) from bandit optimization point of view. By connecting pure-exploration setting in bandits to MDL, it develops instance-dependent sharp regret rates, thereby improving the current instance-agnostic rates in the MDL literature.

**Strengths:**

The paper uses techniques from pure-exploration bandits to develop instance-dependent simple regret rates, which serves as less-conservative complements to the current instance-agnostic MDL error bounds.

**Weaknesses:**

- The paper claims one of its contribution being developing problem-dependent rates for MDL, because “Oftentimes, it is more intuitive to analyze the learner’s performance in a fixed setting, as opposed to considering a worst-case instance for each sample size. When domain knowledge is available, a “one-size-fits-all” rate does not provide any insight on how to take advantage of this information”. However, the upper bound rates developed in this paper depend on the knowledge of the unknown optimality gap; how would this be integrated into domain knowledge remains unclear.
- The problem setting seems very similar to Kirschner et al for distributionally robust online contextual bandit problem, but no discussion is provided on the differences and connections.

**Questions:**

- Following up the first point in the weaknesses, Would the story of the paper rather be, given the knowledge that the uncertainty set $\mathcal{U}$ is fixed, one can develop exponential rates that scale with an unknown but fixed sub-optimality gap of each arm?
- The paragraph in the introduction states “The current literature is populated with distribution-independent rates”, but there were not any relevant literature cited in this paragraph.
- How would the proof be adjust to accommodate the case where the learner interacts with the environment but the optimal arm is non-unique?
- Typo on Line 223: a fixed number times -> a fixed number of times

---

> ### Author Response · Authors · 2024-11-16
>
> Thank you for the thoughtful comments. Below, we address the weaknesses and questions posed.
>
> **Weaknesses**
> - **(Distribution-dependent analysis)** Our motivation for studying distribution-dependent rates is not necessarily to leverage domain knowledge of suboptimality gaps to devise algorithms, but to understand how regret scales with instance-specific quantities (e.g., suboptimality gaps). In the multi-armed bandit literature, it is well-established that such guarantees typically decay much faster with the sample budget compared to their distribution-independent counterpart (e.g., $\exp(-T)$ vs. $1/\sqrt{T}$). Our goal is to bring this perspective to the MDL paradigm.
>
>   A consequence of this is that when domain knowledge is indeed available, distribution-dependent rates can reveal the benefits of exploiting this information. For example, the NUE analysis shows the interplay between variance and sample size allocations, so that when the data is real-valued and the learner has some understanding of the variances across distributions, they can gain insight on how directing more data to higher-variance distributions impacts regret.
>
> - **(DR Bayesian optimization)** The setting of https://arxiv.org/pdf/2002.09038 is quite different from ours: they assume the learner receives some stochastic context $c_t$, sampled from a distribution in an uncertainty set $\mathcal{U}_t$ that changes with time, makes a decision $x_t$ and receives a reward $f(x_t,c_t)$ with additive noise. Their objective is to optimize a cumulative notion of regret. In our setting, we have a fixed uncertainty set and the task is to collect data from it, with the end goal of producing a decision that is distributionally robust in an offline sense.
>
> **Questions**
> - **(Paper motivation)** This interpretation is a good way to put it. As highlighted above, the idea is to understand how the rates scale with (possibly unknown) instance-specific parameters. The stark contrast between the distribution-dependent and independent guarantees stem precisely from this difference in perspective: the former fixes an environment and studies how an algorithms performs in it, whereas the latter applies to worst-case environments for each sample size and, thus, cannot scale with environment-specific quantities.
>
> - **(Omitted literature)** This literature is presented in Section 1.2. We will make the appropriate citations earlier for further clarity.
>
> - **(Non-unique optimum)** The LCB-DR analysis (the other strategies do not assume a unique optimum) remains unchanged if we more generally redefine the complexity measure
> $$
> C_a=\begin{cases}
> \frac{\Delta_\text{DR}(a)}{\Delta_{a,\text{min}}} & a \not\in \text{arg}\max_{a\in\mathcal{A}} \mu_\text{DR}(a) \\\\
> \frac{\Delta_{\text{DR,min}}}{\Delta_{a,\text{min}}} & \text{o.w.}
> \end{cases}
> $$
>
> - **(Typo)** Thank you for pointing this out. We will make the necessary adjustment.

---

### Official Review · Reviewer_pQgn · 2024-11-05

**Soundness:** 2
**Presentation:** 2
**Contribution:** 3
**Rating:** 3
**Confidence:** 3

**Summary:**

This paper addresses the multi-distribution learning problem, where the learner aims to optimize the model's worst-case performance across a set of distributions. The main contribution is a reformulation of this problem as a pure exploration multi-armed bandit task and obtain simple regret bounds that depend on the sub-optimal gap of actions. The first part of the paper studies the non-adaptive case, where the learner cannot interact with the environments. Here, the authors provide simple regret bounds for both uniform exploration (UE) and non-uniform exploration (NUE). The second part explores the interactive case, where environment interaction is permitted, and proposes an LCB-based algorithm that better a lower simple regret than UE.

**Strengths:**

- This paper offers a new perspective by formulating multi-distribution learning as a pure exploration problem in multi-armed bandits.
- Based on this view, gap-dependent bounds are derived for both adaptive and non-adaptive cases for the multi-distribution learning problem.

**Weaknesses:**

- **On the Significance of the Results:** One of my main concerns is the significance of the results achieved in this paper, as they rely on strong assumptions, and their implications are not rigorously discussed.
   - **Assumptions:** The paper appears to address a simplified case where the action space $ \mathcal{A} $ is discrete and finite, and the data space is restricted to 1-dimension in the analysis. In contrast, continuous decision sets are more commonly studied in the literature, such as Blum et al. (2017), Sagawa et al. (2020), and Soma et al. (2022). Although Section 5 discusses an extension to infinite decision sets, the proposed approach using an $ \epsilon/k $-cover would result in a method with prohibitive computational costs.
   - **Results for the Non-Adaptive Case:** It is not entirely clear to me why the results for Non-Uniform Exploration (NUE) would be better than Uniform Exploration (UE), as the NUE outcomes depend on $\min_Q{n_Q}$, which could potentially be very small. The arguments in Section 3.3 are too intuitive to me, with several approximations made that require further justification. For instance, I am confused by the statement "considers a case $\Delta_{DR}(a) \approx B_n$"  in line 321. It is uncertain whether we can disregard the term $\Delta_{DR}(a) - B_n$ in the comparison, as this term varies across arms, and the value of $B_n$ differs between UE and NUE cases.

- **On the Proposed Method in Section 4:** The proposed method for adaptive cases requires knowledge of $H_j$, which depends on the suboptimal gap $\Delta_{a,\min}$ and is generally unknown in practice. Although the authors provide some discussion in Remark 4, it remains unclear how this issue would be addressed. Additionally, the setting of $\epsilon_t$ is also confusing. While this quantity appears to only require to be lower bounded, it is still unclear how to set this value to ensure that Condition Eq. (1) is not violated.
- **About literature review**:  The discussion of the convergence rate for related work in lines 139-152 is inaccurate. Although Soma et al. (2022) claim a result of $O(\frac{\sqrt{B^2 + k}}{T})$, their analysis overlooks the non-oblivious property of the learning process, rendering the result invalid. This issue was identified by Zhang et al. (2023), and the currently best-known result remains $O(\frac{\sqrt{B^2 + k \log k }}{T})$ in this line of research. One may check Section 2.3 in Zhang et al 2023 for a discussion.

**Questions:**

- Is it possible to extend the results to continuous spaces without a significant increase in computational complexity? For example, could pure exploration in linear bandit settings be considered?
- Could you provide more detailed explanations on the comparison between UE and NUE? (Please refer to the first point under weaknesses for a more detailed discussion.)
- How can the parameter $T_j$ be set in practice to ensure the theoretical guarantees?

---

> ### Author Response · Authors · 2024-11-17
>
> $\newcommand{\Eb}{\mathbb{E}}\newcommand{\Xc}{\mathcal{X}}\newcommand{\Ac}{\mathcal{A}}\newcommand{\dam}{\Delta_{a_j,\text{min}}}$
> Thank you for the insightful comments. Below, we address the issues presented:
> - **(1-dimensional data assumption)** The restriction to real-valued data is only for NUE and its comparison to UE. Otherwise, the analyses hold for arbitrary spaces.
> - **(NUE results)** The UE v.s. NUE comparison mirrors the Hoeffding v.s. Bernstein inequalities for variables bounded in $[0,M]$ and with variance $\sigma^2$. In particular, the latter reduces to $\exp(-t^2/M^2)$ v.s. $\exp(-t^2/(\sigma^2+Mt))$, up to constants. For suitably small $t$, this boils down to $M^2$ v.s. $\sigma^2$, where the latter always wins. Similarly, in Section 3.3, we compare the rates when $\Delta_\text{DR}(a)-B_n$ is suitably small, which occurs in small sample regimes, yielding the analogous $M^2/n$ v.s. $\sigma_T^2+\Sigma_T^2+V_T$. As noted, this term indeed varies across arms and may be more ''suitably small'' for some.
>
>   Moreover, while the guarantees of UE and NUE use $B_n=\sqrt{\log k/n}$ and $B_n=G_T$, respectively, we note in Appendix C that either term works for both rates. We use $G_T$ for NUE as it showcases the dependence on variance. However, when comparing the strategies, we assume small sample sizes, where $B_n$ is less significant and the focus is on the quantities above, so that we apply the same $B_n$ to both algorithms.
>
> As for the prohibitive cost of creating a cover, this may indeed be a barrier in full generality. Let us highlight some settings where this is more feasible, the latter being the subject of your first question.
> - **(VC classes)** Binary classification has been a major focus of the literature (e.g., Blum et al. (2017)). We mention this in Section 5 and further develop it in Appendix F. Suppose that the data is of the form $(X,Y)\in\Xc\times\\{0,1\\}$, decisions are binary-valued functions on $\Xc$ and the reward function is $r(a,(x,y)) = \mathbb{I}\\{a(x)=y\\}$. In Appendix F.1, we show that given a finite $\sqrt{\epsilon/k}$-cover $\Ac_\epsilon$ of $(\Ac,L^2(\bar Q_\Xc))$ (we refer to Appendix F.1 for the definitions), we can run the algorithms on it and incur an approximation penalty of $\epsilon$ on the regret. Suppose that $\Ac$ has finite VC dimension $d$. To construct a cover with high-probability, we can sample $O(k^2(d+\log(1/\delta))/\epsilon^2)$ points from $\bar Q_\Xc$ and project $\Ac$ onto this sample. By the Sauer–Shelah lemma, we also know that $|\Ac_\epsilon|\lesssim(k^2(d+\log (1/\delta))/(d\epsilon^2))^d$. With a slightly different analysis, we can also reduce the $k^2$ to $k$.
> - **(Linear classes)** Suppose that $r(a,x)=v^\top\psi(a,x)$, for some known vector $v$ and feature mapping $\psi$ both lying in the unit Euclidean ball in $d$ dimensions (recall that the learner has access to $r$). Let $\psi(a,\bar Q)\coloneqq \Eb_{X\sim\bar Q}[\psi(a,X)]$, where $\bar Q$ is the uniform mixture of $Q_{1:k}$. The problem then reduces to constructing a cover of $\Ac$ w.r.t. the pseudo-metric $d(a,a')=\Vert\psi(a,\bar Q)-\psi(a',\bar Q)\Vert_2$. While we cannot access $\bar Q$ directly, we can sample from it to approximate $\psi(a,\bar Q)$. With $O(d\log(1/(\epsilon\delta))/\epsilon^2)$ samples, we can construct an $\epsilon$-cover of size $O(1/\epsilon^d)$ with high-probability.
>
> Regarding unknown quantities in LCB-DR, we make some remarks:
> - $\epsilon_j$ depends on the unknown quantities $H_{a_1:a_{j-1}}$ and $\dam$. The former are sums of the suboptimality gaps of $a_{1:j-1}$, for which we have already ran UCB-E. Hence, we can estimate them using the current data: $\Delta_{a_r}(Q)\approx\hat\mu_{n_{\bar T_{r}}}(a_r; Q) - \mu_T^\circ(a_r)$. The term $\dam$ is trickier as we have not yet collected samples for $a_j$. One idea is to use the data gathered so far via the proxy $Q\mapsto\hat\mu_{n_{\bar T_{j-1}}}(a_j; Q)$. However, this may not yield a good estimate if the available data is not informative for $a_j$. Given the assumption $r\in[0,1]$, another option is to use $1$ as a crude upper bound of $\dam$. Note that a large value for $\epsilon_j$ is not an issue since the regret is a decreasing function of it.
> - Regarding $T_j$, we make a correction to Remark 4: Theorem 3 holds provided that the lower bound $T_j\gtrsim\epsilon_jH_j-\tilde T_j+k_j$ is satisfied. The regret, however, scales with $\epsilon_j$, and if the bound is loose, the dependence on $T_j$ is worse. The change is in Line 464: if we plug in a different value for $\epsilon_j$, the rate is no longer $O\left(\frac{(C_{a_j}^2\wedge 1)(T_j+\tilde T_j-k_j)}{H_j}\right)$. Ideally, we would choose $T_j$ so that the lower bound is as tight as possible. In practice, if the right-hand side is difficult to estimate, the learner can choose a large value for $\epsilon_j$, making the other terms relatively small, and select $T_j$ slightly above it. As discussed, a large $\epsilon_j$ should not be an issue.

---

> > ### Comment · Reviewer_pQgn · 2024-11-25
> >
> > I appreciate the authors' detailed feedback. However, I remain concerned about the rigor of the arguments and the assumptions presented (e.g., the one-dimensional assumption for the NUE case and the difficulties in setting the parameter). In the comparison of UE and NUE, it is unclear how the term  $\min_{Q \in \mathcal{U}} n_Q$ influences the performance of the two bounds. I believe a more rigorous analysis would be beneficial. For instance, it would be helpful to provide a specific condition (e.g., $n$ being greater than certain quantities) under which NUE is strictly better than UE, rather than relying on the asymptotic arguments in Section 3.3. Additionally, the establishment of covering still appears to be computationally challenging in the stated cases, as its complexity scales exponentially with the dimension and the size of the data.

---

> > > ### Author Response · Authors · 2024-11-26
> > >
> > > $\newcommand{\DDR}{\Delta_{\text{DR}}(a)}$
> > > $\newcommand{\ub}[2]{\underbrace{#1}_{#2}}$
> > > Thank you for the follow-up highlighting these issues. Below, we provide additional clarification on the concerns raised:
> > >
> > > **(Rigor of UE v.s. NUE)** We agree that the UE v.s. NUE comparison is rather handwavy, so let us extend the example of Section 3.4.3 for a more concrete setup: suppose that we have $C\gg 1$ distributions $Q_1,\dots,Q_C$, and the distributions $Q_1,\dots,Q_{C-1}$ share a common variance of 1 and distribution $Q_C$ has the much larger variance of $C$. Consider the non-uniform allocation of $2C$ samples that allocates 1 sample for each $Q_1,\dots,Q_{C-1}$ and $C$ samples to $Q_C$ (more formally, it should be $C+1$, but we ignore the $+1$). In addition, suppose that the reward function is bounded in $r\in[0,C]$. Note that a uniform allocation here would sample 2 times from each distribution, so that applying the same $B_n$ (namely, the first formula of Theorem C.1 scaled by the bound on $r$) to both UE and NUE yields $B_n \asymp C\sqrt{\log C}$ to both bounds. Then using Lemma G.4 (as in Section 3.4.3), we can conclude that the exact bounds, up to contants, are
> > > $$
> > > \ub{\exp\left(-\frac{(\DDR-C\sqrt{\log C})^2}{C^2}\right) }{\text{UE}} \quad\text{v.s.}\quad \ub{\exp\left(-\frac{(\DDR-C\sqrt{\log C})^2}{\sqrt{\log C} + C(\DDR-C\sqrt{\log C})}\right) }{\text{NUE}}
> > > $$
> > > Now, suppose that $\DDR = C\sqrt{\log C} + \epsilon_a/C$ for some small $\epsilon_a>0$. Then, the comparison becomes
> > > $$
> > >     \ub{ C^2 }{\text{UE}} \quad\text{v.s.}\quad \ub{ \sqrt{\log C}+\epsilon_a }{\text{NUE}}
> > > $$
> > > Evidently, when $C$ is large, the NUE strategy of sampling more from the higher-variance distribution $Q_C$ yields a better rate. This is true for any arm, provided that $\epsilon_a\ll C^2$.
> > >
> > > This example illustrates a broader setting where NUE can outperform UE: the term $\min_Q n_Q$ did not pose an issue here since, up to constants, it is equal to the UE allocations. Then, by leveraging the different variances, NUE significantly reduced the dependence on them.
> > >
> > > **(Computational challenges)** Indeed, the exponential dependence on the dimension poses a computational challenge. The primary focus of this work is on the statistical complexity of learning in the MDL problem, and computational efficiency, while an important aspect, is outside the scope of this analysis. We note that such challenges are not unique to our work but are inherent to problems involving high-dimensional distributions. This aligns with the standard PAC learning framework, where a statistical-computational tradeoff often exists. Bridging this gap in the MDL paradigm, whether through new algorithms or alternative covering techniques, is an interesting direction for future research.

---

### Meta-Review · Area_Chair_9ADV · 2024-12-21

**Metareview:**

This is a borderline paper. The paper addresses multi-distribution learning by means of multi-armed bandit algorithms and they attain regret bounds in this setting. The reviewers are particularly critical of the strong assumptions that are imposed and a lack of justification of the made assumptions/discussion of the assumptions. I believe that a major revision is necessary to address the shortcomings pointed out by the reviewers.

**Additional Comments On Reviewer Discussion:**

The reviewers replied to the authors after the rebuttal but no major discussion evolved.

---

### Decision · Program_Chairs · 2025-01-22

Reject